# Linguistic properties and model scale in brain encoding from small to compressed language models

Subba Reddy Oota [1]  Vijay Rowtula [* 2 3]  Satya Sai Srinath Namburi GNVV [* 4]  Khushbu Pahwa [5]
Anant Khandelwal [6]  Manish Gupta [6]  Tanmoy Chakraborty [7]  Bapi Raju Surampudi [2]

## Abstract

Recent work has shown that scaling large language models (LLMs) improves their alignment with human brain activity, yet it remains unclear what drives these gains and which representational properties are responsible. Although larger models often yield better task performance and brain alignment, they are increasingly difficult to analyze mechanistically. This raises a fundamental question: *what is the minimal model capacity required to capture brain-relevant representations?* To address this question, we systematically investigate how constraining model scale and numerical precision affects brain alignment. We compare full-precision LLMs, small language models (SLMs), and compressed variants (quantized and pruned) by predicting fMRI responses during naturalistic language comprehension. Across model families up to 14B parameters, we find that 3B SLMs achieve brain predictivity indistinguishable from larger LLMs, whereas 1B models degrade substantially, particularly in semantic language regions. Brain alignment is remarkably robust to compression: most quantization and pruning methods preserve neural predictivity, with GPTQ as a consistent exception. Linguistic probing reveals a dissociation between task performance and brain predictivity: compression degrades discourse, syntax, and morphology, yet brain predictivity remains largely unchanged. Overall, brain alignment saturates at modest model scales and is resilient to compression, challenging common assumptions about neural scaling and motivating compact models for brain-aligned language modeling.

---

[*]Equal contribution  [1]Technische Universität Berlin, Germany [2]IIIT Hyderabad, India [3]FICO, India [4]GE HealthCare, USA [5]Amazon, USA [6]Microsoft, India [7]IIT Delhi, India. Correspondence to: Subba Reddy Oota <subbareddyoota@gmail.com>.

*Proceedings of the 43$^{rd}$ International Conference on Machine Learning*, Seoul, South Korea. PMLR 306, 2026. Copyright 2026 by the author(s).

## 1. Introduction

Transformer-based language models (e.g., GPT*, BERT), although trained only on text, predict human brain activity to a remarkable degree, capturing neural responses during natural language comprehension across diverse cortical regions (Toneva & Wehbe, 2019; Schrimpf et al., 2021; Goldstein et al., 2022; Oota et al., 2022; Lamarre et al., 2022; Caucheteux & King, 2022; Antonello et al., 2021; Tuckute et al., 2023; Oota et al., 2024a). Recent work has further shown that this alignment improves as models scale, suggesting neural scaling laws analogous to those observed in language modeling (Kaplan et al., 2020; Hoffmann et al., 2022; Li et al., 2024; Matsuyama et al., 2023; Antonello et al., 2024; AlKhamissi et al., 2025). However, these studies have not examined why scaling improves alignment or which representational properties are responsible. Moreover, scaling comes at a steep computational cost (Faiz et al., 2024; Diaz & Madaio, 2024; Villalobos et al., 2024), making mechanistic analysis and controlled comparisons increasingly difficult. This raises a fundamental question: *if the human language system is compact and efficient, what model capacity is actually necessary to predict its neural responses?*

Within NeuroAI, a growing body of work examines how language is processed in the brain and how these processes compare to representations learned by language models. Although larger models often improve task performance and brain alignment, they pose increasing challenges for mechanistic analysis. We therefore use model size and numerical precision as controlled constraints to probe how representational changes impact brain alignment. In the AI community, two widely adopted strategies provide such controls: (i) compressing pretrained LLMs via pruning, distillation, or quantization, and (ii) training small language models (SLMs) that achieve competitive performance with far fewer parameters (Touvron et al., 2023; Yang et al., 2024; Guo et al., 2025).

Despite these advances, studies on brain encoding have overwhelmingly focused on large, full-precision models (Antonello et al., 2024; AlKhamissi et al., 2025), while compression is typically evaluated using engineering benchmarks

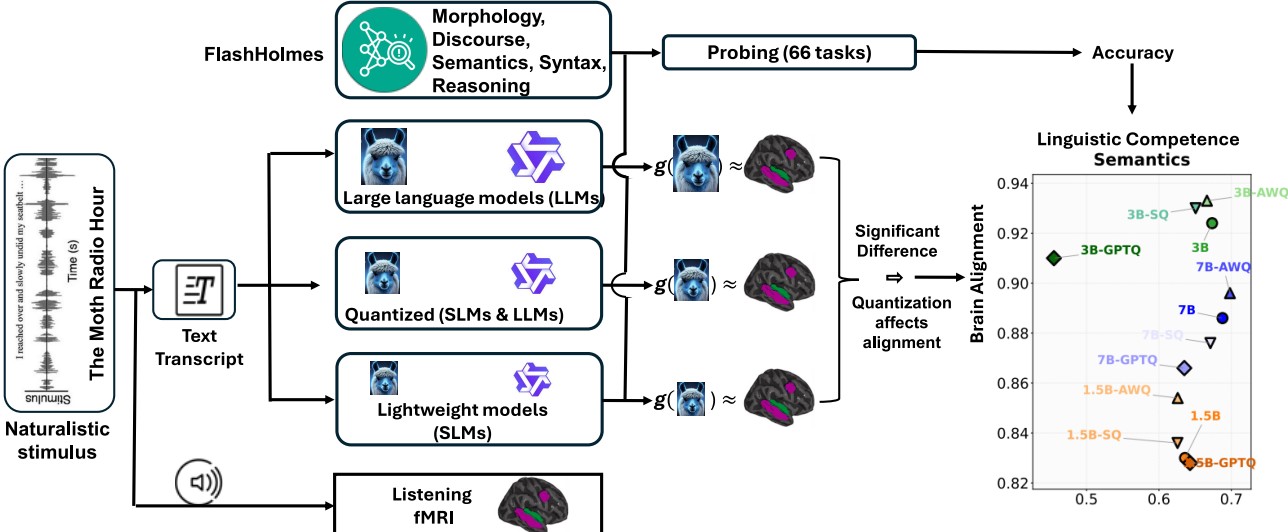

*Figure 1.* **Does linguistic competence drive brain–model alignment?** Participants listened to naturalistic English narratives while fMRI responses were recorded. Language models such as large, small, and their compressed variants, processed the same transcripts, and their internal representations were mapped to brain activity to quantify alignment. In parallel, the models were evaluated on the FlashHolmes benchmark to measure linguistic competence across morphology, syntax, semantics, discourse, and reasoning. By jointly comparing brain alignment and task performance across model scale and compression, we test whether reductions in linguistic competence induced by compression systematically degrade brain predictivity.

rather than neuroscientific criteria. As a result, it remains unclear whether small or compressed models preserve the brain-relevant representational geometry needed for accurate brain predictivity, making it uncertain whether they can be used as controlled probes of brain–model correspondence. Moreover, SLMs and compression methods may selectively impair linguistic competencies (e.g., discourse or syntax) in ways that are not reflected in aggregate task accuracy, raising the possibility of dissociation between linguistic competence and brain alignment. Hence, we ask the following research questions (RQs):

(1) What is the minimal model capacity required to achieve brain alignment comparable to larger LLMs, in both brain encoding and decoding settings?
(2) How do compression methods (quantization and pruning) affect brain alignment for both SLMs and LLMs?
(3) Which linguistic properties are preserved or degraded across small, large, and compressed models, and do these changes correlate with brain alignment?

To address these questions, we systematically investigate how model scale and compression jointly shape brain alignment and linguistic competence. Using fMRI recordings collected while participants listened to naturalistic stories from the Moth Radio Hour dataset (Deniz et al., 2019), we evaluate language models from three model families (LLaMA-3.2 (Touvron et al., 2023), Qwen2.5 (Yang et al., 2024), and DeepSeek-R1 (Guo et al., 2025)) spanning ≈1B–14B parameters. We quantify brain alignment in both (i) **en-**

**coding** (predicting voxel-wise fMRI responses from model representations) and (ii) **decoding** (reconstructing linguistic representations and downstream text from fMRI), thus showing that 3B models are sufficient not only for voxel-wise prediction, but also for reconstructing semantically coherent text from fMRI. We compare full-precision models against compressed variants, including multiple post-training quantization schemes (AWQ, GPTQ, SmoothQuant) and unstructured pruning. To assess linguistic competence independently of brain predictivity, we benchmark all models on FlashHolmes (Waldis et al., 2024), a streamlined suite probing morphology, syntax, semantics, discourse, and reasoning across 66 linguistic phenomena.

This unified framework enables us to test whether compact and compressed models match large LLMs in brain alignment, identify which linguistic properties are preserved or disrupted, and assess whether these changes impact neural predictivity. We evaluate models along two complementary dimensions: voxel-wise fMRI encoding performance and targeted linguistic probing.

Our findings lead to three overarching insights.

(1) Brain alignment saturates early: across model families, 3B SLMs match 7B–14B LLMs in neural predictivity across the whole brain and all major language-relevant regions, including semantic and integrative cortices, with consistent effects across subjects, whereas 1B models degrade substantially, especially in semantic regions. These results suggest that ∼3B parameters are sufficient to reach the saturation

regime of brain alignment, while ∼1B remains below the required capacity for robust brain alignment. In decoding, 3B SLMs enable stable brain-to-language reconstruction with semantic fidelity comparable to larger models, whereas 1B models show marked degradation.

(2) Brain alignment is remarkably robust to compression: most quantization and pruning methods preserve brain predictivity, except GPTQ, which consistently reduces it and produces widespread voxel-level degradation, especially in semantic brain regions (e.g., angular gyrus). Pruning remains stable up to moderate sparsity levels (10-25%), beyond which brain alignment degrades sharply, particularly for smaller models.

(3) Linguistic probing reveals a nuanced dissociation: compression disproportionately degrades discourse, syntax, and morphology-related competencies, yet these impairments do not consistently translate into reduced brain alignment, highlighting a divergence between task-level performance and brain-relevant representations.

Together, our results refine previous claims about neural scaling by demonstrating an early saturation of brain alignment at modest model scales and a striking robustness to compression. These findings position compact and efficiently compressed language models not only as engineering compromises but as principled and cognitively grounded alternatives for brain-aligned language modeling. We make the code publicly available[1]

A detailed discussion of related work on brain–language model alignment, neural scaling, and model compression is provided in Appendix A.

## 2. Methodology

### 2.1. Naturalistic Brain Imaging Dataset

We use a publicly available fMRI dataset (Deniz et al., 2019) collected while nine participants listened to narrative stories from the Moth Radio Hour. The dataset comprises 3,737 training and 291 testing samples (TRs: Repetition Time). These were specifically selected for their ability to elicit unique auditory and high-level linguistic responses in the brain. Following Deniz et al. (2019), we examine this dataset using the Glasser Atlas multi-modal parcellation of the cerebral cortex, targeting 180 ROIs per hemisphere (Glasser et al., 2016).

This includes one early sensory processing region (early auditory) and eight language-relevant regions spanning semantic, syntactic, and discourse-level processing, including the angular gyrus (AG), lateral temporal cortex (ATL and PTL),

[1] https://github.com/subbareddy248/slm-brain-alignment

*Table 1.* Pretrained Transformer-based SLMs and LLMs, and their post-training compressed variants. SQ=SmoothQuant. Config shows #layers and #parameters. Other columns show sizes in GBs.

| Model Family | SLMs | | | | LLMs | | | | |
|---|---|---|---|---|---|---|---|---|---|
| | Config | Orig | AWQ | GPTQ | SQ | Config | Orig | AWQ | GPTQ | SQ |
| LLaMA 3.2 | 16L, 1B | 2.47 | 1.56 | 1.02 | 2.02 | 28L, 8B | 16.1 | 5.73 | 5.70 | 9.08 |
| | 28L, 3B | 6.4 | 3.04 | 2.26 | 4.40 | | | | | |
| Qwen 2.5 | 28L, 1.5B | 3.1 | 1.61 | 1.15 | 2.25 | 28L, 7B | 15.1 | 5.57 | 5.58 | 8.67 |
| | 36L, 3B | 6.1 | 2.69 | 2.10 | 4.02 | | | | | |
| DeepSeek | 28L, 1.5B | 3.55 | 1.62 | 1.61 | 2.25 | 28L, 7B | 15.23 | 5.57 | 5.58 | 8.71 |
| | 28L, 3B | 6.43 | 2.69 | 2.37 | 4.02 | | | | | |

inferior frontal gyrus (IFG and IFGOrb), middle frontal gyrus (MFG), posterior cingulate cortex (PCC) and dorsomedial prefrontal cortex (dmPFC), based on Fedorenko's language parcels (Milton et al., 2021; Desai et al., 2023). These regions are central to the semantic and integrative effects examined in our analysis. More details about dataset and ROI functionality are in Appendix B and Table 6.

**Estimating cross-subject prediction accuracy.** To account for intrinsic noise in biological measurements, we adapt the method proposed by Schrimpf et al. (2021); Oota et al. (2024a); AlKhamissi et al. (2025) to estimate the ceiling value for a model's performance for the Subset-Moth-Radio-Hour fMRI dataset. Note that the estimated cross-subject prediction accuracy is based on the assumption of a perfect model, which might differ from real-world scenarios, yet offers valuable insights into model's performance. We present the average cross-subject prediction accuracy across voxels for the *listening fMRI* dataset in Appendix C.

### 2.2. SLMs and their larger counterparts

To investigate whether small and compressed language models align with human language processing in the brain, we consider multiple modern model families spanning scales from 1B to 14B parameters which are publicly available on Huggingface (Wolf et al., 2020), including LLaMA-3.2 (Touvron et al., 2023), Qwen-2.5 (Yang et al., 2024) and DeepSeek-R1 (Guo et al., 2025). We evaluate both SLMs; 1B–3B and their larger counterparts (7B–14B), along with post-training compressed variants. We report the model parameters and layer details in Table 1. All models are base (non–instruction-tuned) checkpoints to avoid task-specific fine-tuning effects. For each model, we extract representations from all transformer layers and select the single best-performing layer per model, ensuring comparability across architectures with different depths.

**Extracting text representations.** For text transcripts from the Moth Radio Hour dataset, we follow previous work in extracting hidden-state representations from each layer of the language models for a fixed-length input (Toneva & Wehbe, 2019; Aw & Toneva, 2023; Oota et al., 2024a; 2025). To obtain the stimulus features from these pretrained mod-

els, we apply a sliding input window over the text: for each target word $w_t$, we construct an input sequence consisting of the preceding $C$ (=20) words (or fewer at the start of the narrative), tokenize this sequence using the model's standard tokenizer, and pass it through the pretrained model. The choice of 20-word context length follows established practice in brain-language alignment, where prior work systematically varying context length in fMRI encoding models has shown that encoding performance saturates at moderate context lengths with diminishing returns beyond ∼20 (Jain & Huth, 2018; Toneva & Wehbe, 2019; Aw & Toneva, 2023; Oota et al., 2024a). For instance, given a story of $M$ words and considering the context length of 20, while the third word's vector is computed by presenting ($w_1$, $w_2$, $w_3$) as input to the network, the last word's vector $w_M$ is computed by presenting the network with ($w_{M-20}$, ..., $w_M$). The pretrained Transformer model outputs token representations at different layers. We use the #words $\times d$ hidden-state representations from each layer, where $d$ is the model-specific hidden dimension, to obtain word-level representations from each pretrained Transformer language model.

The preprocessing and HRF delays are detailed in Appendix B. Following prior work, we evaluate representations from all layers and report results for the best-performing layer per model, ensuring fair comparison across architectures with differing depths and hidden dimensions.

### 2.3. Post-training Compression: Quantization and Pruning

Quantization can dramatically reduce memory usage and accelerate inference by mapping model weights and activations to lower-precision formats such as INT8, INT4, or FP8 arithmetic. In this work, our goal is not only to assess efficiency, but also to examine how compression, including quantization and unstructured pruning, impacts linguistic competence and brain alignment. To achieve this, we perform three widely-used quantization techniques: (1) Activation-aware Weight Quantization (AWQ) (Lin et al., 2024), which adjusts weight scales using activation statistics to enable highly accurate 4-bit or 8-bit compression with minimal quality loss; (2) GPTQ (Frantar et al., 2023): applies the post-training, gradient-guided weight quantization method that delivers near-lossless INT8/INT4 speed-ups; and (3) SmoothQuant (Liu et al., 2024; Xiao et al., 2023), which jointly quantizes weights and activations by equalizing variance to reduce memory usage and latency while preserving accuracy. We provide more details, including model parameters after quantization in Table 1.

**Unstructured pruning.** To study the effect of parameter removal on brain alignment, we apply post-training unstructured magnitude-based pruning. Specifically, we prune individual weights with the smallest absolute values (L1

norm) across all linear layers, following standard practice in language model compression. We evaluate multiple sparsity levels (10%, 25%, and 50%), without retraining or fine-tuning after pruning, to isolate the effect of parameter removal on representational geometry.

### 2.4. Flash-Holmes Benchmark

FlashHolmes is a streamlined version of the Holmes benchmark (Waldis et al., 2024), designed to efficiently evaluate the linguistic competence of language models. The Flash-Holmes benchmark covers nearly 200 probing datasets that span 66 linguistic tasks. The linguistic tasks are grouped into five major categories: (1) Morphology (19 tasks, e.g., subject–verb agreement and irregular word forms) (Warstadt et al., 2020; Huebner et al., 2021), (2) Syntax (75 tasks, e.g., constituent labeling and filler-gap dependencies) (Conneau et al., 2018; Warstadt et al., 2020), (3) Semantics (67 tasks, e.g., semantic role labeling and natural language inference) (Wang et al., 2018), (4) Discourse (28 tasks, e.g., coreference resolution and discourse relation prediction) (Webber et al., 2019), and (5) Reasoning (19 tasks, e.g., paraphrasticity with negation and antonyms) (Vahtola et al., 2022). Overall, these linguistic tasks allow FlashHolmes to probe a wide spectrum of linguistic phenomena, making it a suitable tool for evaluating both SLMs and LLMs and their quantized variants.

## 3. Experimental Setup

**Models and compression.** We evaluate three modern Transformer-based model families: Qwen2.5, LLaMA-3.2, and DeepSeek-R1, spanning scales from 1B to 14B parameters. For each family, we include small language models (SLMs; 1B–3B) and larger counterparts (7B–14B), all using pretrained base checkpoints. In addition to full-precision models, we evaluate post-training compressed variants using quantization and pruning. All compression methods, sparsity levels, and implementation details are described in Section 3.3. Unless otherwise stated, compressed models are evaluated without retraining using the same encoding and probing pipelines as dense models. We note that all models in this study are pretrained with next-token prediction only, with no post-training such as RLHF; we use base, not instruction-tuned, checkpoints.

**Voxel-wise encoding model.** To perform voxel-wise encoding, we train an fMRI encoding model using bootstrap ridge regression (Tikhonov & Arsenin, 1977) to predict the fMRI recording associated with each voxel as a function of the stimulus representations obtained from the language models. Before the bootstrap ridge regression, we first z-score each feature channel separately for training and testing. This is done to match the features to the fMRI responses,

which were also z-scored for training and testing. Formally, at the time step (t), we encode the stimuli as $X_t \in \mathbb{R}^{N \times D}$ and brain region voxels $Y_t \in \mathbb{R}^{N \times V}$, where $N$ is the number of training examples, $D$ denotes the dimension of the concatenation of delayed 4 TRs, and $V$ denotes the number of voxels. Before fitting the encoding model, we applied PCA to each model's features and reduced them to a shared dimensionality of 1024. This ensured that all models were compared using the same feature dimension, making the encoding comparisons fairer across scales. To find the optimal regularization parameter for each feature space, we use a range of regularization parameters that is explored using cross-validation. The main goal of each fMRI encoding model is to predict brain responses associated with each brain voxel given a stimulus. Following prior work, we train encoding models using representations from all layers and report results for the best-performing layer per model, ensuring fair comparison across architectures with different depths. The detailed hyperparameter settings and statistical significance tests are provided in Appendix D and E.

**Brain decoding.** In addition to encoding, we perform brain decoding experiments that reconstruct linguistic representations and text from fMRI using the same model representations and alignment framework, as discussed in Appendix N.

**Normalized alignment.** The final measure of a model's performance is obtained by calculating Pearson's correlation between the model's predictions and brain recordings. This correlation is then divided by the estimated cross-subject prediction accuracy and averaged across voxels, resulting in a standardized measure of performance referred to as normalized alignment. For normalized alignment, we restrict analyses to voxels with cross-subject prediction accuracy $\geq$ 0.05, ensuring that comparisons focus on reliably stimulus-driven responses.

**Encoding and probing of Flash-Holmes benchmark.** To evaluate the linguistic competence of language models, we use the FlashHolmes benchmark (Waldis et al., 2024), as detailed in Section 2.4. For each task in FlashHolmes, we apply classifier-based probing to the internal representations of SLMs, LLMs, and quantized models. Our hypothesis is that SLMs or quantized models may show reduced linguistic competence on specific tasks, even if overall brain alignment remains intact. In this case, FlashHolmes can reveal which linguistic properties (e.g., syntax or discourse) are disproportionately affected by compression. Thus, Flash-Holmes plays a complementary but essential role: it allows us to determine whether reductions in linguistic competence under compression correspond to reductions in brain alignment, or whether these two measures dissociate. To perform probing, following Waldis et al. (2024), we train a simple linear classifier on the representations obtained from SLMs,

LLMs, and quantized models. Model performance is then aggregated across tasks within each linguistic category, producing scores that reflect the accessibility of different types of linguistic information in the representations.

Overall, by treating compression as a controlled intervention on internal representations, probing analyses help isolate which linguistic properties are preserved or disrupted without directly manipulating model parameters.

## 4. Results

### [RQ1]: Effects on Brain Encoding Performance: 3B SLMs match similar brain alignment as 7B–14B models across model families

To examine whether SLMs achieve brain encoding performance comparable to larger models, we compare the voxelwise encoding performance of three families of language models (Qwen-2.5, LLaMA-3.2, and DeepSeek-R1), at different scales. For both SLMs and LLMs, we also apply post-training quantization and measure normalized brain predictivity at both the whole-brain level and within language-specific regions. Fig. 2 shows average normalized brain predictivity across participants and layers.

*Table 2.* Pairwise differences in Qwen2.5 best-layer encoding scores across 9 subjects. For each subject and model, we take the maximum encoding score across evaluated layers and then compute paired $t$-tests between models. $\Delta$ is the mean difference A–B over subjects.

| Comparison (A–B) | $\Delta$ | $t(8)$ | $p$ (two-sided, approx.) | Interpretation |
|---|---|---|---|---|
| 3B – 14B | 0.000 | -0.00 | 1.00 | 3B $\approx$ 14B (no difference) |
| 3B – 7B | 0.028 | 2.43 | 0.06 | 3B > 7B (small effect, trend) |
| 3B – 1.5B | 0.073 | 4.89 | 0.004 | 3B > 1.5B (clear, significant) |
| 14B – 7B | 0.028 | 2.02 | 0.10 | 14B > 7B (small effect, n.s.) |
| 14B – 1.5B | 0.073 | 3.16 | 0.025 | 14B > 1.5B (clear, significant) |
| 7B – 1.5B | 0.045 | 2.67 | 0.045 | 7B > 1.5B (moderate, significant) |

**Whole-brain analysis.** Across whole brain (Fig. 2, top row), 3B SLMs in all three families (Qwen, LLaMA, DeepSeek) achieve normalized brain alignment comparable to their 7B–14B counterparts, indicating saturation of encoding performance beyond the 3B scale. Focusing on Qwen2.5 (Table 2), subject-wise paired $t$-tests ($n = 9$) show no reliable difference between 3B and 14B ($\Delta = 0.000$, $t(8) = -0.03$, $p = 1.0$), while both 3B and 14B significantly outperform 1.5B (3B vs. 1.5B: $\Delta = 0.07$, $t(8) = 4.89$, $p = 0.004$; 14B vs. 1.5B: $\Delta = 0.07$, $t(8) = 3.16$, $p = 0.025$). We also observe a small but significant advantage of 3B and 14B over 7B in best-layer alignment ($\Delta \approx 0.04$, $p \approx 0.02$–0.04). Overall, these results support our main claim in this regime: scaling beyond $\sim$3B yields at most modest gains in brain alignment, whereas $\sim$1–1.5B models are reliably worse. We observe the same qualitative pattern for LLaMA-3.2 and DeepSeek-R1 (Tables 9 and 10 in Appendix I).

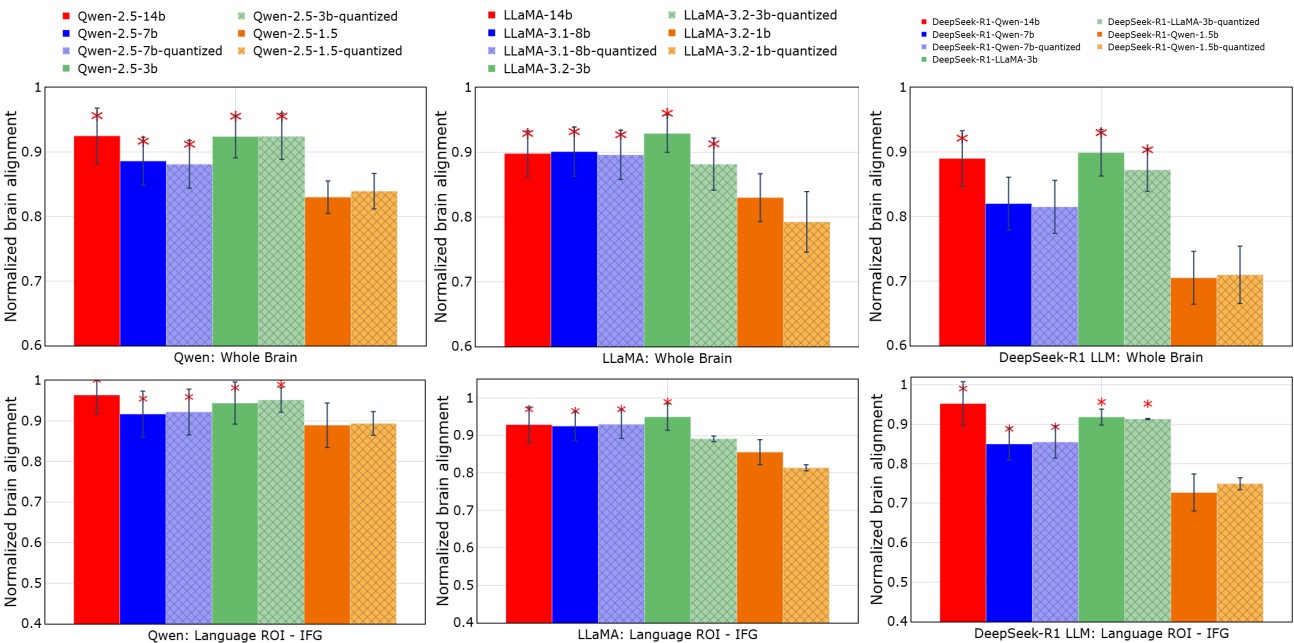

*Figure 2.* Qwen2.5, LLaMA, and DeepSeek-R1: Normalized brain alignment was computed by averaging across participants, layers, and voxels. Red: 14b, Blue: 7b, Green: 3b, Orange: 1.5b, Solid: full-precision SLMs/LLMs, Patterned: quantized models. * at a particular bar indicates that the model's prediction performance is significantly better than 1b/1.5b SLMs. The top row shows whole-brain normalized alignment, while the bottom row focuses on a language-selective ROI (IFG).

*Table 3.* Overall performance metrics for brain-to-text decoding.

| Model | BLEU-1 | WER | METEOR | BERT-F1 | Samples |
|---|---|---|---|---|---|
| LLaMA-3-8B | 0.0699 | 5.7839 | 0.0550 | 0.8108 | 784 |
| LLaMA-3.2-3B | 0.1198 | 4.2237 | 0.1101 | 0.8252 | 784 |
| LLaMA-3.2-1B | 0.1105 | 4.4869 | 0.0990 | 0.8237 | 784 |

Fig. 2 aggregates post-training quantization results by averaging across AWQ, GPTQ, and SmoothQuant, and reports whole-brain alignment averaged across the three model families. Overall, quantization largely preserves whole-brain alignment for both LLMs and 3B SLMs, whereas the smallest 1B–1.5B models show a significant drop in alignment across compression settings ($p < 0.01$).

**Language-ROI analysis.** Within language-selective regions, 3B SLMs and larger LLMs achieve high alignment that is largely preserved under quantization (Fig. 2, bottom). In contrast, 1B–1.5B models show pronounced drops, indicating that sub-3B capacity is insufficient even within core language circuitry. Scale sensitivity is strongest in integrative/semantic regions including angular gyrus, PCC, and dmPFC, and weaker in ATL, PTL, and MFG, where alignment varies comparatively little with model size. Additional ROI results are shown in Figs. 11, 13, and 29.

**Decoding performance: 3B SLMs enable stable brain-to-language reconstruction** We perform end-to-end text stimulus reconstruction from fMRI brain activity. We follow the BrainLLM methodology inspired from Ye et al. (2025), where we use the same Subset-Moth-Radio-Hour dataset (11 stories) with the same train/test split, where ten stories are used for training and one held-out story is used for generation. Concretely, we train a brain-to-text decoder and report standard text-generation metrics-BLEU-1, WER, METEOR, and BERT-F1-for three models: LLaMA-3-8B, LLaMA-3.2-3B, and LLaMA-3.2-1B (Table 3). Across reconstructed segments per model on test dataset, LLaMA-3.2-3B achieves the best performance on all four metrics (BLEU-1 = 0.120, WER = 4.22, METEOR = 0.110, BERT-F1 = 0.825), slightly outperforming LLaMA-3-8B and clearly improving over the LLaMA-3.2-1B baseline (BLEU-1 = 0.070, METEOR = 0.055, BERT-F1 = 0.811). These BERT-F1 scores in the 0.81–0.83 range indicate that the decoded text reliably preserves the semantic content of the original stimulus, while BLEU-1 in the 0.07–0.12 range is in line with prior work where exact word-level recovery from fMRI is known to be challenging. We also include qualitative examples comparing ground-truth text and decoded outputs, and a decoding analysis across ROIs, in Appendix N (Tables 21 and 22). These examples illustrate that the decoder often recovers the overall meaning, emotional tone, and discourse context, even when individual words differ-e.g., reconstructions that correctly express embarrassment, uncertainty, or interactions with children, despite not matching every token verbatim.

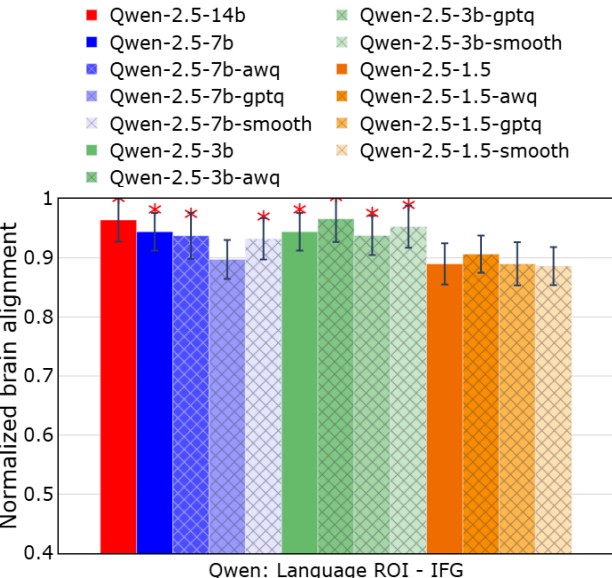

Qwen-2.5-14b
Qwen-2.5-7b
Qwen-2.5-7b-awq
Qwen-2.5-7b-gptq
Qwen-2.5-7b-smooth
Qwen-2.5-3b
Qwen-2.5-3b-awq
Qwen-2.5-3b-gptq
Qwen-2.5-3b-smooth
Qwen-2.5-1.5
Qwen-2.5-1.5-awq
Qwen-2.5-1.5-gptq
Qwen-2.5-1.5-smooth

*Figure 3.* Normalized brain alignment averaged across participants and voxels, using the best-performing layer for Qwen2.5 model. Red: 14b, Blue: 7b/8b, Green: 3b, Orange: 1.5b, Solid: full-precision SLMs/LLMs, Patterned: quantized models. * at a particular bar indicates that the model's prediction performance is significantly better than 1b/1.5b SLMs. Plots for other model families and regions are in Figs. 10 and 12 in Appendix F.

*Table 4.* Pairwise comparisons of brain-alignment differences across quantization methods for Qwen2.5 model. The Table reports mean differences ($\Delta$), $t$-statistics, and two-sided significance tests for 7B (left), 3B (right), and 1.5B (bottom).

| Comparison (A–B) | $\Delta$ | $t(8)$ | Sig. |
|---|---|---|---|
| Qwen2.5-7B–AWQ | -0.020 | -6.10 | $p < 0.001$ |
| Qwen2.5-7B–GPTQ | 0.020 | 6.20 | $p < 0.001$ |
| Qwen2.5-7B–SmoothQuant | -0.005 | -3.50 | $p < 0.016$ |
| AWQ–GPTQ | 0.040 | 7.10 | $p < 0.001$ |
| AWQ–SmoothQuant | 0.015 | 4.20 | $p < 0.008$ |
| GPTQ–SmoothQuant | -0.025 | -4.90 | $p < 0.004$ |

**Encoding performance on the Reading fMRI dataset.**
To test generalization across paradigms, we additionally perform voxel-wise encoding on the Subset-Moth-Radio-Hour *Reading* fMRI dataset (Deniz et al., 2019) (same nine subjects, different task i.e. Reading). Using Qwen2.5 models (1.5B/3B/7B/14B), Fig. 30 shows that 3B SLMs achieve alignment comparable to 7B and 14B, while 1.5B exhibits a clear drop. Overall, the 3B saturation pattern generalizes to the reading setting.

## [RQ2]: Most quantization and pruning methods preserve brain alignment, except GPTQ,

**Effect of quantization on brain encoding.** Fig. 3 shows Qwen2.5 encoding in IFG across sizes and quantization methods. At 3B SLMs, post-training quantization (AWQ, GPTQ, SmoothQuant) largely preserves alignment with only marginal changes from FP16, whereas 1B–1.5B models re-

*Table 5.* Comparison of quantization and pruning for Qwen2.5-3B.

| Model variant | Method | Sparsity | Normalized Brain Alignment |
|---|---|---|---|
| Qwen-2.5-3B | FP16 (baseline) | 0% | $0.924 \pm 0.033$ |
| Qwen-2.5-3B-AWQ | Quantization (AWQ) | 0% | $0.933 \pm 0.035$ |
| Qwen-2.5-3B-GPTQ | Quantization (GPTQ) | 0% | $0.910 \pm 0.037$ |
| Qwen-2.5-3B-Smooth | Quantization (SmoothQuant) | 0% | $0.930 \pm 0.035$ |
| Qwen-2.5-3B-0.1 | Pruning | 10% | $0.910 \pm 0.032$ |
| Qwen-2.5-3B-0.25 | Pruning | 25% | $0.908 \pm 0.033$ |
| Qwen-2.5-3B-0.5 | Pruning | 50% | $0.907 \pm 0.043$ |

main under-aligned, reinforcing that their limited capacity, rather than representational redundancy, is the primary bottleneck. We quantify these effects with subject-wise best-layer scores and paired $t$-tests across methods (Table 4). For 7B, AWQ and SmoothQuant significantly outperform FP16 and GPTQ, and GPTQ is significantly worse than FP16. For 3B, quantized variants do not differ significantly from FP16, but AWQ/SmoothQuant significantly outperform GPTQ, indicating that well-designed quantization preserves alignment while GPTQ induces modest degradation. For 1.5B, AWQ improves over FP16, whereas GPTQ and SmoothQuant do not differ reliably from FP16 (and differences among quantized variants are not significant after correction). We observe the similar quantization effects for LLaMA-3.2 and DeepSeek-R1 (Tables 11 in Appendix J).

**Qualitative voxel-wise changes.** Fig. 4 visualizes voxel-wise percentage changes in brain alignment across scale and quantization. (i) Scaling down from LLMs to 3B SLMs (Fig. 4 (a)): reductions in brain alignment are negligible in the bilateral temporal lobe and remain under 5% in the parietal cortex and IFGorb. Large cortical regions remain white, indicating that 3B SLMs preserve brain-relevant representations comparable to LLMs. The blue-marked voxels are sparse and localized, suggesting only limited information loss. (ii) 3B SLMs to 3B SLMs GPTQ (Fig. 4 (b)): applying GPTQ leads to widespread orange-marked voxels, reflecting consistent losses across distributed cortical regions. While some areas remain preserved (white voxels), the extent of information loss is greater than that observed with downscaling alone, confirming that GPTQ disproportionately disrupts brain-relevant alignment. (iii) 3B SLMs to 1.5B SLMs (Fig. 4 (c)): relative to the 3B baseline, we observe extensive red-marked voxels, especially in temporal and language-related regions, indicating larger drops in alignment. This demonstrates the limits of scaling, as ultra-small models fail to capture brain-relevant representations. (iv) 3B SLMs → 3B SLMs AWQ (Fig. 4 (d)): AWQ produces localized green-marked voxels, indicating regions of improved alignment relative to the uncompressed 3B baseline, particularly in the IFG and AG. Most of the cortex remains unchanged (white), suggesting that AWQ maintains representational fidelity while offering modest regional gains. Results for other participants for Qwen2.5 and LLaMA models are in Appendix G.

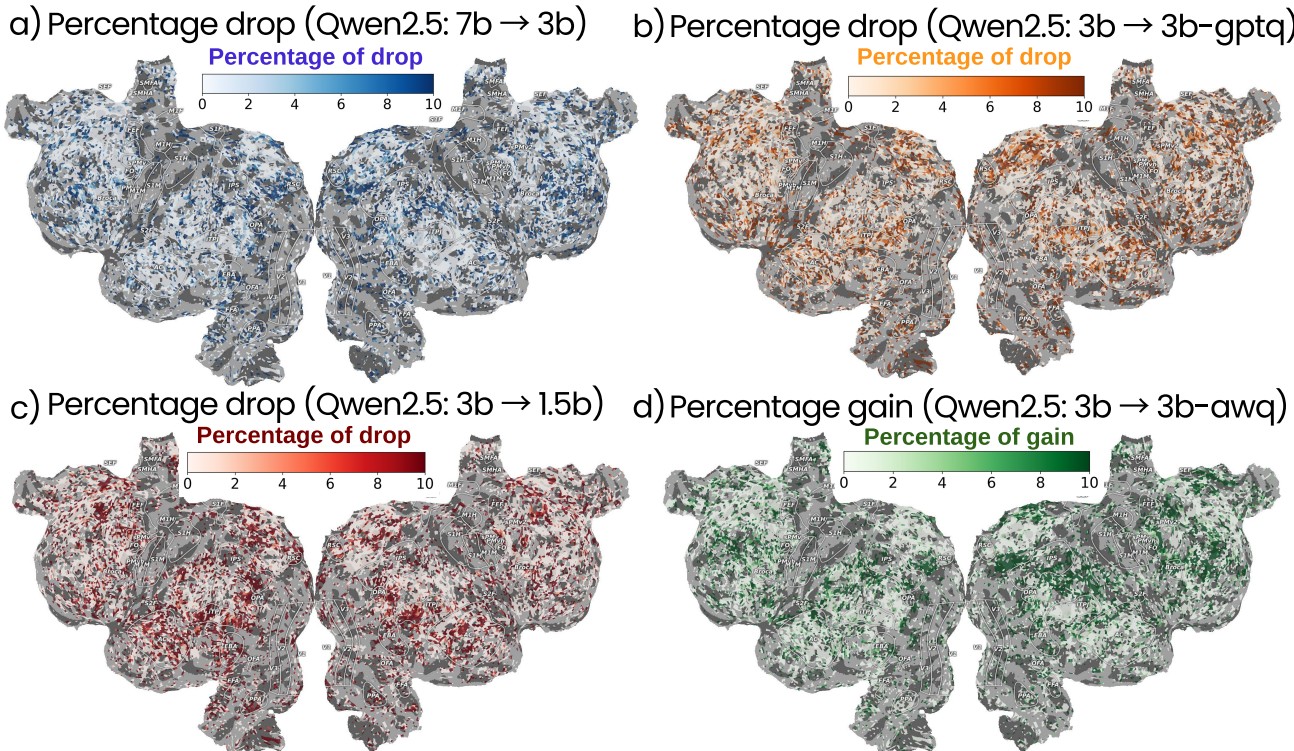

*Figure 4.* Qwen2.5: Percentage change in brain alignment across model scales and quantization methods, shown on the flattened cortical surface of a representative subject (subject-5). Blue, orange, and red voxels indicate regions of information loss ((a) LLMs → 3B SLMs, (b) 3B SLMs → 3B SLMs GPTQ, (c) 3B SLMs → 1.5B SLMs, respectively), (d) while green voxels highlight regions of improvement for 3B SLMs AWQ over 3B SLMs. White voxels denote regions with no change. Results for other participants for Qwen2.5 and LLaMA models are in Appendix G.

Taken together, these results support Hypothesis 1: 3B SLMs achieve brain alignment comparable to larger LLMs and remain robust under AWQ and SmoothQuant quantization. By contrast, 1B–1.5B models consistently underperform, and GPTQ disproportionately disrupts brain-relevant alignment. Importantly, this pattern holds across both the Qwen and LLaMA model families.

**Pruning preserves alignment up to moderate sparsity (10–25%), but degrades at high sparsity for 1.5B.** We evaluate unstructured magnitude pruning of linear layers at sparsity levels 0.10, 0.25, and 0.50 and report normalized brain alignment (Table 5). For Qwen2.5-3B, alignment remains in a narrow range under pruning up to 50% (0.907–0.910; ± 0.032–0.043), comparable to FP16 and post-training quantization (AWQ/SmoothQuant: 0.930–0.933 vs. FP16: 0.924; GPTQ: 0.910). In contrast, for Qwen2.5-1.5B (see Table 23 in Appendix O), pruning at 10–25% largely preserves alignment, whereas 50% pruning yields a marked drop. Overall, these results complement our quantization findings: moderate pruning and quantization can preserve brain alignment surprisingly well, but aggressive sparsification can harm alignment, especially in smaller models, and may introduce trade-offs with linguistic competence.

**[RQ3]: Impact of linguistic competence in language models and brains**

While previous analyses show that 3B SLMs match LLMs in brain alignment, alignment drops for ∼1B models and under GPTQ. To further investigate the linguistic competence of SLMs, LLMs, and their quantized counterparts, and to examine whether linguistic competence influences brain alignment, we benchmark these models on FlashHolmes and analyze brain alignment trends across linguistic tasks.

Figs. 5 illustrate how scaling and quantization affect the relationship between linguistic competence and brain alignment across SLMs, LLMs, and their quantized variants. We make the following observations: For LLMs and 3B SLMs, brain alignment is largely preserved under AWQ and SmoothQuant, while GPTQ consistently degrades discourse/reasoning/morphology and also reduces brain alignment, suggesting disproportionately degrades higher-order linguistic skills. In contrast, 3B SLMs generally maintain (and sometimes improve) FlashHolmes performance while keeping alignment comparable to FP16, indicating that well-designed quantization can preserve both linguistic competence and brain-relevant representations. Finally, 1B models (and their quantized variants) retain task performance across

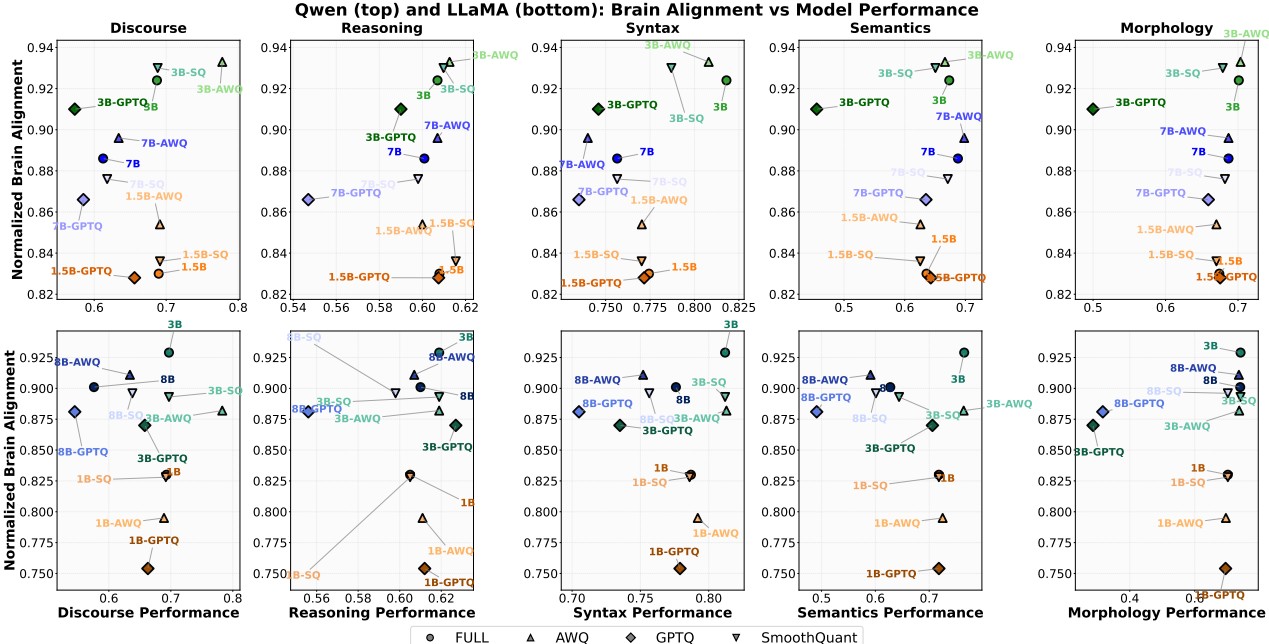

*Figure 5.* Tradeoff between normalized brain alignment and linguistic competence performance on FlashHolmes Tasks (Qwen and LLaMA Model Families). Blue: 7b/8b, Green: 3b, Orange: 1.5b.

linguistic categories but show markedly lower brain alignment than 3B, revealing a clear dissociation between task accuracy and neural alignment.

Overall, our analysis reveals a dissociation between linguistic competence and brain alignment across SLMs, LLMs, and their quantized variants. AWQ and SmoothQuant largely preserve both neural predictivity and probing performance, whereas GPTQ tends to reduce both—particularly higher-order competencies such as discourse, reasoning, and syntax/morphology. In contrast, ∼1B SLMs illustrate the limits of scale: they can maintain task performance while failing to capture brain-relevant representational structure. Tables 7 and 8 in Appendix H report detailed FlashHolmes results across five competence categories for SLMs/LLMs and their quantized variants.

## 5. Discussion and Conclusion

We comprehensively evaluate large and small language models, along with compressed variants, for fMRI-based brain encoding during naturalistic language comprehension. Across model families and scales, brain alignment saturates at modest sizes: 3B SLMs match 7B–14B LLMs, whereas 1B–1.5B models consistently underperform. Notably, this saturation largely persists under post-training compression.

Our compression analyses show that brain alignment is largely robust to post-training efficiency methods. AWQ and SmoothQuant preserve near-baseline alignment, whereas

GPTQ produces consistent losses, especially in semantic and discourse-related regions. This underscores that compression methods are not interchangeable: they perturb brain-relevant representations in qualitatively different ways. Finally, combining voxel-wise encoding with large-scale linguistic probing reveals a dissociation between linguistic competence and brain alignment. Compression can degrade specific linguistic skills without impacting brain alignment, while ultra-small models may retain task performance yet fail to capture brain-relevant representations. Lastly, we discuss limitations in Appendix Q.

## Impact Statement

This work advances brain–language model alignment research by identifying the model scale and compression regimes sufficient for capturing brain-relevant language representations during naturalistic comprehension. We show that brain alignment saturates at modest model sizes and remains robust to most post-training quantization methods and moderate levels of pruning, challenging the assumption that increasingly large, full-precision models are necessary for modeling neural language processing.

Scientifically, these findings motivate the use of smaller and more interpretable models as principled tools for studying brain–language computations, and position compression as a controlled intervention for probing which linguistic properties are essential for neural alignment. Practically,

our results support compact and efficiently compressed language models as accessible foundations for fMRI-based brain encoding and decoding analyses, potentially lowering computational barriers and improving reproducibility in NeuroAI research.

This work does not involve the collection of new neural data and relies exclusively on the publicly available Subset-Moth-Radio-Hour dataset (https://gin.g-node.org/denizenslab/narratives_reading_listening_fmri). We do not foresee direct harmful applications arising from this work.

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

# Overview of Appendix Sections

## A. Related work

**Small and compressed language models.** Scaling laws characterize how task performance improves with model size, data, and compute (Kaplan et al., 2020; Hoffmann et al., 2022; Diaz & Madaio, 2024). In parallel, recent work has produced strong small language models (SLMs) and efficient model variants (e.g., LLaMA and Qwen families, Gemma, DeepSeek) that achieve competitive NLP performance at modest parameter counts (Touvron et al., 2023; Yang et al., 2024; Gemma Team et al., 2024; Guo et al., 2025). For NeuroAI, these models offer a complementary opportunity: constraining capacity provides a controlled way to probe which representational properties are sufficient for brain alignment.

A related line of work studies compression, including pruning, distillation, and post-training quantization (Gupta & Agrawal, 2022). Post-training quantization methods such as AWQ, GPTQ, and SmoothQuant can substantially reduce memory and compute while often preserving benchmark accuracy (Lin et al., 2024; Frantar et al., 2023; Xiao et al., 2023; Namburi et al., 2023; Kuzmin et al., 2023). However, compression is typically evaluated with engineering metrics, and its consequences for brain alignment and neural predictivity remain underexplored. Our work fills this gap by systematically testing model scale and compression (quantization and pruning) as controlled perturbations and evaluating their effects on neural predictivity and linguistic competence.

**Scaling laws for brain encoding.** A growing body of work has demonstrated that representations from large language models can predict brain activity evoked by text and speech with high fidelity (Antonello et al., 2024; Matsuyama et al., 2023; AlKhamissi et al., 2025). In a seminal study, Antonello et al. (2024) compared small and large models (from OPT-125M to OPT-175B and LLaMA-66B), showing that larger models yield substantial gains in fMRI encoding performance. Subsequent work by AlKhamissi et al. (2025) identified language-selective networks within LLMs that mirror functional specialization in the human brain. Together, these studies establish that scale matters for brain encoding under full-precision settings. Our work departs from this line of research by showing that brain alignment does not grow monotonically with scale: instead, it saturates at relatively modest model sizes and remains robust under substantial compression. By extending scaling analysis to efficiency-oriented models, including compressed LLMs and modern SLMs across multiple families, we reveal qualitative differences in how scale and compression affect brain-relevant representations.

**Linguistic competence and neural alignment.** Prior work has also examined which linguistic properties of language models are predictive of brain activity. Some studies adopt direct interventions on model representations—such as residualization or feature ablation—to estimate the causal contribution of specific linguistic features to neural alignment (Toneva et al., 2022b; Oota et al., 2024b;a; Srijith et al., 2025). Other studies follow an indirect approach, first measuring a model's brain predictivity and then relating it to the performance on linguistic tasks (Schrimpf et al., 2021; Goldstein et al., 2022). Our work is complementary to both paradigms. Rather than manipulating representations directly, we use compression as a natural intervention on representational geometry and examine its effects on both brain alignment and linguistic competence across a wide

spectrum of tasks. This approach allows us to uncover the dissociation between task-level linguistic performance and neural predictivity, revealing which linguistic competencies are critical for brain alignment and which can degrade without disrupting it.

## B. Naturalistic Listening fMRI Dataset

We use the publicly available naturalistic story listening fMRI dataset provided by (Deniz et al., 2019). The dataset consists of 11 stories, 9 participants, and all participants listened to all the stories. The speech stimuli consisted of 10- to 15 min stories taken from The Moth Radio Hour and used previously (Huth et al., 2016). The 10 selected stories cover a wide range of topics and are highly engaging. The model validation dataset consisted of one 10 min story. All stimuli were played at 44.1 kHz using the pygame library in Python. The audio of each story was down-sampled to 11.5 kHz and the Penn Phonetics Lab Forced Aligner was used to automatically align the audio to the transcript. Finally the aligned transcripts were converted into separate word and phoneme representations using Praat's TextGrid object. The word representation of each story is a list of pairs (W, t), where W is a word and t is the onset time in seconds.

The total number of words in each story as follows: Story1: 2174; Story2: 1469; Story3: 1964; Story4: 1893; Story5: 2209; Story6: 2786; Story7: 3218; Story8: 2675; Story9: 1868; Story10: 1641; Story11: 1839 (test dataset)

To align the stimulus presentation rate with the slower fMRI data acquisition rate (TR = 2.0045 sec), where multiple words correspond to a single TR, we downsample the stimulus features to match fMRI recording times using a 3-lobed Lanczos filter (Duchon, 1979), thus creating chunk-embeddings for each TR. To account for the slowness of the hemodynamic response (HRF), we model HRF using a finite response filter (FIR) per voxel and for each subject separately with 4 temporal delays corresponding to 8 secs.

## C. Cross-subject prediction accuracy

We present the average estimated cross-subject prediction accuracy across voxels for the *naturalistic listening fMRI* dataset in Fig. 6. We observe that the average estimated cross-subject prediction accuracy across voxels for the listening dataset is higher across subjects.

## D. Hyperparameter Details

**Implementation details for reproducibility.** All experiments were conducted on a machine with 2 NVIDIA A100 GPUs with 40GB GPU RAM. We used bootstrap ridge-regression with the following parameters: MSE loss function, and L2-decay ($\lambda$) varied from $10^1$ to $10^3$; best $\lambda$ was chosen by tuning on validation data that comprised a randomly chosen 10% subset from train set used only for hyperparameter tuning.

## E. Statistical significance

To determine if normalized predictivity scores are significantly higher than chance, we use block permutation tests. We employ the standard implementation of a block permutation test for fMRI data, which is to split the fMRI data into blocks of 10 contiguous TRs and permute the order of these blocks, while maintaining the original order of the TRs within each block. By permuting predictions 5000 times, we create an empirical distribution for chance performance, from which we estimate the p-value of the actual performance. To estimate the statistical significance of performance differences, such as between the model's predictions and chance or quantized model predictions and chance, we utilized the Wilcoxon signed-rank test, applying it to the mean normalized predictivity for the participants. In all cases, we denote significant differences with an asterisk *, indicating cases where p $\leq 0.05$.

## F. Normalized brain alignment for language ROIs

Figs. 9 show the average normalized brain alignment across the whole brain for both language model families, comparing SLMs, LLMs, and the grouped quantized variants.

Figs. 10 and 12 present the average normalized brain alignment across language-selective ROIs–including AG, ATL, PTL, IFGOrb, MFG, PCC, dmPFC, and AC–for both language model families, comparing SLMs, LLMs, and their quantized variants.

## G. Contrast of estimated model prediction accuracy for various subjects across the two model families

Figs. 14, 15, 16, 17, and 18 show voxel-wise percentage changes in brain alignment across model scales and quantization strategies for Qwen2.5 in the remaining participants. Corresponding results for LLaMA-3.2 are provided in Figs. 19, 20, 21, 22, 23, and 24.

We make the following observations across two families of language models: (i) Scaling down from LLMs to 3B SLMs: reductions in brain alignment are negligible in the bilateral temporal lobe and remain under 5% in the parietal cortex and IFGorb. Large cortical regions remain white, indicating that 3B SLMs preserve brain-relevant representations comparable to LLMs. The blue-marked voxels are sparse and localized, suggesting only limited information

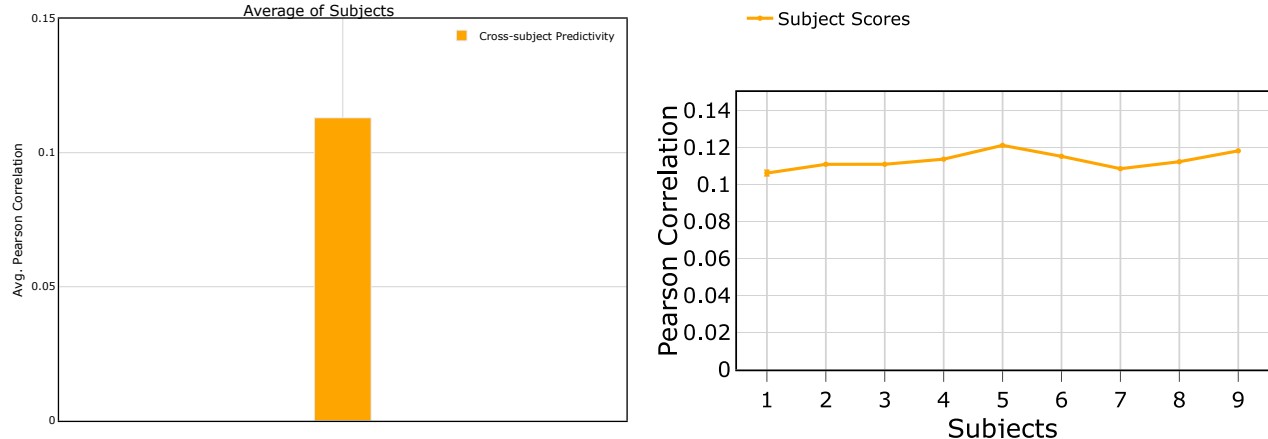

*Figure 6.* The estimated cross-subject prediction accuracy was computed across all participants for the Subset-Moth-Radio-Hour naturalistic listening fMRI dataset. The average cross-subject prediction accuracy is shown across predicted voxels where each voxel ceiling value is $\geq 0.05$.

*Table 6.* Detailed functional description of various brain regions.

| Early auditory | The early auditory region is the earliest cortical region for speech processing. This region is specialized for processing elementary speech sounds, as well as other temporally complex acoustical signals, such as music. |
|---|---|
| Late Language | Late language regions contribute to various linguistic processes. Regions 44 and 45 (Broca's region) are vital for speech production and grammar comprehension (Friederici, 2011). The IFJ, PG, and TPOJ clusters are involved in semantic processing, syntactic interpretation, and discourse comprehension (Deniz et al., 2019; Toneva et al., 2022a). STGa and STS play roles in phonological processing and auditory-linguistic integration (Vaidya et al., 2022; Millet et al., 2022; Gong et al., 2023). TA2 is implicated in auditory processing, especially in the context of language. |

loss. (ii) 3B SLMs to 3B SLMs GPTQ: applying GPTQ leads to widespread orange-marked voxels, reflecting consistent losses across distributed cortical regions. While some areas remain preserved (white voxels), the extent of information loss is greater than that observed with downscaling alone, confirming that GPTQ disproportionately disrupts brain-relevant alignment. (iii) 3B SLMs to 1.5B SLMs: relative to the 3B baseline, we observe extensive red-marked voxels, especially in temporal and language-related regions, indicating larger drops in alignment. This demonstrates the limits of scaling, as ultra-small models fail to capture brain-relevant representations. (iv) 3B SLMs → 3B SLMs AWQ: AWQ produces localized green-marked voxels, indicating regions of improved alignment relative to the 3B baseline, particularly in the IFG and AG. Most of the cortex remains unchanged (white), suggesting that AWQ maintains representational fidelity while offering modest regional gains.

## H. Impact of Linguistic Competence for Qwen2.5 and LLaMA-3.2 Models

From Table 7, we find that discourse competence emerges as the most influential factor. As models scale from smaller to larger versions, discourse probing scores (e.g., bridging, coreference) increase substantially, and these improvements align closely with gains in brain predictivity. By contrast, morphology tasks exert minimal impact: although accuracy improves modestly with scale, alignment scores remain largely unchanged, suggesting morphology is not a primary driver of brain alignment. Syntax and semantics show moderate effects. Both improve steadily with scale, and their contributions parallel alignment increases, though their influence is consistently weaker than discourse. Reasoning tasks (e.g., negation detection, correspondence) exhibit stable performance across model sizes and quantization settings, with alignment remaining relatively robust. This suggests reasoning is less sensitive to scaling but also less explanatory of alignment gains.

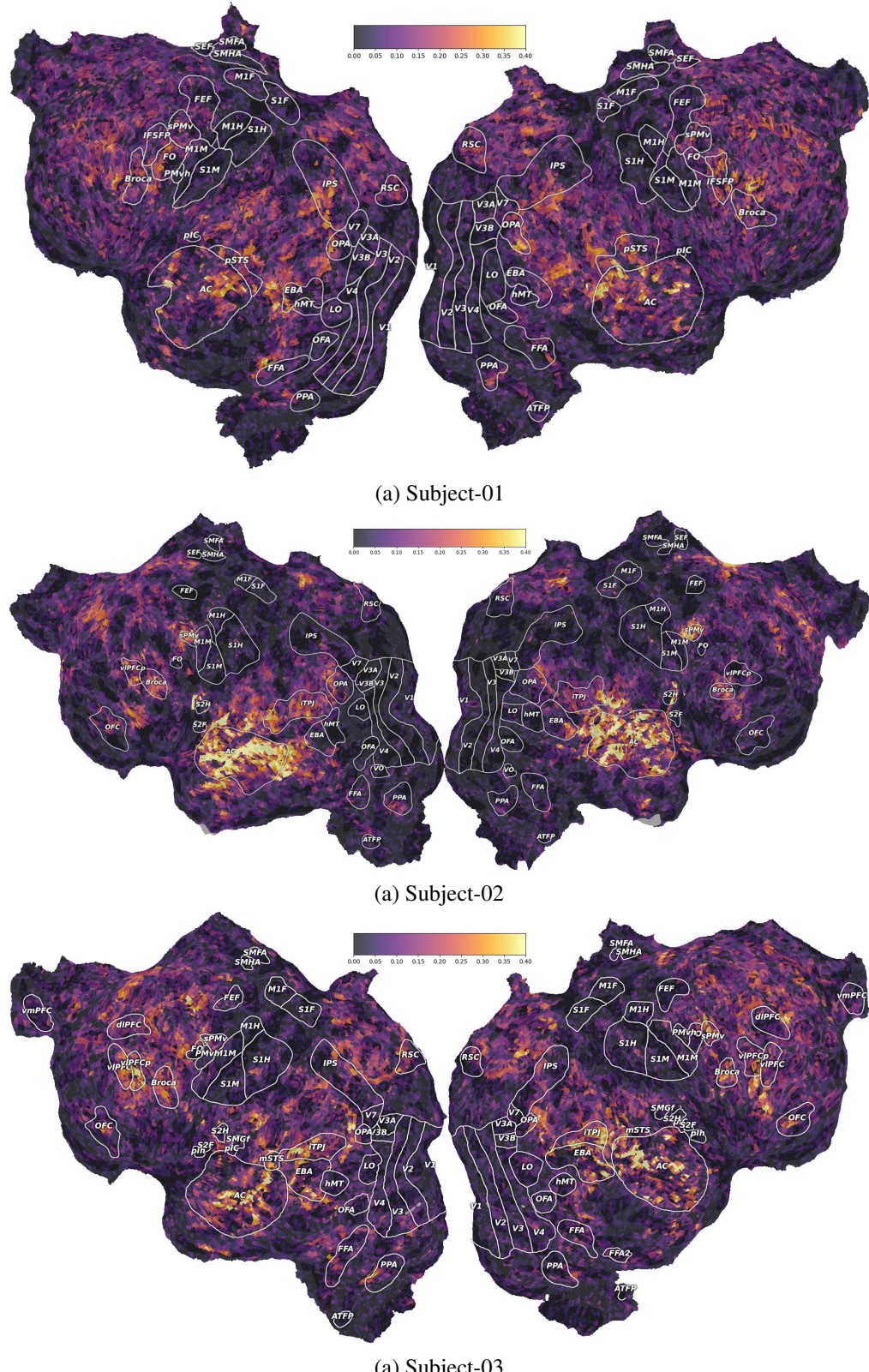

(a) Subject-01

(a) Subject-02

(a) Subject-03

*Figure 7.* Contrast of estimated cross-subject prediction accuracy for the participants for the listening condition. The color bar denotes Pearson Correlation.

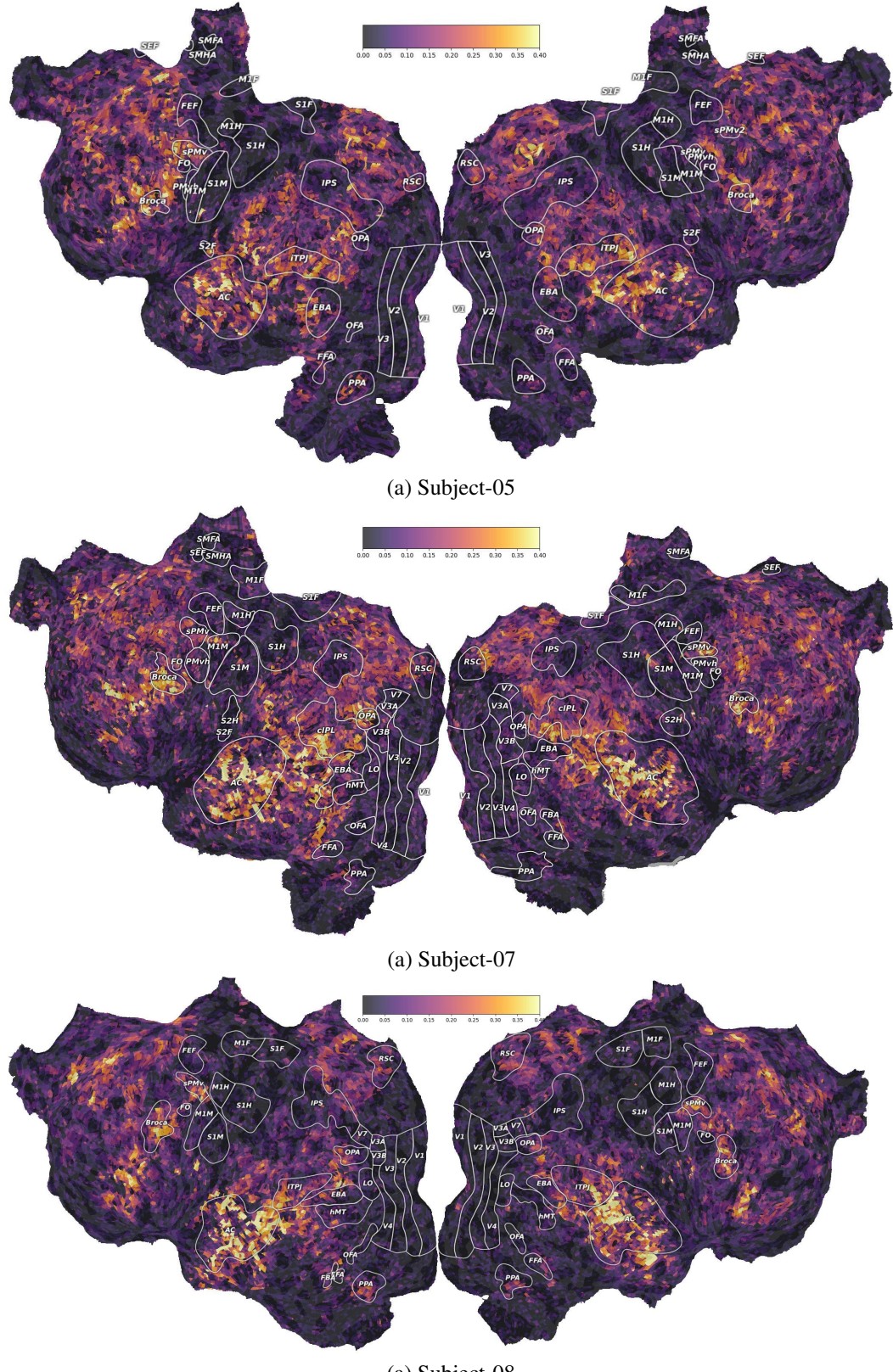

(a) Subject-05

(a) Subject-07

(a) Subject-08

*Figure 8.* Contrast of estimated cross-subject prediction accuracy for the participants for the listening condition. The color bar denotes Pearson Correlation.

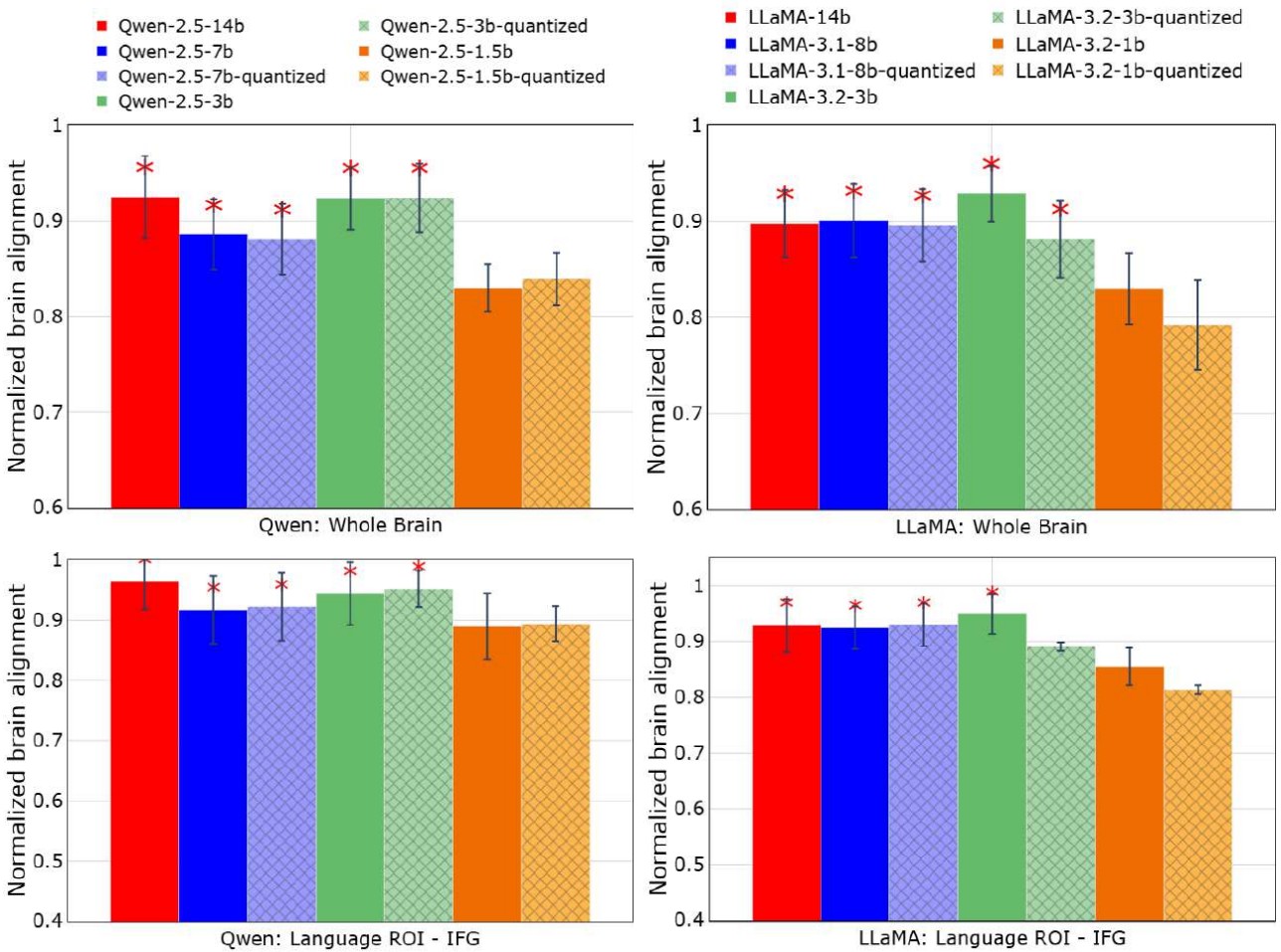

*Figure 9.* Qwen2.5 and LLaMA: Normalized brain alignment was computed by averaging across participants, layers, and voxels. Red: 14b, Blue: 7b, Green: 3b, Orange: 1.5b, Solid: full-precision SLMs/LLMs, Patterned: quantized models. * at a particular bar indicates that the model's prediction performance is significantly better than 1b/1.5b SLMs. The top row shows whole-brain normalized alignment, while the bottom row focuses on a language-selective ROI (IFG).

Taken together, these findings indicate that discourse-level representations are most critical for capturing brain-relevant information, followed by syntax and semantics, while morphology contributes little. Reasoning, though robust to compression, does not account for the major alignment differences between small and larger models.

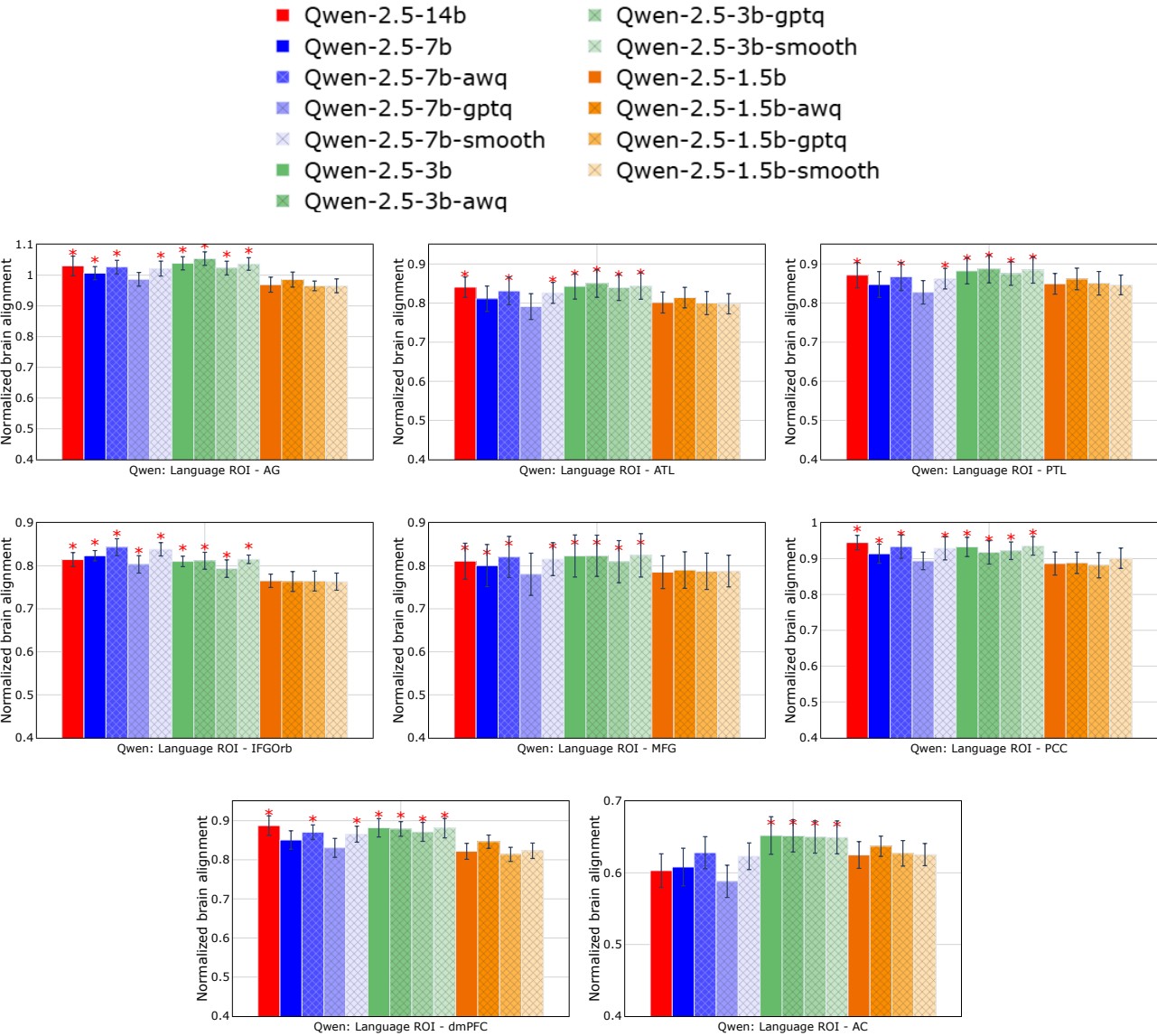

*Figure 10.* Normalized Predictivity of SLMs, LLMs, and Quantized Language Models for Qwen-2.5 models.

*Table 7.* Representative FlashHolmes task scores for Qwen-2.5 (1.5B, 3B, and 7B). Quantized 3B models remain close to full 3B across tasks, while 1.5B models show larger drops, especially in discourse and reasoning. The 7B model achieves the strongest scores overall.

| Category | Task | Qwen-2.5 1.5B | | | | Qwen-2.5 3B | | | | Qwen-2.5 7B |
| --- | --- | --- | --- | --- | --- | --- | --- | --- | --- | --- |
| | | FULL | AWQ | GPTQ | Smooth | FULL | AWQ | GPTQ | Smooth | FULL |
| Discourse | Bridging (edge) | 0.789 | **0.799** | **0.798** | **0.799** | **0.798** | 0.794 | 0.433 | 0.794 | 0.683 |
| | Bridging (sentence) | **0.800** | **0.800** | **0.800** | **0.800** | **0.800** | 0.799 | 0.410 | **0.799** | 0.667 |
| | Coreference | 0.411 | 0.399 | 0.405 | 0.399 | 0.352 | 0.433 | 0.410 | 0.357 | **0.794** |
| Morphology | Constituent (depth) | 0.802 | 0.801 | **0.809** | 0.801 | 0.752 | 0.752 | 0.801 | 0.752 | 0.755 |
| | Constituent (length) | 0.827 | 0.826 | **0.847** | 0.826 | 0.796 | 0.795 | 0.826 | 0.795 | 0.725 |
| Reasoning | Negation span classify | 0.742 | 0.741 | 0.746 | 0.743 | 0.812 | 0.810 | 0.808 | 0.811 | 0.950 |
| | Negation correspondence | 0.605 | 0.602 | 0.609 | 0.603 | 0.689 | 0.687 | 0.685 | 0.688 | 0.611 |
| | SemAntoNeg | 0.667 | 0.666 | 0.669 | 0.667 | 0.701 | 0.699 | 0.698 | 0.700 | 0.667 |
| Semantics | Object animacy | 0.994 | 0.994 | 0.991 | 0.994 | 0.988 | 0.988 | 0.994 | 0.988 | 0.805 |
| | Object gender | 0.546 | 0.535 | 0.532 | 0.535 | 0.482 | 0.402 | 0.531 | 0.402 | 0.807 |
| | Object number | 0.744 | 0.738 | 0.738 | 0.738 | 0.716 | 0.712 | 0.738 | 0.712 | 0.781 |
| Syntax | Adjunct island | 0.704 | 0.643 | 0.689 | 0.643 | 0.678 | 0.677 | 0.643 | 0.676 | 0.515 |
| | Anaphor gender agr. | 0.609 | 0.548 | 0.573 | 0.548 | 0.622 | 0.619 | 0.548 | 0.619 | 0.695 |
| | Anaphor number agr. | 0.631 | 0.611 | 0.582 | 0.613 | 0.647 | 0.661 | 0.613 | 0.662 | 0.860 |

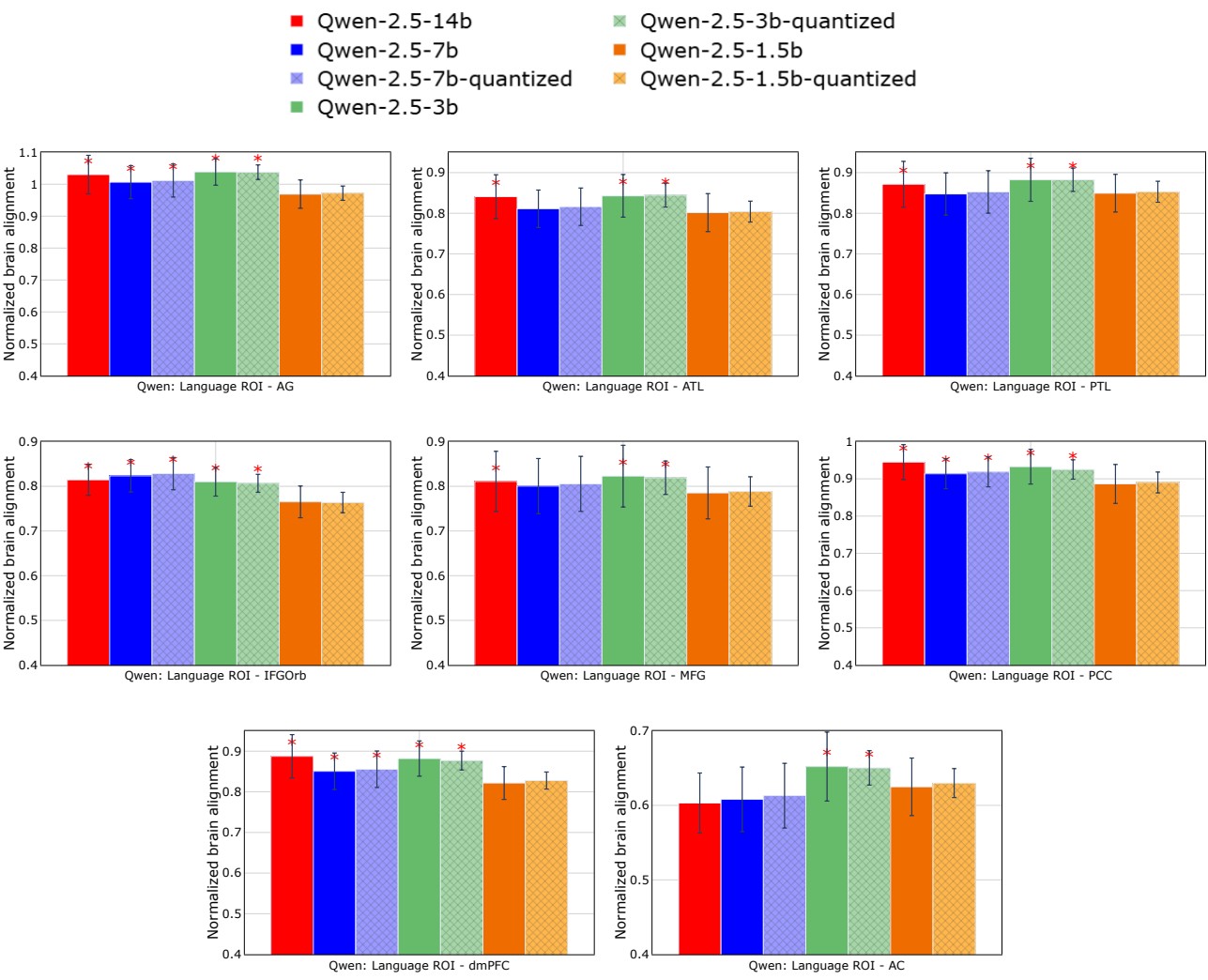

*Figure 11.* Normalized predictivity of Qwen2.5 SLMs and LLMs, including grouped comparisons of the base and quantized variants.

*Table 8.* Representative FlashHolmes task scores for LLaMA-3.2 models (1B, 3B, and 8B). Quantized 3B models remain close to full 3B, while the 8B model achieves the strongest scores overall.

| Category | Task | LLaMA-3.2 1B | | | | LLaMA-3.2 3B | | | | LLaMA-3 8B |
| --- | --- | --- | --- | --- | --- | --- | --- | --- | --- | --- |
| | | FULL | AWQ | GPTQ | Smooth | FULL | AWQ | GPTQ | Smooth | FULL |
| Discourse | Bridging (edge) | 0.792 | 0.801 | 0.800 | 0.799 | 0.801 | 0.799 | 0.433 | 0.798 | 0.703 |
| | Bridging (sentence) | 0.800 | 0.800 | 0.800 | 0.800 | 0.800 | 0.799 | 0.410 | 0.799 | 0.667 |
| | Coreference | 0.392 | 0.401 | 0.405 | 0.400 | 0.417 | 0.433 | 0.410 | 0.411 | 0.776 |
| Morphology | Constituent (depth) | 0.838 | 0.830 | 0.841 | 0.831 | 0.842 | 0.839 | 0.845 | 0.840 | 0.735 |
| | Constituent (length) | 0.877 | 0.875 | 0.890 | 0.876 | 0.897 | 0.896 | 0.895 | 0.894 | 0.705 |
| Reasoning | Negation span classify | 0.745 | 0.743 | 0.747 | 0.744 | 0.818 | 0.816 | 0.814 | 0.817 | 0.944 |
| | Negation correspondence | 0.612 | 0.610 | 0.614 | 0.611 | 0.701 | 0.699 | 0.697 | 0.700 | 0.611 |
| | SemAntoNeg | 0.672 | 0.670 | 0.673 | 0.671 | 0.709 | 0.707 | 0.706 | 0.708 | 0.667 |
| Semantics | Object animacy | 0.981 | 0.981 | 0.980 | 0.981 | 0.989 | 0.989 | 0.988 | 0.989 | 0.882 |
| | Object gender | 0.626 | 0.623 | 0.628 | 0.624 | 0.644 | 0.640 | 0.639 | 0.642 | 0.879 |
| | Object number | 0.713 | 0.710 | 0.714 | 0.711 | 0.720 | 0.718 | 0.717 | 0.719 | 0.876 |
| Syntax | Adjunct island | 0.668 | 0.667 | 0.669 | 0.667 | 0.744 | 0.742 | 0.741 | 0.743 | 0.700 |
| | Anaphor gender agr. | 0.678 | 0.675 | 0.680 | 0.676 | 0.739 | 0.738 | 0.736 | 0.737 | 0.650 |
| | Anaphor number agr. | 0.660 | 0.659 | 0.662 | 0.660 | 0.671 | 0.670 | 0.669 | 0.671 | 0.845 |

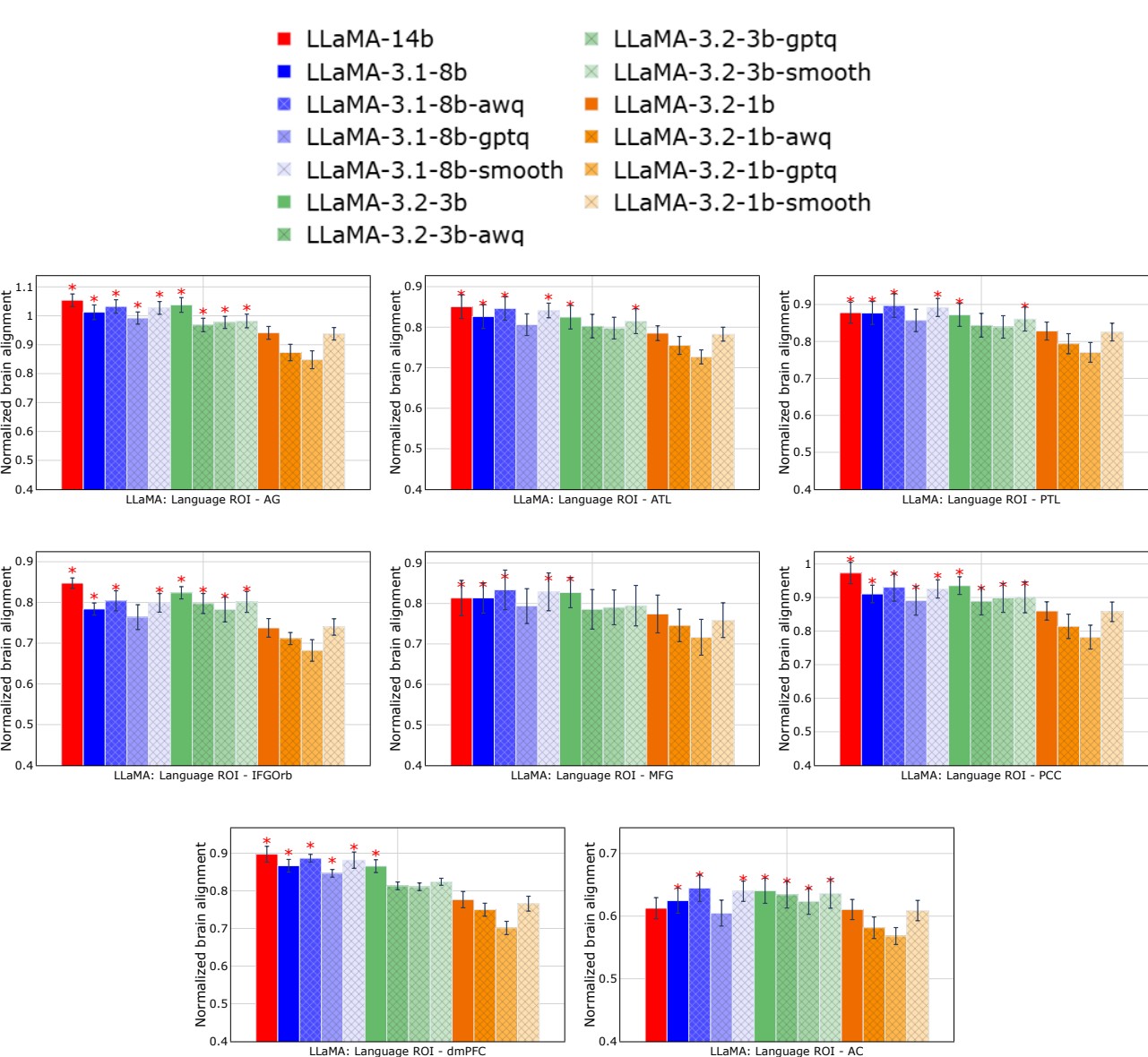

*Figure 12.* Normalized Predictivity of SLMs, LLMs, and Quantized Language Models for LLaMA-3.2 models.

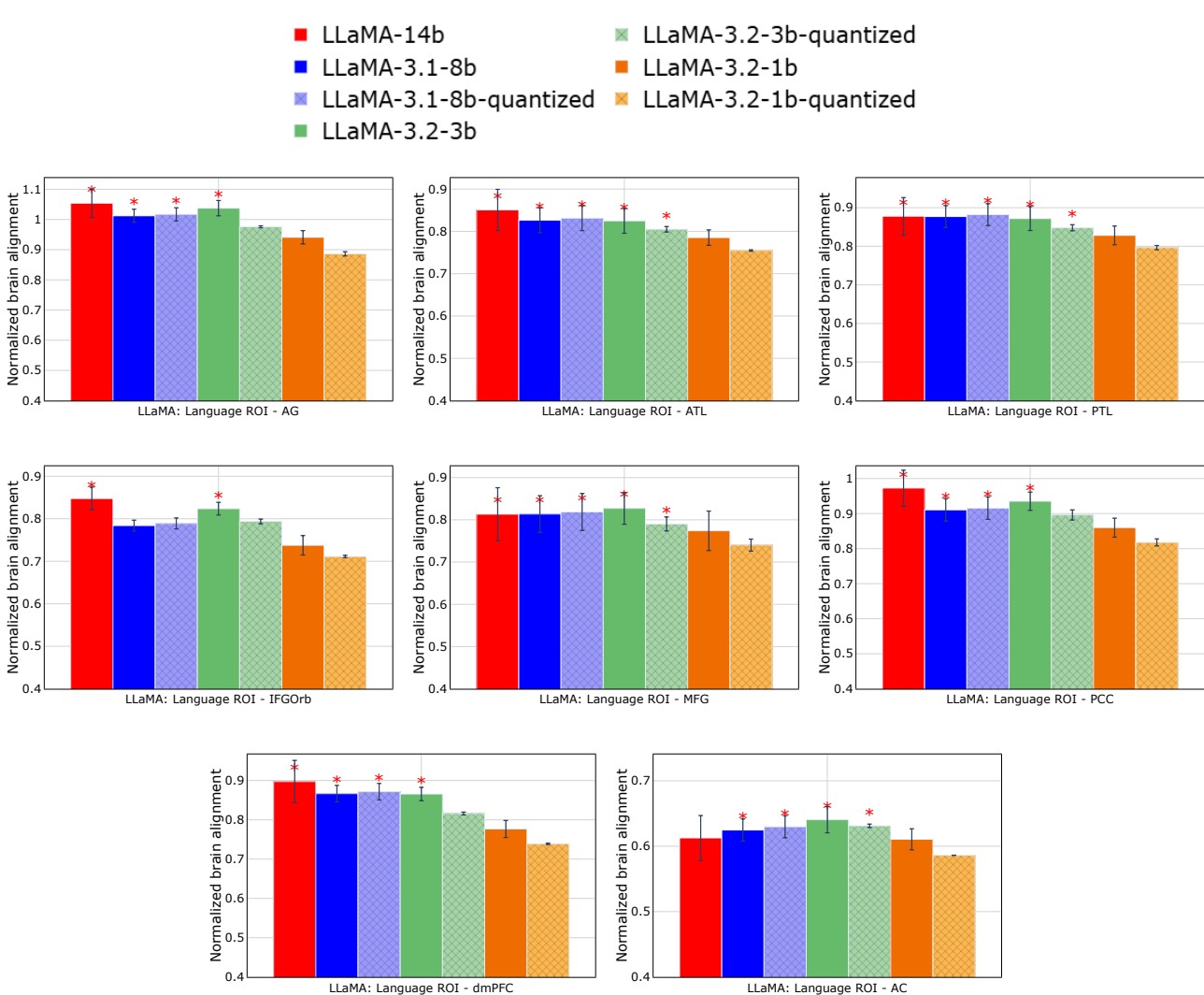

*Figure 13.* Normalized predictivity of LLaMA3.2 SLMs and LLMs, including grouped comparisons of the base and quantized variants.

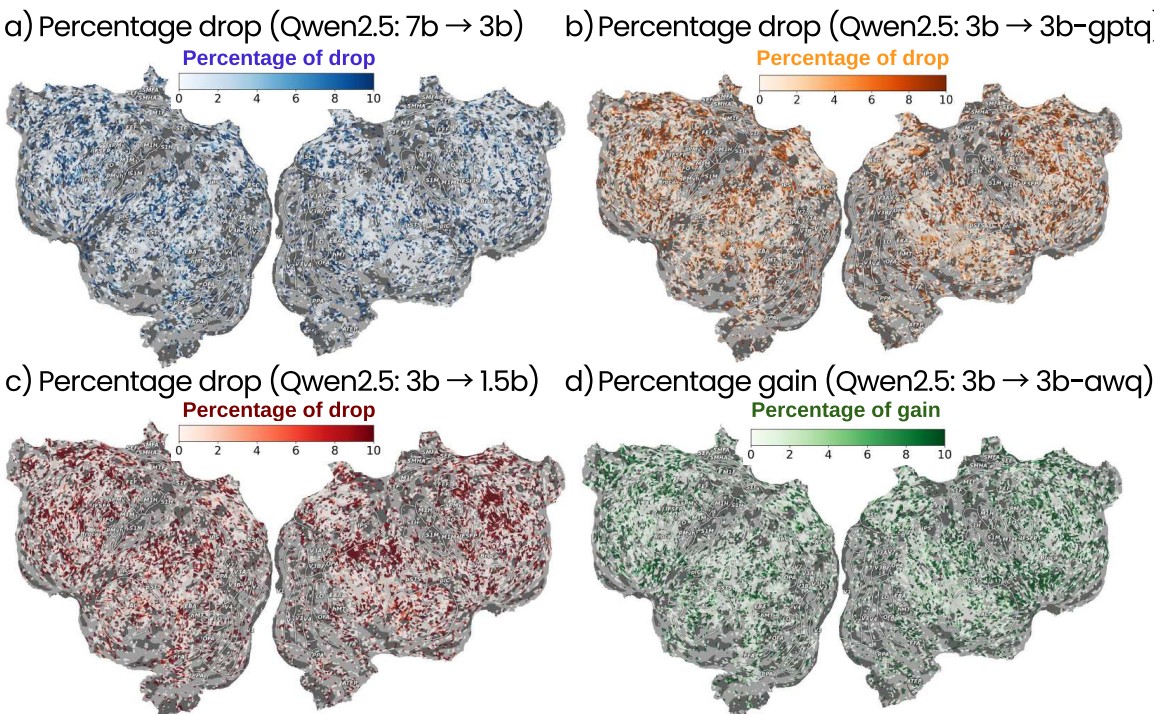

*Figure 14.* Qwen2.5: Percentage change in brain alignment across model scales and quantization methods, shown on the flattened cortical surface of a representative subject (subject-1). Blue, orange, and red voxels indicate regions of information loss ((a) LLMs → 3B SLMs, (b) 3B SLMs → 3B SLMs GPTQ, (c) 3B SLMs → 1.5B SLMs, respectively), (d) while green voxels highlight regions of improvement for 3B SLMs AWQ over 3B SLMs. White voxels denote regions with no change.

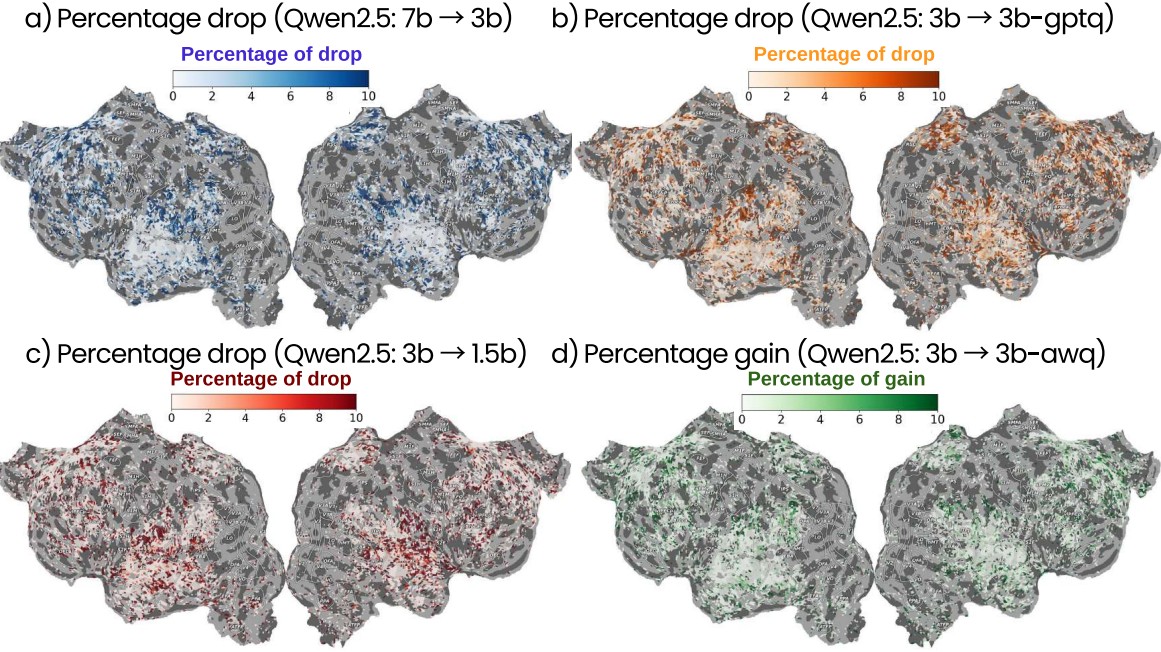

*Figure 15.* Qwen2.5: Percentage change in brain alignment across model scales and quantization methods, shown on the flattened cortical surface of a representative subject (subject-2). Blue, orange, and red voxels indicate regions of information loss ((a) LLMs → 3B SLMs, (b) 3B SLMs → 3B SLMs GPTQ, (c) 3B SLMs → 1.5B SLMs, respectively), (d) while green voxels highlight regions of improvement for 3B SLMs AWQ over 3B SLMs. White voxels denote regions with no change.

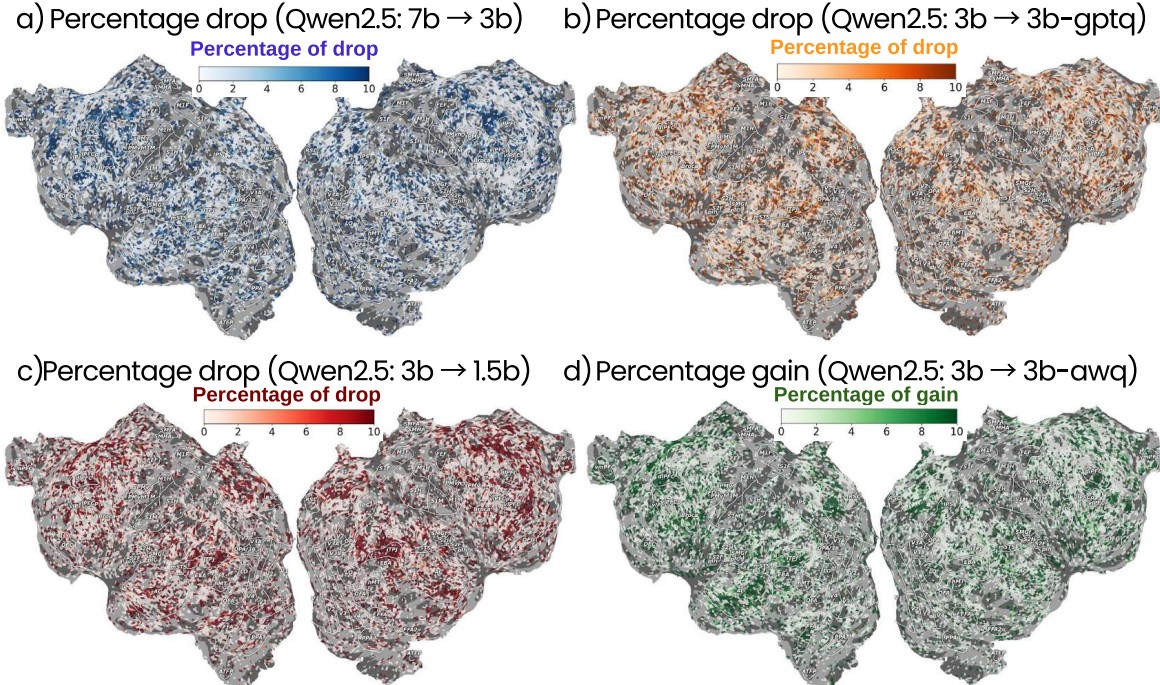

*Figure 16.* Qwen2.5: Percentage change in brain alignment across model scales and quantization methods, shown on the flattened cortical surface of a representative subject (subject-3). Blue, orange, and red voxels indicate regions of information loss ((a) LLMs → 3B SLMs, (b) 3B SLMs → 3B SLMs GPTQ, (c) 3B SLMs → 1.5B SLMs, respectively), (d) while green voxels highlight regions of improvement for 3B SLMs AWQ over 3B SLMs. White voxels denote regions with no change.

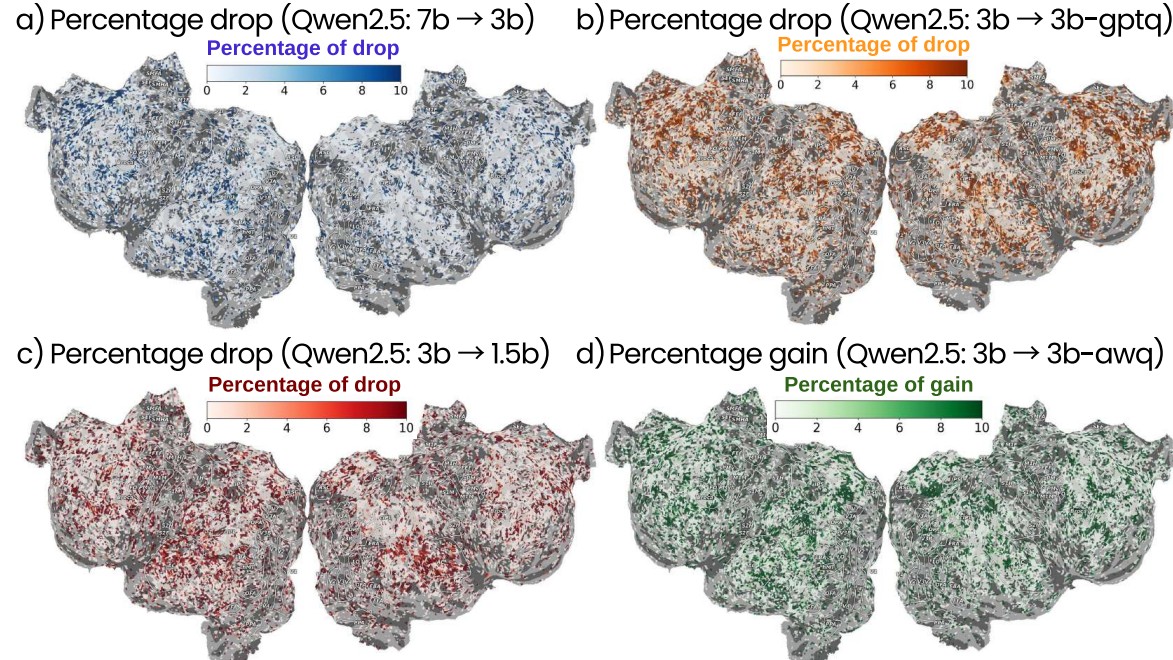

*Figure 17.* Qwen2.5: Percentage change in brain alignment across model scales and quantization methods, shown on the flattened cortical surface of a representative subject (subject-7). Blue, orange, and red voxels indicate regions of information loss ((a) LLMs → 3B SLMs, (b) 3B SLMs → 3B SLMs GPTQ, (c) 3B SLMs → 1.5B SLMs, respectively), (d) while green voxels highlight regions of improvement for 3B SLMs AWQ over 3B SLMs. White voxels denote regions with no change.

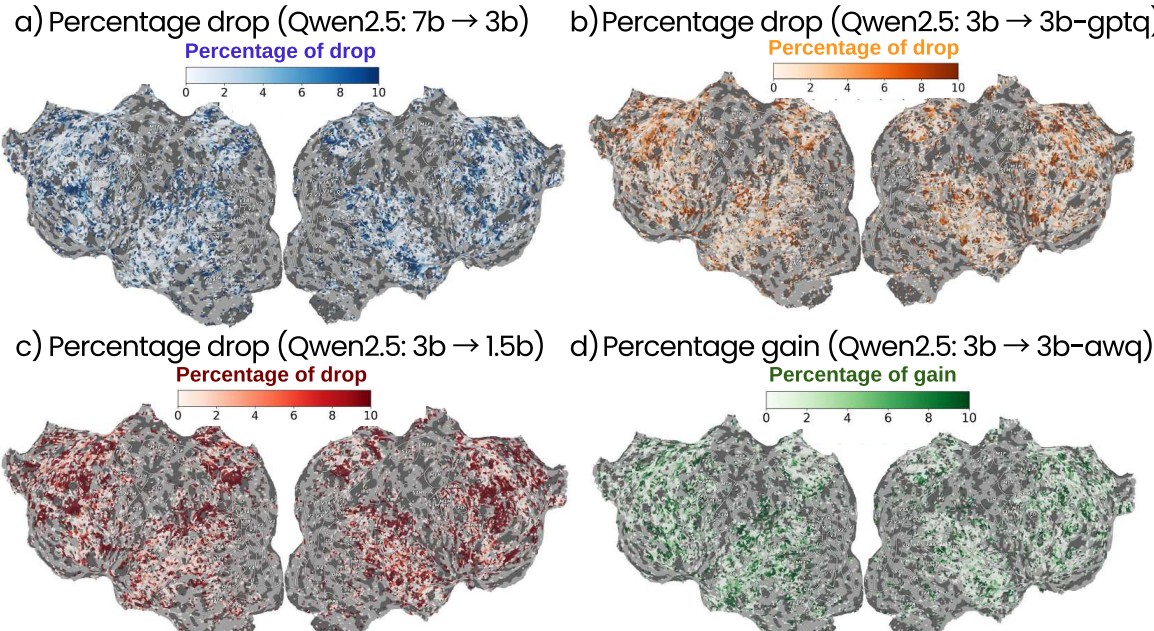

*Figure 18.* Qwen2.5: Percentage change in brain alignment across model scales and quantization methods, shown on the flattened cortical surface of a representative subject (subject-8). Blue, orange, and red voxels indicate regions of information loss ((a) LLMs → 3B SLMs, (b) 3B SLMs → 3B SLMs GPTQ, (c) 3B SLMs → 1.5B SLMs, respectively), (d) while green voxels highlight regions of improvement for 3B SLMs AWQ over 3B SLMs. White voxels denote regions with no change.

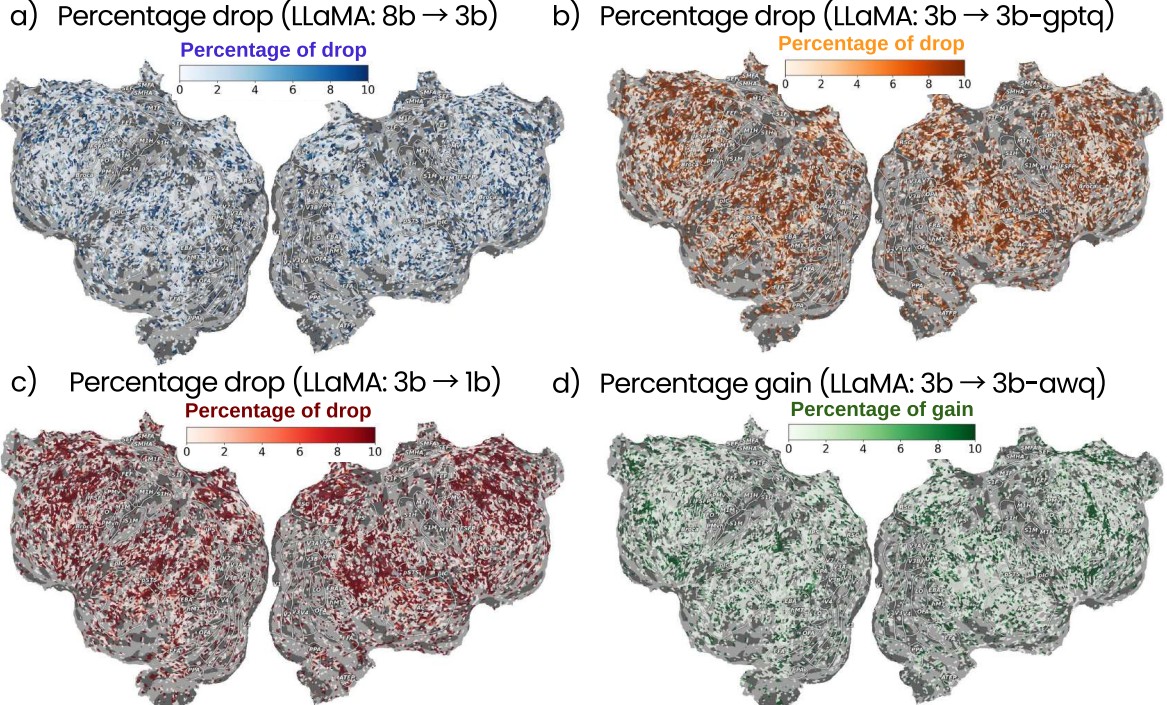

*Figure 19.* LLaMA-3.2: Percentage change in brain alignment across model scales and quantization methods, shown on the flattened cortical surface of a representative subject (subject-1). Blue, orange, and red voxels indicate regions of information loss ((a) LLMs → 3B SLMs, (b) 3B SLMs → 3B SLMs GPTQ, (c) 3B SLMs → 1.5B SLMs, respectively), (d) while green voxels highlight regions of improvement for 3B SLMs AWQ over 3B SLMs. White voxels denote regions with no change.

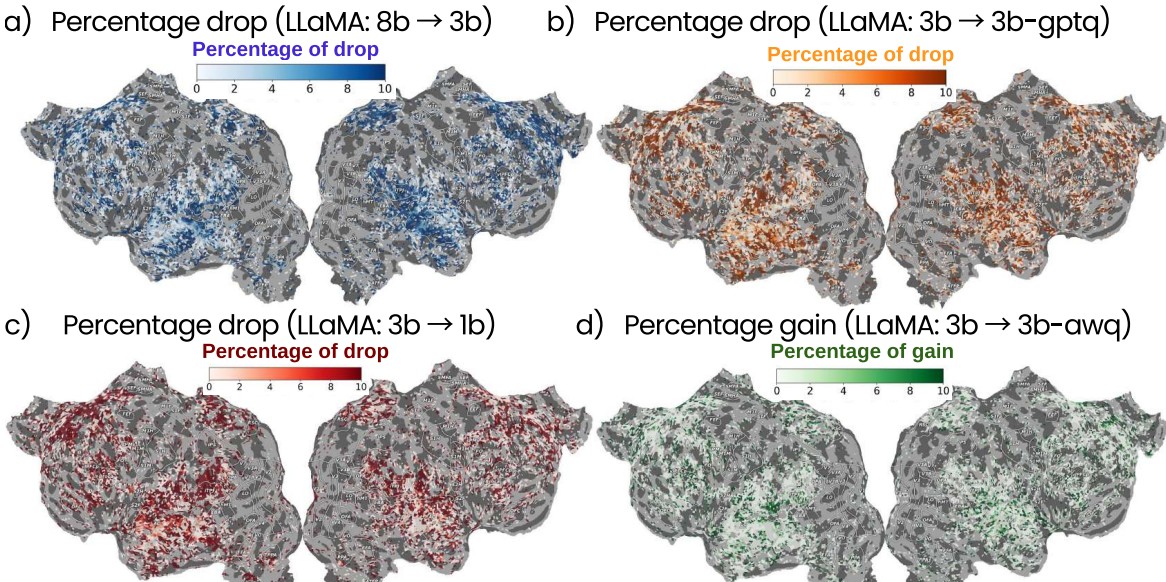

*Figure 20.* LLaMA-3.2: Percentage change in brain alignment across model scales and quantization methods, shown on the flattened cortical surface of a representative subject (subject-2). Blue, orange, and red voxels indicate regions of information loss ((a) LLMs → 3B SLMs, (b) 3B SLMs → 3B SLMs GPTQ, (c) 3B SLMs → 1.5B SLMs, respectively), (d) while green voxels highlight regions of improvement for 3B SLMs AWQ over 3B SLMs. White voxels denote regions with no change.

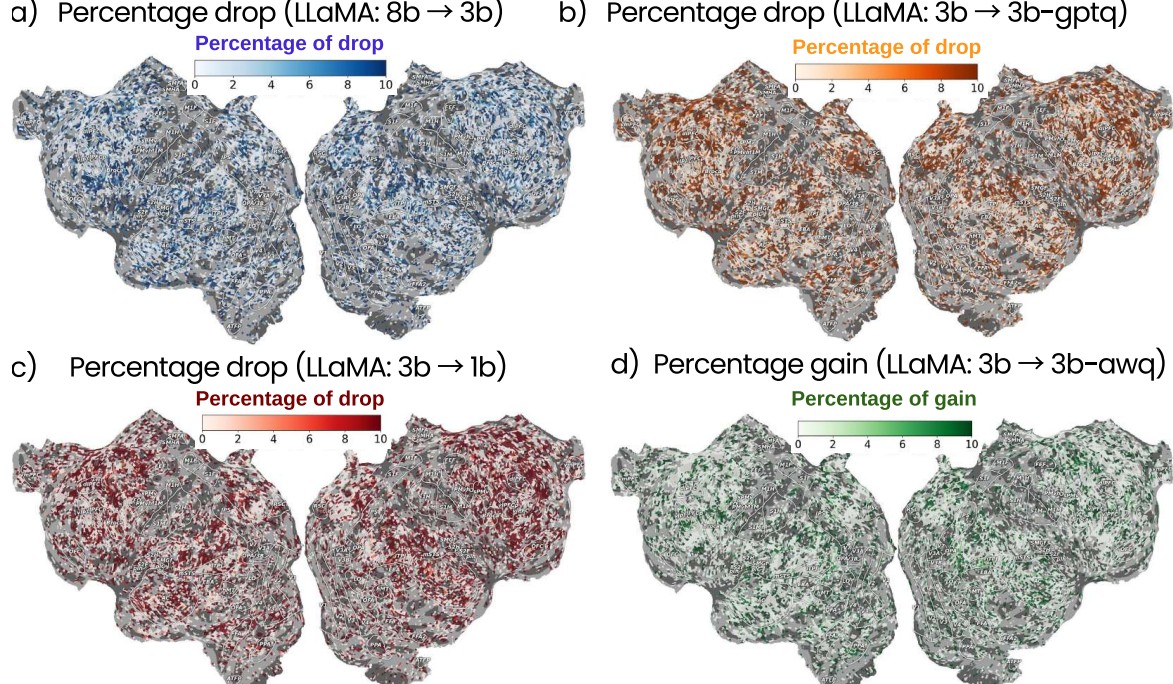

*Figure 21.* LLaMA-3.2: Percentage change in brain alignment across model scales and quantization methods, shown on the flattened cortical surface of a representative subject (subject-3). Blue, orange, and red voxels indicate regions of information loss ((a) LLMs → 3B SLMs, (b) 3B SLMs → 3B SLMs GPTQ, (c) 3B SLMs → 1.5B SLMs, respectively), (d) while green voxels highlight regions of improvement for 3B SLMs AWQ over 3B SLMs. White voxels denote regions with no change.

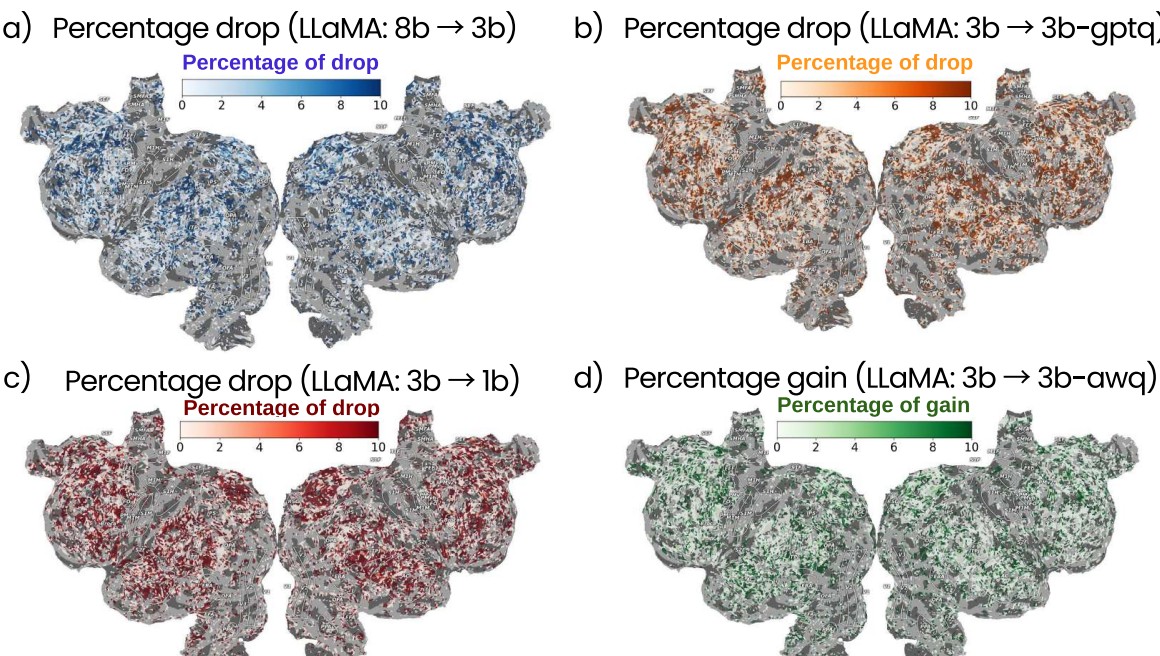

*Figure 22.* LLaMA-3.2: Percentage change in brain alignment across model scales and quantization methods, shown on the flattened cortical surface of a representative subject (subject-7). Blue, orange, and red voxels indicate regions of information loss ((a) LLMs → 3B SLMs, (b) 3B SLMs → 3B SLMs GPTQ, (c) 3B SLMs → 1.5B SLMs, respectively), (d) while green voxels highlight regions of improvement for 3B SLMs AWQ over 3B SLMs. White voxels denote regions with no change.

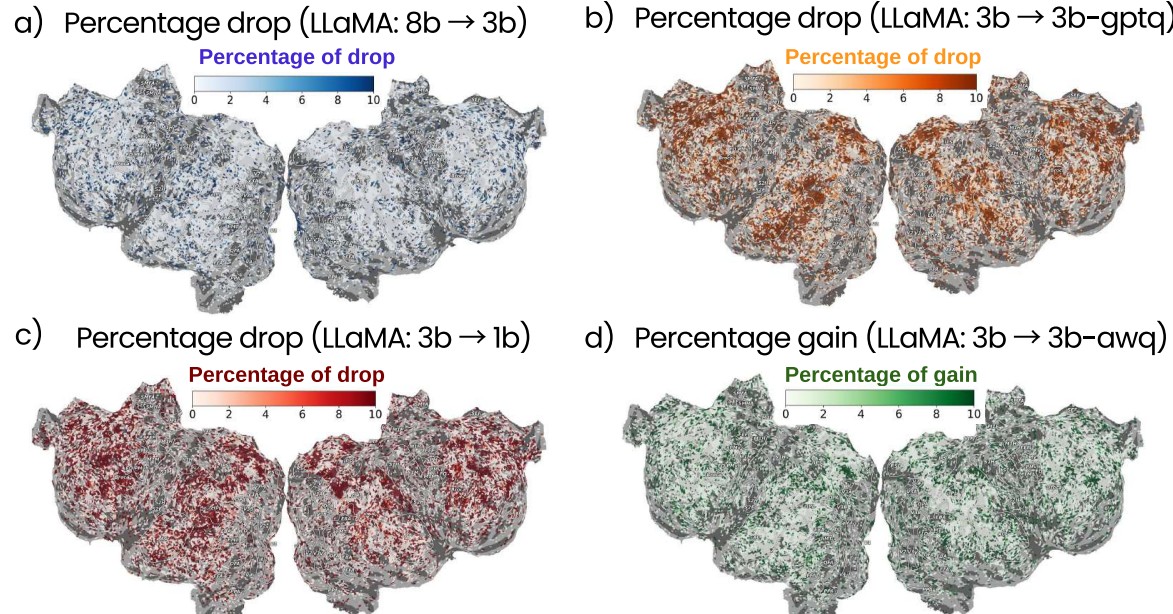

*Figure 23.* LLaMA-3.2: Percentage change in brain alignment across model scales and quantization methods, shown on the flattened cortical surface of a representative subject (subject-8). Blue, orange, and red voxels indicate regions of information loss ((a) LLMs → 3B SLMs, (b) 3B SLMs → 3B SLMs GPTQ, (c) 3B SLMs → 1.5B SLMs, respectively), (d) while green voxels highlight regions of improvement for 3B SLMs AWQ over 3B SLMs. White voxels denote regions with no change.

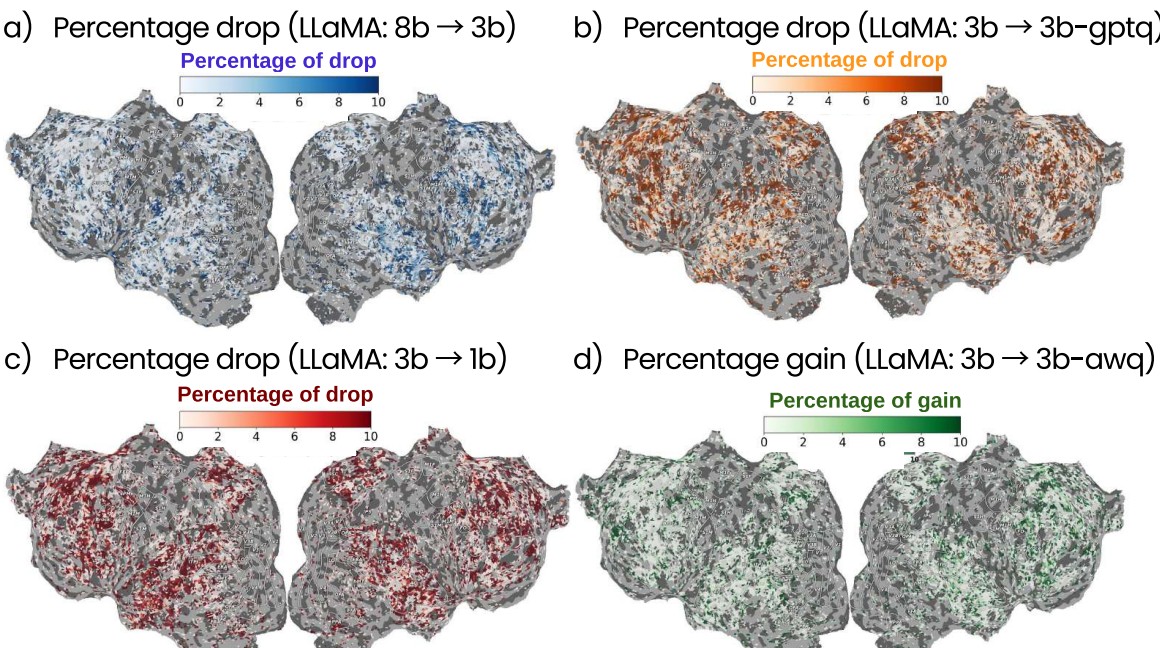

*Figure 24.* LLaMA-3.2: Percentage change in brain alignment across model scales and quantization methods, shown on the flattened cortical surface of a representative subject (subject-8). Blue, orange, and red voxels indicate regions of information loss ((a) LLMs → 3B SLMs, (b) 3B SLMs → 3B SLMs GPTQ, (c) 3B SLMs → 1.5B SLMs, respectively), (d) while green voxels highlight regions of improvement for 3B SLMs AWQ over 3B SLMs. White voxels denote regions with no change.

# I. Quantitative Analysis across model families

We quantify scaling differences by performing statistical significance across subjects for the best selective layer per model: Qwen2.5 in Table 2, LLaMA in Table 9 and DeepSeek 10. For Qwen2.5 model, the resulting mean best-layer scores are: 1.5B: 0.85+0.09, 3B: 0.92+0.08, 7B: 0.895+0.09, 14B: 0.93+0.10. Paired tests over subjects (n = 9) show that 3B and 14B are statistically indistinguishable (mean difference -0.0004, t(8) = -0.03, p ≈ 0.98), while both 3B and 14B significantly outperform the 1.5B model (3B vs. 1.5B: $\Delta$ = 0.07, t(8) = 4.89, p ≈ 0.004; 14B vs. 1.5B: $\Delta$ = 0.07, t(8) = 3.16, p ≈ 0.025). We also find a modest but significant advantage of 3B and 14B over 7B in best-layer alignment ($\Delta$ ≈ 0.04, p ≈ 0.02-0.04). These tests support our main claim in this regime: beyond 3B, increasing model size up to 14B yields at most modest gains in brain alignment, whereas 1B-1.5B models are reliably worse.

For the LLaMA model, as shown in Table 9, we find that (i) 14B ≈ 7B: No difference ($\Delta$ = -0.00, p ≈ 0.95) - statistically identical (ii) 3B ≈ 14B/7B: Slight advantage but not significant (p > 0.05) and (iii) All vs 1B: Highly significant differences (p < 0.001 for 3B and 7B). Overall, the 3B, 7B, and 14B models form a statistically equivalent top tier, all significantly outperforming the 1B model.

Analysis of DeepSeek models (14B, 7B, 3B, 1B parameters), as shown in Table 10 reveals a clear scaling hierarchy with the 14B and 3B models forming a statistically equivalent top tier. We make the following observations from Table 10: (i) 14B ≈ 3B: No significant difference (p ≈ 0.61), indicating 3B achieves 14B-level performance with 80% fewer parameters! (ii) 14B, 3B >> 7B: Highly significant advantages (p < 0.01), (iii) All >> 1B: Very large differences (all p < 0.001). Overall, across three independent model families (Qwen, LLaMA, DeepSeek) and using best-layer scores with paired tests over nine subjects, we find a consistent pattern: 1B-1.5B models are reliably worse in brain alignment, while 3B models already reach the same level as their 7B-14B counterparts. In Qwen and DeepSeek, 3B and 14B are statistically indistinguishable, whereas both significantly outperform the smallest models; in LLaMA, 3B and 14B again lie in a narrow, non-significantly different range, with 7B closely tracking 14B and clearly above 1B. These results do not overturn global scaling laws, but they do indicate a local plateau in the compressed 1-14B regime: once model capacity reaches ~3B, further scaling yields at most modest gains in brain predictivity, while going below this threshold leads to a robust drop in alignment.

*Table 9.* Pairwise differences in LLaMA best-layer scores across models (paired *t*-tests over 9 subjects). Δ is mean(A–B).

| Comparison | Δ (A–B) | $t(8)$ | Sig. (2-sided) |
|---|---|---|---|
| 3B – 14B | 0.03 | 2.50 | n.s. ($p \approx 0.05$) |
| 3B – 7B | 0.03 | 1.97 | n.s. ($p \approx 0.11$) |
| 3B – 1B | 0.10 | 6.89 | $p < 0.001$ |
| 14B – 7B | -0.00 | -0.07 | n.s. ($p \approx 0.95$) |
| 14B – 1B | 0.07 | 3.12 | $p < 0.05$ |
| 7B – 1B | 0.07 | 11.43 | $p < 0.001$ |

*Table 10.* Pairwise differences in DeepSeek best-layer scores (paired *t*-tests over 9 subjects). Δ is mean(A–B).

| Comparison | Δ (A–B) | $t(8)$ | Sig. (2-sided) |
|---|---|---|---|
| 14B – 3B | -0.01 | -0.54 | n.s. ($p \approx 0.61$) |
| 14B – 7B | 0.07 | 4.84 | $p < 0.01$ |
| 14B – 1B | 0.19 | 11.18 | $p < 0.001$ |
| 3B – 7B | 0.08 | 4.18 | $p < 0.01$ |
| 3B – 1B | 0.19 | 10.85 | $p < 0.001$ |
| 7B – 1B | 0.11 | 9.28 | $p < 0.001$ |

## J. Statistical Validation of Quantization Effects

**Quantization Effects - Qwen2.5.** We now provide formal statistical tests and variability measures for the quantization comparisons. For each Qwen2.5 model (1.5B, 3B, 7B), and for each quantization method (Full (FP16), AWQ, GPTQ, SmoothQuant), we compute best-layer brain alignment per subject and run paired *t*-tests across subjects between methods (Table 11). Negative Δ in rows of the form "FP16–AWQ" indicates that AWQ outperforms FP16. For Qwen2.5–7B (Table 11, left), AWQ and SmoothQuant are significantly better than both FP16 and GPTQ (e.g., FP16–AWQ: $\Delta = -0.020$, $t(8) = -6.10$, $p < 0.001$; AWQ–GPTQ: $\Delta = 0.040$, $t(8) = 7.10$, $p < 0.001$), while GPTQ is significantly worse than FP16. For Qwen2.5–3B (Table 11, right), none of the quantized variants differ significantly from FP16, but AWQ and SmoothQuant significantly outperform GPTQ, suggesting that well-designed quantization preserves alignment whereas GPTQ exhibits a modest degradation. For Qwen2.5–1.5B (Table 11, bottom), AWQ is significantly better than FP16 ($\Delta = -0.024$, $t(8) = -4.04$, $p < 0.01$), whereas GPTQ and SmoothQuant are statistically indistinguishable from FP16, and differences among the three quantized variants do not reach significance after correction.

We also summarize quantization performance at the level of mean ± standard deviation across subjects in Table 12. Across all Qwen sizes, AWQ and SmoothQuant closely or slightly exceed full models (FP16) in mean best-layer alignment (differences on the order of 0.01–0.02, within one standard deviation), whereas GPTQ tends to be lower than FP16, especially for 7B and 3B. Together, Table 8 and Table 9 show that (i) some apparent improvements in the figures are within noise and are now explicitly reported as non-significant, and (ii) the main qualitative pattern is statistically supported: well-designed quantization (AWQ/SmoothQuant) preserves brain alignment at near-full-precision levels, while GPTQ

produce a modest but reliable degradation.

**Quantization effects in LLaMA-3.** We performed the same best-layer, paired *t*-test analysis for LLaMA-3 models (1B, 3B, 8B). For LLaMA-3-8B, all pairwise differences between FP16, AWQ, GPTQ, and SmoothQuant are highly significant (Table 13, left). Negative Δ in rows of the form "FP16–AWQ" indicates AWQ > FP16; specifically, FP16–AWQ is negative ($\Delta = -0.010$, $p < 0.001$), while FP16–GPTQ is positive ($\Delta = 0.020$, $p < 0.001$) and AWQ-GPTQ is strongly positive ($\Delta = 0.030$, $p < 0.001$). This implies the ordering AWQ > FP16 *ge* SmoothQuant > GPTQ for 8B. For LLaMA-3-3B (Table 13, right), GPTQ is significantly worse than FP16 ($\Delta = 0.059$, $p < 0.05$), while AWQ and SmoothQuant are not significantly different from FP16. SmoothQuant significantly outperforms both AWQ and GPTQ (AWQ–SmoothQuant: $\Delta = -0.011$, $p < 0.05$; GPTQ–SmoothQuant: $\Delta = -0.023$, $p < 0.01$), indicating that GPTQ is the main outlier, with AWQ/SmoothQuant and FP16 forming a higher-performing cluster. For LLaMA-3-1B (Table 13, bottom), GPTQ again shows a clear degradation: FP16–GPTQ is positive and significant ($\Delta = 0.076$, $p < 0.01$), and GPTQ–SmoothQuant is strongly negative ($\Delta = -0.074$, $p < 0.01$). In contrast, AWQ and SmoothQuant are statistically indistinguishable from FP16 (all $p > 0.1$), and differences among the three quantized variants other than GPTQ do not reach significance.

Table 14 summarizes quantization performance for LLaMA-3 models (8B, 3B, 1B). For LLaMA-3-8B, AWQ achieves the highest mean alignment (0.911), followed by FP16 and SmoothQuant, with GPTQ lowest (0.881). For LLaMA-3-3B, FP16 has the highest mean (0.929), while all three quantized variants are somewhat lower, with SmoothQuant > AWQ > GPTQ. For LLaMA-3-1B, FP16 and SmoothQuant are nearly identical and clearly above AWQ and GPTQ, with GPTQ again lowest. Overall, the LLaMA results align with our Qwen analyses: GPTQ consistently yields lower brain alignment than FP16, whereas AWQ and SmoothQuant generally preserve full-precision performance, sometimes even slightly improving upon it. This validates our conclusion that carefully designed quantization (AWQ/SmoothQuant) can maintain brain alignment at near full-precision levels, while some schemes (GPTQ) introduce a modest but reliable degradation.

## K. ROI-Specific Analysis, Best Layer Selection and Subject Variability.

In our analyses, we extract activations from every transformer layer, and fit a separate voxel-wise encoding model for each layer. For each model, we then compute brain alignment layer-by-layer across the language ROIs and identify the best layer as the one with the highest mean normalized predictivity. The main size/quantization comparisons are

*Table 11.* Pairwise comparisons of brain-alignment differences across quantization methods for Qwen2.5 models. Each Table reports mean differences ($\Delta$), $t$-statistics, and two-sided significance tests for 7B (left), 3B (right), and 1.5B (bottom).

| Comparison (A–B) | $\Delta$ | $t(8)$ | Sig. |
|---|---|---|---|
| Qwen2.5-7B–AWQ | -0.020 | -6.10 | $p < 0.001$ |
| Qwen2.5-7B–GPTQ | 0.020 | 6.20 | $p < 0.001$ |
| Qwen2.5-7B–SmoothQuant | -0.005 | -3.50 | $p < 0.016$ |
| AWQ–GPTQ | 0.040 | 7.10 | $p < 0.001$ |
| AWQ–SmoothQuant | 0.015 | 4.20 | $p < 0.008$ |
| GPTQ–SmoothQuant | -0.025 | -4.90 | $p < 0.004$ |

| Comparison (A–B) | $\Delta$ | $t(8)$ | Sig. |
|---|---|---|---|
| Qwen2.5-3B–AWQ | -0.010 | -1.23 | n.s. ($p \approx 0.28$) |
| Qwen2.5-3B–GPTQ | 0.014 | 2.24 | n.s. ($p \approx 0.08$) |
| Qwen2.5-3B–SmoothQuant | -0.007 | -1.55 | n.s. ($p \approx 0.18$) |
| AWQ–GPTQ | 0.024 | 3.10 | $p < 0.05$ |
| AWQ–SmoothQuant | 0.003 | 0.34 | n.s. ($p \approx 0.75$) |
| GPTQ–SmoothQuant | -0.020 | -3.12 | $p < 0.05$ |

| Comparison (A–B) | $\Delta$ | $t(8)$ | Sig. |
|---|---|---|---|
| Qwen2.5-1.5B–AWQ | -0.024 | -4.04 | $p < 0.01$ |
| Qwen2.5-1.5B–GPTQ | 0.002 | 0.18 | n.s. ($p \approx 0.86$) |
| Qwen2.5-1.5B–SmoothQuant | -0.004 | -0.71 | n.s. ($p \approx 0.51$) |
| AWQ–GPTQ | 0.026 | 2.03 | n.s. ($p \approx 0.10$) |
| AWQ–SmoothQuant | 0.020 | 2.43 | n.s. ($p \approx 0.06$) |
| GPTQ–SmoothQuant | -0.006 | -0.42 | n.s. ($p \approx 0.69$) |

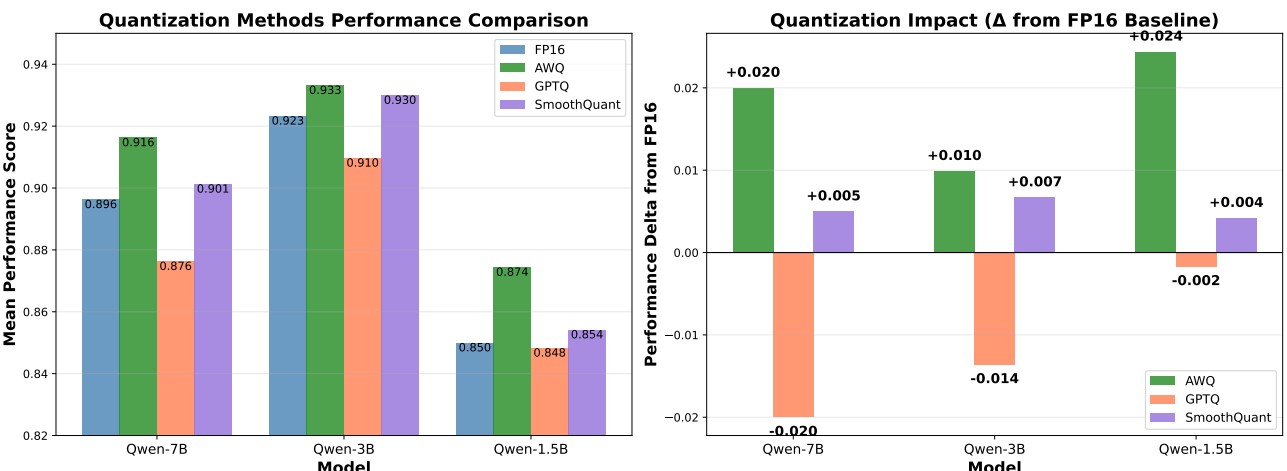

*Figure 25.* Qwen2.5 Quantization Analysis: (left) Quantization methods comparison, (right) Quantization impact

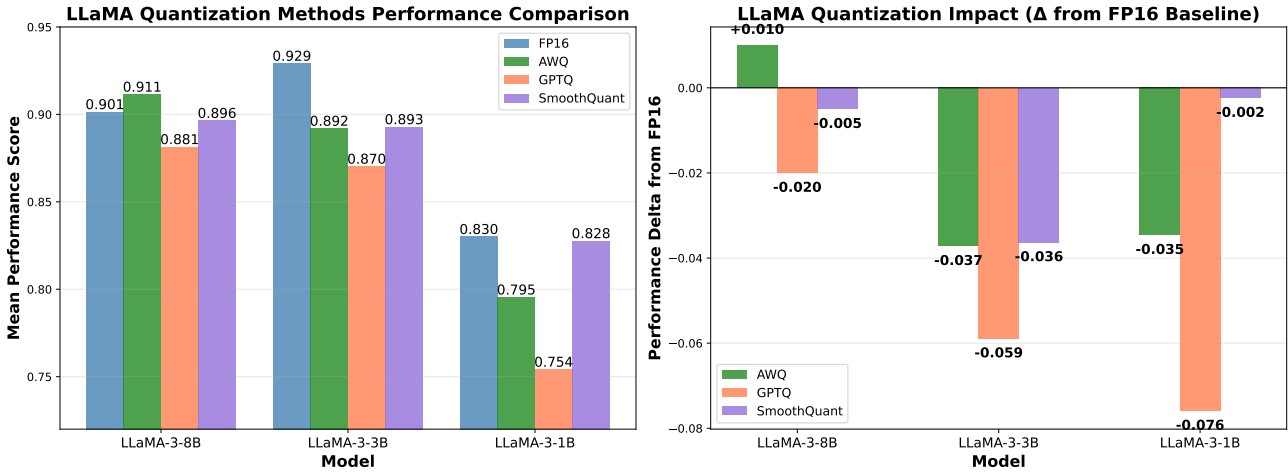

*Figure 26.* LLaMA-3 Quantization Analysis: (left) Quantization methods comparison, (right) Quantization impact

reported using this model-specific best layer (see Tables 16 and 19).

From Table 16, across language ROIs we find that the best layers are highly consistent within a given model: the same (or adjacent) layer tends to be optimal across ROIs, so we

treat the best layer as a model-level property when summarizing results. Overall, across models we observe the familiar pattern that middle-to-late layers yield the strongest brain alignment, with early layers performing clearly worse.

Regarding quantization, we also examined whether AWQ,

*Table 12.* Quantization method performance across Qwen models (mean ± std over 9 subjects).

| Model | Full precision (FP16) | AWQ | GPTQ | SmoothQuant |
|---|---|---|---|---|
| Qwen-7B | 0.886 ± 0.092 | 0.906 ± 0.092 | 0.866 ± 0.092 | 0.891 ± 0.092 |
| Qwen-3B | 0.923 ± 0.080 | 0.933 ± 0.085 | 0.910 ± 0.091 | 0.930 ± 0.085 |
| Qwen-1.5B | 0.850 ± 0.087 | 0.874 ± 0.099 | 0.848 ± 0.088 | 0.854 ± 0.084 |

GPTQ, or SmoothQuant systematically shift the optimal layer. We did not observe any systematic change: for a given architecture, the best layer under quantization is typically the same as in FP16 or within ±1–2 layers in the same middle/late portion of the network. In other words, quantization affects the magnitude of brain alignment (as analyzed in Tables 12 and 14), but not the qualitative position of the brain-optimal layers. We now clarify this procedure and these observations in the main text and refer explicitly to the layer-wise summaries in Tables 16 and 19.

The best layers are consistent in each model across ROIs; so, we consider best layers specific to each model while reporting the results. Overall, across models, middle to late layers show better brain alignment. Mostly 3b model is the best in terms of brain alignment across ROIs.

## L. Encoding performance of DeepSeek Models

We have now extended our evaluation to an additional model family: Deepseek-R1-Distill at 1.5B, 3B, 7B and 14B. The normalized brain alignment of DeepSeek models across whole brain and language ROI (IFG) is presented in Fig. 27. We also group the quantized variants and present a comparison of the base vs. quantized models across language ROIs in Fig 27. From Fig 27, we observe the same trend: 3B SLMs maintain brain alignment comparable to 14B models, while 1B-1.5B models consistently drop in brain alignment. Notably, the DeepSeek-R1-Distill 14B model shows only a modest improvement over the 7B version, and its alignment is matched by the 3B DeepSeek model, suggesting that 3B SLMs provide sufficient representational capacity for studying brain-LM alignment within this scale regime. We also present the average normalized brain alignment across Language ROIs for DeepSeek-R1 model, comparing SLMs, LLMs, and individual quantized variants in Fig. 28 and the grouped quantized variants in Fig. 29.

## M. Encoding Performance on Naturalistic Reading fMRI Dataset

We have now extended our experiments to an additional dataset i.e. we performed voxelwise encoding on the Moth Radio Hour Reading fMRI dataset (Deniz et al., 2019), which contains the same nine subjects and large number of samples under a different linguistic paradigm (i.e. reading). This additional evaluation allows us to assess the general-

izability of our findings across datasets and tasks. We use Qwen models (Qwen2.5-1.5b, Qwen2.5-3b, Qwen2.5-7b and Qwen2.5-14b) to validate the brain alignment to examine whether 3b SLMs maintain similar brain alignment to larger versions of the models. From Fig. 30, we observe that 3B SLMs yield brain alignment comparable to the 7B and 14B Qwen2.5 models, whereas 1.5B SLMs exhibit a clear drop in brain alignment on the Reading Brain dataset.

## N. Decoder gap: Brain Decoding (Stimulus Reconstruction)

To verify that the brain-aligned representations learned by the small language models (SLMs) studied here are rich enough to support stimulus reconstruction, we perform brain decoding to reconstruct text stimuli from fMRI brain activity using LLaMA-3-8B, and two SLMS (LLaMA-3-3B and LLaMA-3-1B).

Inspired by BrainLLM Ye et al. (2025), we perform end-to-end text stimulus reconstruction from fMRI brain activity. We follow the same BrainLLM methodology, where we use the same Moth Radio Hour dataset (11 stories) with the same train/test split, where ten stories are used for training and one held-out story is used for generation. Concretely, we train a brain-to-text decoder and report standard text-generation metrics-BLEU-1, WER, METEOR, and BERT-F1-for three models: LLaMA-3-8B, LLaMA-3.2-3B, and LLaMA-3.2-1B (Table 3). Across reconstructed segments per model on test dataset, LLaMA-3.2-3B achieves the best performance on all four metrics (BLEU-1 = 0.120, WER = 4.22, METEOR = 0.110, BERT-F1 = 0.825), slightly outperforming LLaMA-3-8B and clearly improving over the LLaMA-3.2-1B baseline (BLEU-1 = 0.070, METEOR = 0.055, BERT-F1 = 0.811). These BERT-F1 scores in the 0.81–0.83 range indicate that the decoded text reliably preserves the semantic content of the original stimulus, while BLEU-1 in the 0.07–0.12 range is in line with prior work where exact word-level recovery from fMRI is known to be challenging.

To make the reconstruction quality more interpretable, we also include qualitative examples comparing ground-truth text and decoded outputs (Table 21). These examples illustrate that the decoder often recovers the overall meaning, emotional tone, and discourse context, even when individual words differ-e.g., reconstructions that correctly express embarrassment, uncertainty, or interactions with children, despite not matching every token verbatim.

**Regional decoding.** We further extend the decoding analysis to three ROI groups, Auditory Cortex (AC), Association, and Prefrontal using the same three models (Table 22). The regional results are as follows: (i) As shown in Table 22, we observe a consistent ordering across all metrics: Frontal

*Table 13.* Pairwise comparisons of brain-alignment differences across quantization methods for LLaMA-3 models. Each Table reports mean differences ($\Delta$), $t$-statistics, and two-sided significance tests for 8B (left), 3B (right), and 1B (bottom).

| Comparison (A–B) | $\Delta$ | $t(8)$ | Sig. |
|---|---|---|---|
| LLaMA-3-8B – AWQ | -0.010 | -inf | $p < 0.001$ |
| LLaMA-3-8B – GPTQ | 0.020 | inf | $p < 0.001$ |
| LLaMA-3-8B – SmoothQuant | 0.005 | 213621227803258.03 | $p < 0.001$ |
| AWQ – GPTQ | 0.030 | inf | $p < 0.001$ |
| AWQ – SmoothQuant | 0.015 | 640834349907832.12 | $p < 0.001$ |
| GPTQ – SmoothQuant | -0.015 | -640834349907835.25 | $p < 0.001$ |

| Comparison (A–B) | $\Delta$ | $t(8)$ | Sig. |
|---|---|---|---|
| LLaMA-3-3B – AWQ | 0.047 | 1.85 | n.s. ($p \approx 0.12$) |
| LLaMA-3-3B – GPTQ | 0.059 | 2.57 | $p < 0.05$ |
| LLaMA-3-3B – SmoothQuant | 0.036 | 1.47 | n.s. ($p \approx 0.20$) |
| AWQ – GPTQ | 0.012 | 1.87 | n.s. ($p \approx 0.12$) |
| AWQ – SmoothQuant | -0.011 | -3.28 | $p < 0.05$ |
| GPTQ – SmoothQuant | -0.023 | -4.08 | $p < 0.01$ |

| Comparison (A–B) | $\Delta$ | $t(8)$ | Sig. |
|---|---|---|---|
| LLaMA-3-1B – AWQ | 0.035 | 1.88 | n.s. ($p \approx 0.12$) |
| LLaMA-3-1B – GPTQ | 0.076 | 6.64 | $p < 0.01$ |
| LLaMA-3-1B – SmoothQuant | 0.002 | 0.56 | n.s. ($p \approx 0.60$) |
| AWQ – GPTQ | 0.041 | 3.20 | $p < 0.05$ |
| AWQ – SmoothQuant | -0.032 | -1.78 | n.s. ($p \approx 0.14$) |
| GPTQ – SmoothQuant | -0.074 | -5.61 | $p < 0.01$ |

*Figure 27.* DeepseekR1: Normalized brain alignment was computed by averaging across participants, layers, and voxels. Red: 14b, Blue: 7b, Green: 3b, Orange: 1.5b, Solid: full-precision SLMs/LLMs, Patterned: quantized models. * at a particular bar indicates that the model's prediction performance is significantly better than 1b/1.5b SLMs. The top row shows whole-brain normalized alignment, while the bottom row focuses on a language-selective ROI (IFG).

> Association > Auditory Cortex, which closely mirrors the regional pattern observed in the encoding analysis. (ii) Across all three ROI groups, the 3B model matches or out-performs the 8B model, while the 1B model consistently underperforms, consistent with our main sufficiency claims on brain encoding.

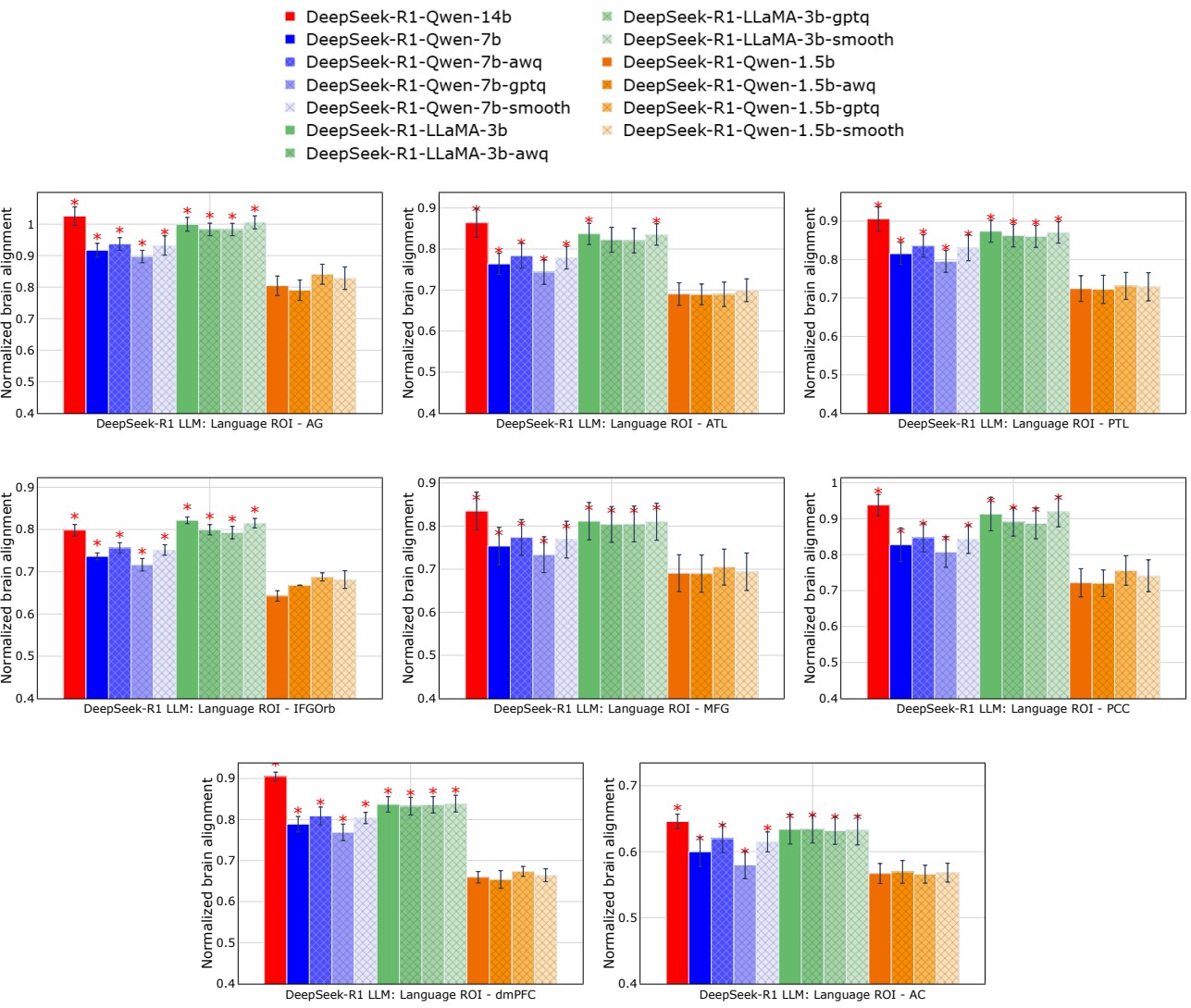

*Figure 28.* Normalized Predictivity of SLMs, LLMs, and Quantized Language Models for DeepSeek-R1 models.

*Table 14.* Quantization method performance across LLaMA models (mean ± std over 9 subjects).

| Model | FP16 | AWQ | GPTQ | SmoothQuant |
|---|---|---|---|---|
| LLaMA-3-8B | 0.901 ± 0.094 | 0.911 ± 0.094 | 0.881 ± 0.094 | 0.896 ± 0.094 |
| LLaMA-3-3B | 0.929 ± 0.071 | 0.882 ± 0.102 | 0.870 ± 0.092 | 0.893 ± 0.102 |
| LLaMA-3-1B | 0.830 ± 0.091 | 0.795 ± 0.133 | 0.754 ± 0.115 | 0.828 ± 0.092 |

Overall, these new brain decoding experiments show that our SLMs are not only good encoders of brain activity, but also support meaningful decoding: they can reconstruct linguistically coherent text from fMRI with high semantic fidelity and reasonable word-level accuracy. We emphasize that decoding is not the primary focus of the present work, but we report it as supporting evidence that the SLM representations identified by our voxel-wise encoding analyses are rich enough to support meaningful stimulus reconstruction.

## O. Effect of Pruning

We have now included unstructured pruning which is equivalent to quantization and now report preliminary results for Qwen-2.5 models. In particular, we perform unstructured magnitude pruning on the linear layers of Qwen2.5-3B and Qwen2.5-1.5B, at sparsity levels 0.1, 0.25, and 0.5. For Qwen2.5-3B, Table 5 summarizes brain alignment (mean ± s.e.m. across subjects) for the base, quantized variants, and pruned models, showing that AWQ and SmoothQuant slightly improve over the FP16 baseline (0.933 ± 0.035 and 0.930 ± 0.035 vs. 0.924 ± 0.033), GPTQ is modestly lower (0.910 ± 0.037), and unstructured pruning up to 50% keeps alignment in a narrow range (0.910–0.907 with s.e.m. 0.032–0.043).

These results suggest that, for Qwen2.5-3B, moderate un-

*Table 15.* Qwen2.5 Cross-Model Size Comparisons

| Comparison | Base Mean | Comp Mean | Difference | Effect Size | % Sig | Median p | Interpretation |
|---|---|---|---|---|---|---|---|
| 7B vs 3B | 0.7208 | 0.6996 | -0.0213 | 0.596 | 0.0% | 0.549 | Not significant |
| 14B vs 3B | 0.7208 | 0.6912 | -0.0296 | 1.184 | 27.8% | 0.111 | Trend ($p < 0.15$) |
| 14B vs 7B | 0.6996 | 0.6912 | -0.0083 | 0.874 | 7.8% | 0.356 | Not significant |

*Table 16.* ROI-Specific Layer Performance Summary

| ROI | Model (Qwen2.5) | Overall Mean±SD | Best Layer | Best Layer Mean±SD | Worst Layer | Layer Range |
|---|---|---|---|---|---|---|
| AG | 1.5B | 0.8341±0.1546 | L14 | 0.9466±0.1099 | L1 | 0.5478–0.9466 |
| AG | 3B | 0.8558±0.1823 | L22 | 1.0091±0.1157 | L1 | 0.5418–1.0091 |
| AG | 7B | 0.8417±0.2119 | L15 | 0.9818±0.1390 | L1 | 0.4535–0.9818 |
| AG | 14B | 0.8178±0.2085 | L24 | 1.0143±0.1431 | L1 | 0.4913–1.0143 |
| ATL | 1.5B | 0.7161±0.1248 | L14 | 0.7904±0.1302 | L1 | 0.5296–0.7904 |
| ATL | 3B | 0.7251±0.1422 | L21 | 0.8304±0.1309 | L1 | 0.5108–0.8304 |
| ATL | 7B | 0.7016±0.1611 | L16 | 0.7891±0.1072 | L1 | 0.4620–0.7891 |
| ATL | 14B | 0.6974±0.1579 | L25 | 0.8362±0.1333 | L1 | 0.4630–0.8362 |
| PTL | 1.5B | 0.7697±0.1300 | L15 | 0.8335±0.1078 | L1 | 0.5944–0.8335 |
| PTL | 3B | 0.7763±0.1394 | L21 | 0.8725±0.1313 | L1 | 0.5843–0.8725 |
| PTL | 7B | 0.7443±0.1616 | L15 | 0.8353±0.1231 | L3 | 0.5166–0.8353 |
| PTL | 14B | 0.7471±0.1513 | L25 | 0.8592±0.1386 | L1 | 0.5353–0.8592 |
| IFG | 1.5B | 0.7726±0.1716 | L14 | 0.8730±0.1480 | L1 | 0.5366–0.8730 |
| IFG | 3B | 0.7801±0.1872 | L21 | 0.9309±0.1377 | L1 | 0.5058–0.9309 |
| IFG | 7B | 0.7665±0.1996 | L15 | 0.9045±0.1305 | L3 | 0.4263–0.9045 |
| IFG | 14B | 0.7639±0.2079 | L25 | 0.9516±0.1027 | L1 | 0.4938–0.9516 |
| MFG | 1.5B | 0.6929±0.1494 | L15 | 0.7618±0.1540 | L1 | 0.5068–0.7618 |
| MFG | 3B | 0.6984±0.1732 | L21 | 0.8006±0.1746 | L1 | 0.5213–0.8006 |
| MFG | 7B | 0.6716±0.1883 | L15 | 0.7732±0.1553 | L1 | 0.4283–0.7732 |
| MFG | 14B | 0.6689±0.1642 | L24 | 0.7921±0.1506 | L1 | 0.4682–0.7921 |
| IFGOrb | 1.5B | 0.6193±0.1589 | L14 | 0.7249±0.1007 | L1 | 0.4160–0.7249 |
| IFGOrb | 3B | 0.6401±0.1712 | L22 | 0.7647±0.0866 | L1 | 0.4072–0.7647 |
| IFGOrb | 7B | 0.6403±0.1966 | L15 | 0.7765±0.1080 | L1 | 0.2891–0.7765 |
| IFGOrb | 14B | 0.6159±0.1878 | L25 | 0.7803±0.0772 | L3 | 0.3628–0.7803 |
| PCC | 1.5B | 0.7638±0.1732 | L15 | 0.8618±0.1261 | L1 | 0.4904–0.8618 |
| PCC | 3B | 0.7651±0.1878 | L22 | 0.9093±0.1203 | L1 | 0.4654–0.9093 |
| PCC | 7B | 0.7509±0.2118 | L15 | 0.8904±0.0938 | L1 | 0.3571–0.8904 |
| PCC | 14B | 0.7235±0.2095 | L24 | 0.9367±0.1163 | L1 | 0.4205–0.9367 |
| dmPFC | 1.5B | 0.6884±0.1392 | L14 | 0.8089±0.0991 | L1 | 0.4443–0.8089 |
| dmPFC | 3B | 0.6964±0.1726 | L21 | 0.8685±0.1049 | L1 | 0.4192–0.8685 |
| dmPFC | 7B | 0.6888±0.1825 | L15 | 0.8226±0.1018 | L1 | 0.3628–0.8226 |
| dmPFC | 14B | 0.6692±0.1913 | L25 | 0.8859±0.1320 | L1 | 0.3712–0.8859 |
| AC | 1.5B | 0.5587±0.0906 | L15 | 0.5963±0.0762 | L1 | 0.4727–0.5963 |
| AC | 3B | 0.5634±0.1056 | L21 | 0.6303±0.0890 | L1 | 0.4668–0.6303 |
| AC | 7B | 0.5204±0.1243 | L15 | 0.5867±0.1017 | L3 | 0.3757–0.5867 |
| AC | 14B | 0.5241±0.0909 | L24 | 0.5831±0.1042 | L1 | 0.4095–0.5831 |

*Table 17.* Qwen2.5: Subject Variability at Optimal Layers

| Model | Layer | Mean ± SD | SEM | 95% CI | CV (%) | Variability |
|---|---|---|---|---|---|---|
| 1.5B | L14 | 0.7956 ± 0.0892 | 0.0364 | [0.724, 0.867] | 11.21% | LOW |
| 3B | L22 | 0.8309 ± 0.0835 | 0.0341 | [0.764, 0.898] | 10.05% | LOW |
| 7B | L15 | 0.8171 ± 0.0852 | 0.0348 | [0.749, 0.885] | 10.43% | LOW |
| 14B | L24 | 0.8411 ± 0.0883 | 0.0361 | [0.770, 0.912] | 10.50% | LOW |

*Table 18.* Cross-Model Comparisons for Llama-3

| Comparison | Base Mean | Comp Mean | Difference | Effect Size | % Sig | Median p | Interpretation |
|---|---|---|---|---|---|---|---|
| 8B vs 3B | 0.7409 | 0.7300 | -0.0110 | 0.425 | 5.2% | 0.417 | Not significant |
| 14B vs 3B | 0.7409 | 0.7494 | +0.0084 | 0.811 | 27.4% | 0.227 | Moderate effect |
| 14B vs 8B | 0.7285 | 0.7517 | +0.0232 | 0.577 | 14.6% | 0.320 | Not significant |

structured pruning (10–25%) preserves brain alignment at a level comparable to quantized or full-precision models, with little change in SER, while aggressive pruning (50%) begins to degrade linguistic competence (SER increases) despite only a small drop in alignment. We view these pruning experiments as complementary to our main quantization results: they show that both post-training quantization and unstructured pruning can preserve brain alignment to a surprising degree, but they also highlight potential trade-offs with linguistic competence at high sparsity.

We acknowledge that we still do not systematically explore structured compression across model components, where different methods could be compared under matched compression ratios to assess their differential impact on both linguistic competence and brain alignment. We now explicitly note that this is an important direction for future work, but falls outside the scope of the current paper given the substantial additional experiments it would require.

Although our model suite includes the DeepSeek-R1-Distill family (which is itself a product of knowledge distillation), in this work we do not systematically study distillation as a compression method. We treat DeepSeek as an additional, pretrained model family for evaluation, and focus our controlled compression experiments on post-training quantization (and preliminary unstructured pruning). A careful comparison of different distillation strategies under matched compression ratios is therefore an important direction for future work.

*Table 19.* LLaMA3: ROI-Specific Layer Performance Summary

| ROI | Model (LLaMA-3) | Optimal Layer | Optimal Value | SD | CI Low | CI High | High-Perf Range | Total Layers |
|---|---|---|---|---|---|---|---|---|
| AG | 1B | 9 | 0.9189 | 0.0867 | 0.8495 | 0.9882 | 7–14 | 16 |
| ATL | 1B | 9 | 0.7751 | 0.0789 | 0.7119 | 0.8382 | 5–14 | 16 |
| PTL | 1B | 8 | 0.8149 | 0.1134 | 0.7242 | 0.9056 | 5–14 | 16 |
| IFG | 1B | 8 | 0.8312 | 0.1209 | 0.7345 | 0.9279 | 5–15 | 16 |
| MFG | 1B | 9 | 0.7477 | 0.1623 | 0.6178 | 0.8775 | 7–15 | 16 |
| IFGOrb | 1B | 7 | 0.6937 | 0.0868 | 0.6242 | 0.7631 | 5–15 | 16 |
| PCC | 1B | 7 | 0.8364 | 0.1109 | 0.7476 | 0.9251 | 5–14 | 16 |
| dmPFC | 1B | 9 | 0.7591 | 0.0995 | 0.6795 | 0.8387 | 7–11 | 16 |
| EarlyAud | 1B | 8 | 0.6010 | 0.0802 | 0.5368 | 0.6651 | 5–9 | 16 |
| AG | 3B | 12 | 1.0206 | 0.1073 | 0.9347 | 1.1065 | 9–19 | 28 |
| ATL | 3B | 12 | 0.8150 | 0.1111 | 0.7261 | 0.9039 | 9–21 | 28 |
| PTL | 3B | 13 | 0.8654 | 0.1246 | 0.7657 | 0.9651 | 9–21 | 28 |
| IFG | 3B | 13 | 0.9255 | 0.1244 | 0.8259 | 1.0250 | 9–22 | 28 |
| MFG | 3B | 13 | 0.8127 | 0.1439 | 0.6975 | 0.9278 | 11–22 | 28 |
| IFGOrb | 3B | 12 | 0.8030 | 0.0739 | 0.7439 | 0.8622 | 11–17 | 28 |
| PCC | 3B | 14 | 0.9151 | 0.1105 | 0.8267 | 1.0036 | 9–22 | 28 |
| dmPFC | 3B | 12 | 0.8464 | 0.0846 | 0.7787 | 0.9141 | 11–17 | 28 |
| EarlyAud | 3B | 13 | 0.6271 | 0.0905 | 0.5547 | 0.6995 | 12–17 | 28 |
| AG | 7B | 14 | 0.9802 | 0.1095 | 0.8927 | 1.0678 | 7–29 | 32 |
| ATL | 7B | 14 | 0.8098 | 0.1198 | 0.7139 | 0.9056 | 7–25 | 32 |
| PTL | 7B | 14 | 0.8644 | 0.1082 | 0.7778 | 0.9509 | 7–25 | 32 |
| IFG | 7B | 14 | 0.8974 | 0.1270 | 0.7958 | 0.9990 | 7–25 | 32 |
| MFG | 7B | 13 | 0.7792 | 0.1082 | 0.6926 | 0.8658 | 7–26 | 32 |
| IFGOrb | 7B | 14 | 0.7415 | 0.1004 | 0.6612 | 0.8219 | 7–29 | 32 |
| PCC | 7B | 15 | 0.8835 | 0.1203 | 0.7872 | 0.9797 | 7–29 | 32 |
| dmPFC | 7B | 14 | 0.8353 | 0.1043 | 0.7519 | 0.9188 | 7–18 | 32 |
| EarlyAud | 7B | 13 | 0.5940 | 0.0756 | 0.5336 | 0.6545 | 7–21 | 32 |
| AG | 14B | 17 | 0.9861 | 0.1001 | 0.9061 | 1.0662 | 5–38 | 39 |
| ATL | 14B | 16 | 0.7910 | 0.1175 | 0.6970 | 0.8851 | 6–38 | 39 |
| PTL | 14B | 16 | 0.8204 | 0.1121 | 0.7308 | 0.9101 | 6–35 | 39 |
| IFG | 14B | 16 | 0.8709 | 0.1055 | 0.7865 | 0.9553 | 8–35 | 39 |
| MFG | 14B | 13 | 0.7340 | 0.1315 | 0.6288 | 0.8393 | 5–20 | 39 |
| IFGOrb | 14B | 17 | 0.7772 | 0.0460 | 0.7404 | 0.8141 | 12–30 | 39 |
| PCC | 14B | 16 | 0.9076 | 0.1142 | 0.8162 | 0.9990 | 8–35 | 39 |
| dmPFC | 14B | 16 | 0.8389 | 0.1310 | 0.7341 | 0.9436 | 14–23 | 39 |
| EarlyAud | 14B | 17 | 0.5544 | 0.0720 | 0.4969 | 0.6120 | 5–20 | 39 |

*Table 20.* LLaMA3: Subject Variability at Optimal Layers

| Model | Layer | Mean ± SD | SEM | 95% CI | CV (%) | Variability |
|---|---|---|---|---|---|---|
| 1B | L8 | 0.8003 ± 0.1065 | 0.0435 | [0.688, 0.912] | 13.31% | MODERATE |
| 3B | L13 | 0.9083 ± 0.0824 | 0.0337 | [0.822, 0.995] | 9.08% | LOW |
| 8B | L10 | 0.8588 ± 0.1115 | 0.0455 | [0.742, 0.976] | 12.98% | MODERATE |
| 14B | L16 | 0.8679 ± 0.0936 | 0.0382 | [0.770, 0.966] | 10.78% | MODERATE |

## P. Model Size vs. Brain Alignment

Figs. 31, 32, and 33 plot Model size (GB) vs. Average Normalized brain alignment for Qwen2.5, LLaMA-3, and DeepSeek-R1 models (1.5B/3B/7B/14B) and their AWQ, GPTQ, and SmoothQuant variants. Across all three families, the 3B SLMs and their AWQ/SmoothQuant variants generally lie slightly above or very close to their FP16 counterparts at substantially reduced size, whereas GPTQ variants tend to fall slightly below the FP16 models despite achieving stronger compression.

## Q. Limitations

While prior work has evaluated brain encoding with LLMs up to 72B parameters (Antonello et al., 2024), our analysis of efficiency-oriented regimes extends up to 14B parameters and includes additional architectures such as DeepSeek (Appendices I and L). These results confirm that the observed performance plateau at the 3B scale generalizes across model families; however, evaluating extremely large models (e.g., 70B+) under post-training compression remains computationally challenging and is an important direction for future work. Second, parameter count is not perfectly isolated from training data, token budgets, and optimization, particularly across model families. We mitigate this by basing our primary claims on within-family comparisons (Qwen2.5, LLaMA-3.2, DeepSeek-R1), where architecture, tokenizer, and training pipeline are held fixed, and on compression interventions that vary effective capacity without altering training; we therefore do not claim a purely causal effect of parameter count, but interpret our results as evidence that effective capacity is a primary driver of brain alignment. Third, although our primary focus is on quantization, we validate our conclusions using unstructured pruning (Appendix O), showing that moderate sparsity preserves brain alignment comparable to quantization. A broader comparison with other compression strategies, such as structured pruning or knowledge distillation, would further clarify how different efficiency interventions affect neural representations. Fourth, our experiments focus primarily on fMRI data collected during naturalistic listening, supplemented by validation on a naturalistic reading dataset (Appendix M). While this demonstrates robustness across tasks, future work incorporating higher temporal resolution modalities such as

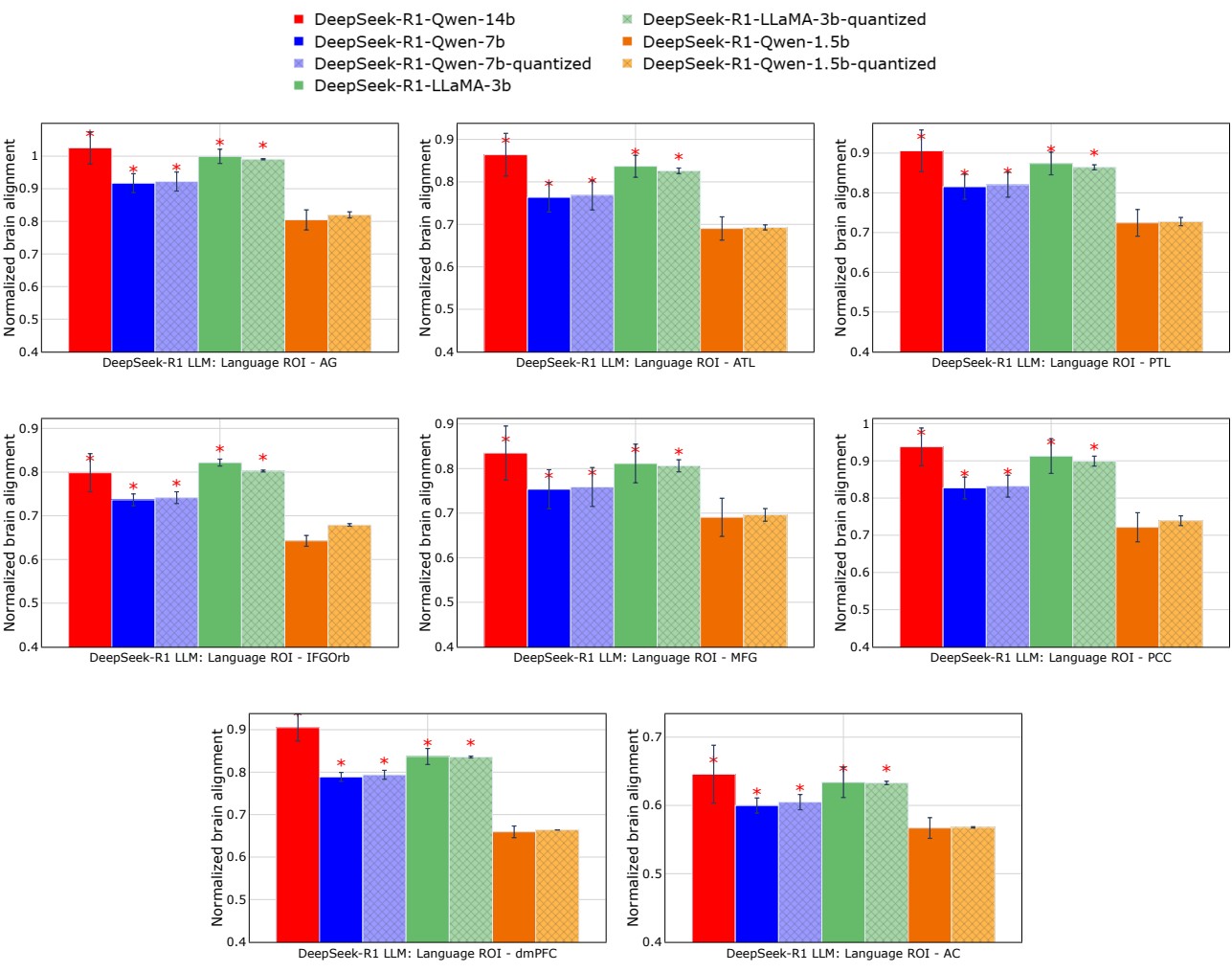

*Figure 29.* Normalized predictivity of DeepSeek-R1 SLMs and LLMs, including grouped comparisons of the base and quantized variants.

*Table 21.* Example fMRI-based stimulus reconstructions for LLaMA-3-8B, LLaMA-3.2-3B, and LLaMA-3.2-1B.

| Example | Variant | Text | Metrics |
|---|---|---|---|
| 1 | Ground Truth | *little sad leaving here about* | – |
| | LLaMA-3-8B | huge fan of the show so i'm really excited to be able to share this with you guys and i hope you'll enjoy it as much as i enjoyed watching | BLEU-1 = 0.0000, WER = 6.0000, METEOR = 0.0000 |
| | LLaMA-3.2-3B | in this beautiful city. I started to walk in this beautiful in this beautiful city. I started to walk in this beautiful in this beautiful city. I started to | BLEU-1 = 0.0000, WER = 5.8000, METEOR = 0.0000 |
| | LLaMA-3.2-1B | , you know what i'm talking about. You know what i'm talking about. You know what i'm talking about. You know what i'm talking about | BLEU-1 = 0.0400, WER = 4.8000, METEOR = 0.0714 |
| 2 | Ground Truth | *and that i had to hold onto him* | – |
| | LLaMA-3-8B | i was going to have to take him to the hospital but i had no idea what to do with him so i went to a friend of mine and he told | BLEU-1 = 0.3750, WER = 3.5000, METEOR = 0.2885 |
| | LLaMA-3.2-3B | , she's crying and then she sees me and she com, she's crying and then she sees me and she com, she's crying and then she sees | BLEU-1 = 0.1852, WER = 3.2500, METEOR = 0.0505 |
| | LLaMA-3.2-1B | could see her face and she was smiling at me and then she said "hi" and i said "hi" back to her and then she turned around and | BLEU-1 = 0.2500, WER = 3.1250, METEOR = 0.1500 |
| 3 | Ground Truth | *to be a teacher with all these kids* | – |
| | LLaMA-3-8B | for them to see their children i had no idea how hard it was for them to see their children i had no idea how hard it was for them to see | BLEU-1 = 0.0938, WER = 3.8750, METEOR = 0.0962 |
| | LLaMA-3.2-3B | is a great place to live, work, and play. It's also home to some of the best restaurants, bars, and coffee shops in the city. | BLEU-1 = 0.1200, WER = 3.0000, METEOR = 0.2635 |
| | LLaMA-3.2-1B | and i said you know like and i said you know like and i said you know like and i said you know like and i said you know like and i | BLEU-1 = 0.0000, WER = 4.0000, METEOR = 0.0000 |

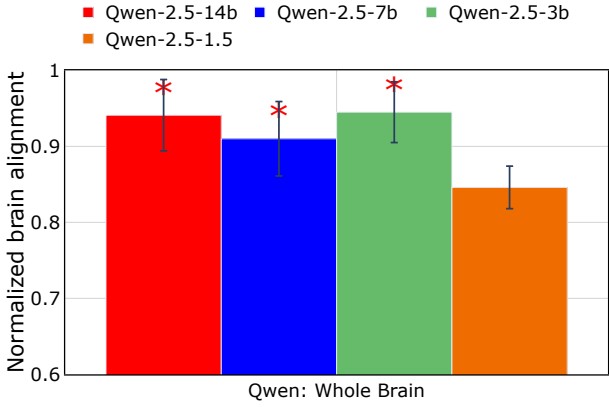

*Figure 30.* Reading brain dataset (Qwen-2.5): Normalized brain alignment was computed by averaging across participants, layers, and voxels. Red: 14b, Blue: 7b, Green: 3b, Orange: 1.5b, Solid: full-precision SLMs/LLMs. * at a particular bar indicates that the model's prediction performance is significantly better than 1.5b SLMs. The plot shows whole-brain normalized alignment.

*Table 22.* Decoding performance comparison across ROIs and LLaMA models.

| ROI | Model | BLEU-1 | WER | METEOR | BERT-F1 |
|---|---|---|---|---|---|
| Frontal | LLaMA-3-8B | 0.124 | 4.302 | 0.121 | 0.852 |
| | LLaMA-3.2-3B | 0.143 | 3.817 | 0.131 | 0.861 |
| | LLaMA-3.2-1B | 0.092 | 5.006 | 0.072 | 0.821 |
| Association | LLaMA-3-8B | 0.121 | 4.193 | 0.111 | 0.841 |
| | LLaMA-3.2-3B | 0.128 | 4.012 | 0.119 | 0.849 |
| | LLaMA-3.2-1B | 0.083 | 5.487 | 0.061 | 0.801 |
| Auditory | LLaMA-3-8B | 0.103 | 4.287 | 0.092 | 0.821 |
| | LLaMA-3.2-3B | 0.112 | 4.108 | 0.101 | 0.832 |
| | LLaMA-3.2-1B | 0.071 | 5.624 | 0.053 | 0.771 |

*Table 23.* Comparison of quantization and pruning for Qwen2.5-1.5B (mean $\pm$ SEM across subjects).

| Variant | Mean $\pm$ SEM | 95% CI | Notes |
|---|---|---|---|
| FP16 (baseline) | 0.830 $\pm$ 0.025 | [0.781, 0.879] | full precision |
| AWQ | 0.854 $\pm$ 0.031 | [0.793, 0.915] | INT4 quantization |
| GPTQ | 0.828 $\pm$ 0.026 | [0.777, 0.879] | INT4 quantization |
| SmoothQuant | 0.836 $\pm$ 0.025 | [0.787, 0.885] | INT4 quantization |
| Prune 10% | 0.847 $\pm$ 0.026 | [0.796, 0.898] | unstructured pruning |
| Prune 25% | 0.824 $\pm$ 0.025 | [0.775, 0.873] | unstructured pruning |
| Prune 50% | 0.608 $\pm$ 0.053 | [0.504, 0.712] | unstructured pruning |

MEG or ECoG could better capture the dynamics of language processing. Fifth, while our $\sim$3B SLMs saturation result is consistent across two naturalistic fMRI datasets (podcast listening and story reading) involving continuous narrative comprehension, the threshold may not generalize to all cognitive contexts. Prior work suggests that brain alignment tracks formal linguistic competence (syntax, morphology) more closely than functional competence (reasoning, world knowledge), and that the relationship between model size and alignment weakens for capabilities beyond core language processing (AlKhamissi et al., 2025). Stimuli demanding deeper reasoning, abstract problem solving, or broader world knowledge may therefore shift this saturation point, and testing across such stimuli is an important

direction for future work. Sixth, while our main analyses emphasize brain encoding, we include complementary decoding experiments (Appendix N) that assess stimulus reconstruction from brain activity. These results suggest that encoding alignment does not always guarantee decoding fidelity, motivating future work that more directly links neural alignment to downstream brain–computer interface performance. Finally, our scaling analysis does not disentangle depth from width. The model families we study (Qwen2.5, LLaMA-3.2, DeepSeek-R1) vary parameter count by changing both axes simultaneously, so the observed saturation at the 3B SLMs scale cannot be attributed cleanly to parameter count alone. Prior work suggests that compositional depth may be the more relevant capacity dimension for brain alignment: encoding performance has been linked to the intrinsic dimensionality of intermediate (rather than output) layers (Antonello et al., 2024), and brain alignment more closely tracks formal linguistic competence (which relies on compositional rules) than functional competence (which benefits more from width) (AlKhamissi et al., 2025). Evaluating depth and width as independent factors at matched parameter count is a natural extension of this work.

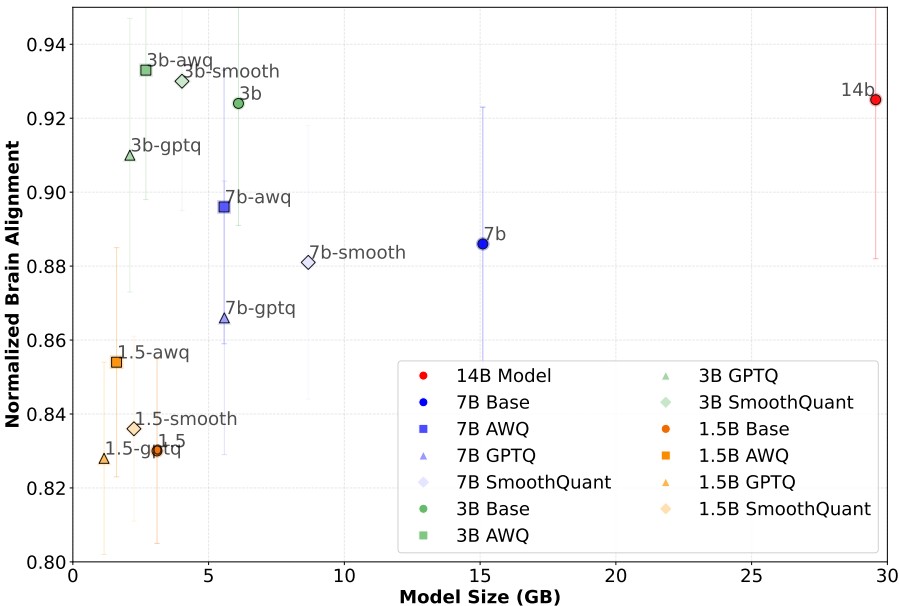

*Figure 31.* Qwen2.5: Plot of Model Size (x-axis) vs. Normalized Brain Alignment (y-axis).

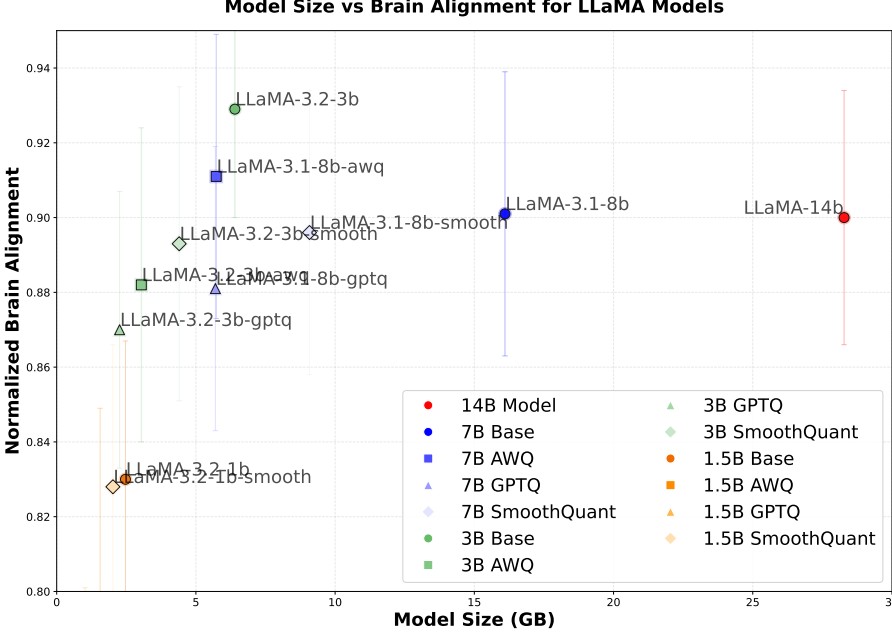

*Figure 32.* LLaMA-3: Plot of Model Size (x-axis) vs. Normalized Brain Alignment (y-axis).

*Table 24.* DeepSeek-R1: Pretrained Transformer-based language models spanning small (SLMs) and large (LLMs) scales, along with their post-training compressed variants.

| Model Family | SLMs | | LLMs | |
|---|---|---|---|---|
| | Size | Layers | Size | Layers |
| DeepSeek-R1 | 1.5B (3.55GB) | 28 | 7B (15.23GB) | 28 |
| | 3B (6.43GB) | 28 | 14B (29.54) | 48 |

| Model Family | SLMs (GB) | | | LLMs (GB) | | |
|---|---|---|---|---|---|---|
| | AWQ | GPTQ | SmoothQuant | AWQ | GPTQ | SmoothQuant |
| DeepSeek-R1 | 1.62 | 1.61 | 2.25 | 5.57 | 5.58 | 8.71 |
| | 2.69 | 2.37 | 4.02 | | | |

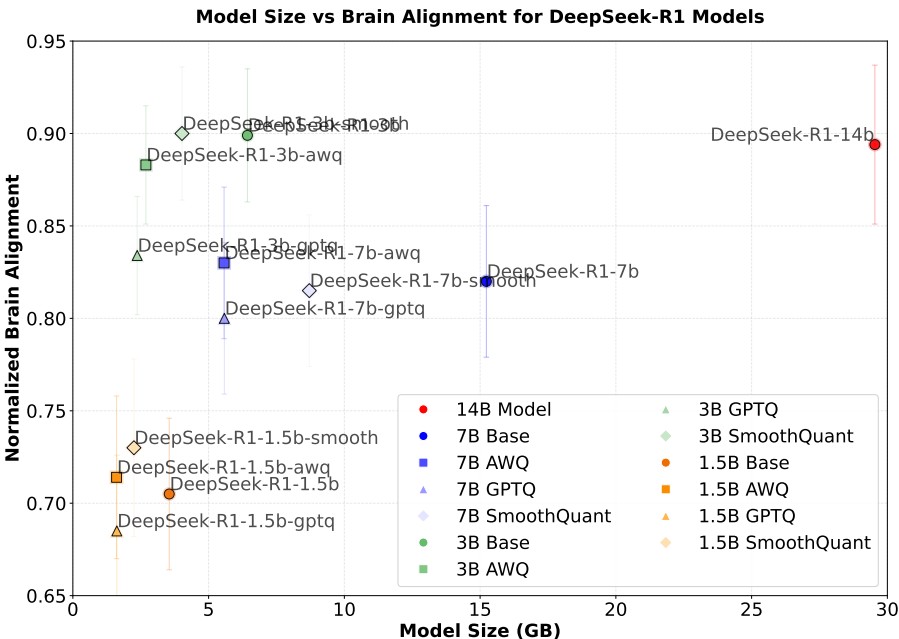

*Figure 33.* DeepSeek-R1: Plot of Model Size (x-axis) vs. Normalized Brain Alignment (y-axis).

