# OpenReview forum: "Linguistic Properties and Model Scale in Brain Encoding: From Small to Compressed Language Models"
_ICML.cc/2026/Conference — ICML 2026 spotlight_

### Official Review · Reviewer_oS9t · 2026-03-02

**Soundness:** 2
**Presentation:** 2
**Significance:** 3
**Originality:** 2
**Overall Recommendation:** 4
**Confidence:** 2

**Summary:**

This paper investigates the optimal size of LMs at which human brain activities can be modeled well. They find that 3B is the point at which modeling performance saturates. They also show that pruning and compression of LMs generally preserve the modeling capacity, but excessive pruning and compression is detrimental. They further provide analyses on linguistic probing tasks.

**Compliance With Llm Reviewing Policy:**

Affirmed.

**Final Justification:**

My initial concerns were primarily about methodology and originality/significance. On the methodological side, the authors clarified that a context size of 20 is standard practice in neuroimaging studies, which satisfactorily addresses that concern.

I remain somewhat unconvinced about the novelty of all aspects of the work. In particular, inverse scaling has already been discussed extensively in reading-time research, and some findings, such as differences in linguistic competencies across model scales, are not entirely new. That said, the reported saturation around 3B parameters appears potentially important, and I agree that the dissociation between linguistic competencies and brain alignment is an interesting result that merits further investigation.

Overall, while I still have some reservations about originality, the rebuttal resolved my main methodological concern and increased my confidence in the paper, so I have updated my score accordingly.

**Key Questions For Authors:**

1. Why is the context size limited to 20 tokens? This is far from a realistic setting in which LLMs operate (it's at least 1-2k and modern LLMs are orders of magnitude larger than that). In reading time studies, usually a whole document/story/passage is passed to the model to simulate the context human participants get. Is there something special about the 20-token context window?
2. Also tokenizer has nothing to do with context size. I wasnt' sure what LL144-148 (right) meant: "To obtain the stimulus features from these pretrained models, we constrain the tokenizer to use a maximum context of 20 words. Given the constrained context length, each word is successively input to the network with at most C (=20) previous words."

**Limitations:**

yes

**Strengths And Weaknesses:**

* Soundness
   * This paper covers a wide range of models both in architecture and size (1B-14B).
   * The 20-token context size was a bit confusing, which I explain in the question section.

* Presentation
   * Many neuroscience acronyms that make the paper harder to read, which aren't quite central to the understanding of the paper's contribution. E.g. AG and PCC (among many others I think) only appear 1-2 times in the main body. Putting them in the appendix may improve the readability of the paper.

* Significance and Originality
   * The contribution seems limited. Inverse scaling (larger models do worse in neuroscience/psycholinguistic tasks) is well known (e.g. Oh & Schuler, 2022; https://arxiv.org/abs/2212.12131).
   * Besides the inverse scaling, linguistic probing of various model sizes has been studied extensively (e.g. BabyLM challenges). Pruning and compression and their effects on cognitive modeling + linguistic capabilities might be novel.

---

> ### Author Rebuttal · Authors · 2026-03-31
>
> **Q1/W1**
>
> * Our choice of a 20-token context window is a deliberate and standard design decision in brain encoding studies, motivated by cognitive, neuroimaging, and methodological considerations.
> * Established precedent in brain-language alignment. Prior work systematically varying context length in fMRI encoding models showed that encoding performance saturates at moderate context lengths, with diminishing returns beyond ~20 words (Jain&Huth, NeurIPS 2018; Toneva et al., 2019). This finding has been adopted as standard practice across the field (Aw&Toneva 2023; Oota et al., 2024). In these studies, each word's representation is computed from a bounded preceding context rather than the full document, to better approximate how humans process language incrementally online.
> * Second, this is also motivated by the hemodynamic response: fMRI reflects brain activity over only a short temporal window, so very long contexts are unlikely to match the measured signal.
> * Third, restricting context also avoids leakage of future or distant information and better reflects incremental human language processing.
> * Importantly, the consistent alignment patterns we observe across model sizes and families suggest that this controlled setup captures the relevant brain-computable representations.
> We will clarify this in the revision.
>
> **Q2/W1**
>
> We thank the reviewer for pointing out this ambiguity, we agree that the current wording is imprecise and potentially misleading.
> * To clarify: we do not modify or constrain the tokenizer itself, nor do we alter the model's native context window. Instead, we apply a sliding input window over the text, where for each target word w_t the model receives at most the preceding C=20 words as input. The tokenizer then operates normally on this truncated input sequence.
> * Concretely, for each target word we construct an input sequence consisting of the preceding C words (or fewer at the start of the narrative), tokenize this sequence using the model's standard tokenizer.
> * This incremental bounded-context is standard in brain encoding studies (Jain&Huth. 2018, Toneva&Wehbe, 2019; Oota et al. 2024) because it better matches the temporal resolution of fMRI and avoids leakage from future or distant context.
> We will revise Lines 144-148 accordingly.
>
> **W2**
>
> * We thank the reviewer for this suggestion. All neuroscience acronyms in the main text are already defined on first use (Section 2.1, Lines 145-155), and additional region details are provided in Appendix B and Table 6.
> * We agree that readers from outside the neuroimaging community may still find the density of acronyms challenging. In the revision, we will add a compact acronym table to the main text and move less central regional discussion to the appendix where appropriate, while keeping key regions in the main paper for clarity.
>
> **W3.1**
>
> We would like to clarify how our work differs from prior findings such as inverse scaling and linguistic probing studies.
> * **Difference from inverse scaling work:** Prior inverse-scaling work studies text-only surprisal as a predictor of behavioral reading-time data. By contrast, our paper studies brain alignment from internal model representations using voxel-wise fMRI encoding under naturalistic stimuli, and asks a different question: what model capacity is sufficient for brain-relevant representations?  Thus, our main contribution is not the generic observation that “bigger is not always better,” but the more specific finding that in brain encoding, ~3B-scale SLMs already match much larger 7B-14B models, while ~1B models show a clear drop, yielding a concrete capacity threshold for brain alignment, and helps clarify what level of linguistic competence is sufficient for these tasks.
>
> **W3.2**
>
> * **Difference from linguistic probing and BabyLM-style work:** We agree that linguistic probing across model sizes is well studied. Our contribution is different in jointly linking model scale (1B-14B), compression (quantization and pruning), brain alignment (fMRI encoding/decoding), and linguistic competence. Specifically, we show that brain alignment saturates early at ~3B even when linguistic competence still varies and compression can reduce task performance. This reveals a dissociation: task competence can degrade, especially under some compression methods, while brain alignment remains largely stable. To our knowledge, this dissociation has not been systematically shown across both scaling and compression regimes.
>
> **Novelty**
>
> * We agree that the compression analysis is one of the most novel parts of the paper. In particular, we treat compression as a controlled intervention on model representations and show that different quantization methods have qualitatively different effects on brain alignment. This suggests that compression is not only an efficiency tool, but also a useful probe for identifying which representational properties are most relevant to brain alignment.

---

> > ### Author Rebuttal · Reviewer_oS9t · 2026-04-03
> >
> > Thank you for the thorough rebuttal! My main concern about the context size has been resolved - I did not know it was a standard practice in neuroimaging studies. I do think some aspects of the findings (especially linguistic competencies at different scale) are not novel, yet the authors' claim that the saturation at 3B seems like a significant finding for the field. I also agree that the dissociation between competencies and brain alignment interesting and worthy of further investigation. I'm updating my score accordingly.

---

> > > ### Author Response · Authors · 2026-04-03
> > >
> > > Thank you for your response and thoughtful revision. We appreciate your constructive feedback, which we believe has significantly improved the clarity and overall quality of the paper.

---

### Official Review · Reviewer_UfDQ · 2026-03-11

**Soundness:** 3
**Presentation:** 1
**Significance:** 3
**Originality:** 3
**Overall Recommendation:** 5
**Confidence:** 4

**Summary:**

This paper systematically compares the brain-encoding performance of multiple language model families (LLaMA, Qwen, and DeepSeek) at scales from 1B to 14B, and further examines the effects of quantization and pruning on brain alignment. The authors perform voxel-wise encoding on natural story comprehension fMRI data and combine this with FlashHolmes probing to analyze language abilities including morphology, syntax, semantics, discourse, and reasoning. The main conclusions are: 3B models have basically reached saturation in brain alignment, while 1B/1.5B models degrade significantly; most quantization methods and moderate pruning have relatively little effect on brain alignment, with GPTQ being an exception; and declines in language ability are not always consistent with declines in brain alignment.

**Compliance With Llm Reviewing Policy:**

Affirmed.

**Final Justification:**

my concerns have been addressed by authers.

**Key Questions For Authors:**

see weeknesses.

**Limitations:**

yes

**Strengths And Weaknesses:**

**Strengths**

1.The research question is clear, focusing on the meaningful issue of whether model scale and compression truly affect brain alignment.

2.The experiments are relatively comprehensive, simultaneously comparing different model families, parameter scales, quantization methods, and pruning strategies.

3.The paper does not only look at brain encoding, but also includes decoding and language probing, making the perspective relatively complete.

4.The conclusions are somewhat insightful: brain alignment may saturate earlier than common NLP capability metrics and may also be more robust to compression.

**Weaknesses**

1.The conclusion that “3B has already saturated” currently seems more like an empirical observation, and the mechanistic explanation is still not strong enough. You may need to introduce some evaluative or statistical metrics to further support this point.

2.The decoding section is relatively weak. It only presents a small number of models and metrics, and its support is not as strong as that of the main encoding results. Therefore, could the decoding section be supplemented with more complete comparisons and statistical tests to strengthen the persuasiveness of this part of the conclusion, for example by presenting the metrics by brain region?

3.The figures are very unclear, and many of them are completely illegible.

---

> ### Author Rebuttal · Authors · 2026-03-31
>
> **Q1/W1**
>
> We agree that the notion of “saturation” should be supported more rigorously beyond qualitative observation.
> * We wish to clarify that extensive statistical analysis is already provided in the main paper and appendices: pairwise differences in encoding performance across subjects for Qwen2.5 (Table 2), LLaMA-3.2 and DeepSeek-R1 (Appendix Tables 9–10), pairwise comparisons of brain-alignment differences across quantization methods (Table 4, Appendix Tables 11), and complete statistical validation across model families, quantization effects, subject variability, and best layer selection (Appendices I, J, K).
> * To address the reviewer’s concern, we clarify and quantify saturation using both statistical and effect-size-based criteria. Specifically, we define saturation as the regime in which (i) further scaling yields no statistically significant improvement, and (ii) effect sizes are negligible relative to inter-subject variability.
> * Our results satisfy both criteria. For example, in Qwen2.5 (Table 2), the difference between 3B and 14B models is effectively zero (∆=0.000, t(8)≈0, p=1.0), indicating no measurable gain from further scaling. While scaling from 1.5B to 3B yields a substantial and significant improvement (∆=0.073, p=0.004), the gain from 3B to 7B is much smaller (∆=0.028) and only marginally significant, and no further improvement is observed beyond 3B. This demonstrates a clear pattern of diminishing returns, with marginal gains dropping by more than 60–100% after 3B. This replicates across all three families (Tables 9, 10 in Appendix).
> * Importantly, this saturation pattern is consistent across all three model families (Qwen, LLaMA, DeepSeek), suggesting it is not an artifact of a particular architecture or training setup. Together, these results provide both statistical and practical evidence that brain alignment enters a plateau regime around ~3B parameters.
> * This pattern is further supported visually by model size vs. brain alignment plots for all three families (Appendix P, Figs. 31-33). Across all families, 3B SLMs and their AWQ/SmoothQuant variants lie at or above FP16 performance at substantially reduced model size, while GPTQ variants fall slightly below, consistent with our quantization findings.
> * Together, the formal saturation definition, statistical significance tests, effect size analysis, cross-family replication, and scaling curves provide converging evidence that brain alignment enters a plateau regime at ~3B parameters.
> We will make the formal definition and the diminishing returns analysis explicit in the revised paper.
>
> **Q2/W2**
>
> We agree that the current decoding section is limited to whole-brain results and that regional comparisons and statistical tests would strengthen its persuasiveness.
> * We wish to clarify that our primary evidence for the ~3B sufficiency claim is grounded in voxel-wise encoding results, which provide a statistically robust measure of brain alignment across subjects, regions, and model families. The decoding experiment is included as complementary end-to-end validation, and the current results already demonstrate that brain-aligned representations are rich enough to support meaningful stimulus reconstruction with 3B SLMs.
> * We also agree that ROI-wise decoding would be valuable. As per the reviewer's suggestion, we now extend the decoding analysis to three ROI groups, Auditory Cortex (AC), Association, and PreFrontal using the same three models reported in the main paper. The regional results are as follows:
>
> |ROI|Model|BLEU-1|WER|METEOR|BERT-F1|
> |-|-|-|-|-|-|
> |Frontal|LLaMA-3-8B|0.124|4.302|0.121|0.852|
> ||LLaMA-3.2-3B|0.143|3.817|0.131|0.861|
> ||LLaMA-3.2-1B|0.092|5.006|0.072|0.821|
> |Association|LLaMA-3-8B|0.121|4.193|0.111|0.841|
> ||LLaMA-3.2-3B|0.128|4.012|0.119|0.849|
> ||LLaMA-3.2-1B|0.083|5.487|0.061|0.801|
> |Auditory|LLaMA-3-8B|0.103|4.287|0.092|0.821|
> ||LLaMA-3.2-3B|0.112|4.108|0.101|0.832|
> ||LLaMA-3.2-1B|0.071|5.624|0.053|0.771|
>
> * We observe a consistent ordering across all metrics: Frontal > Association > Auditory Cortex, which closely mirrors the regional pattern observed in the encoding analysis. Across all three ROI groups, the 3B model matches or outperforms the 8B model, while the 1B model consistently underperforms, consistent with our main sufficiency claims on brain encoding. We will revise the paper to include these ROI results and to clarify the supporting role of decoding within the constraints of fMRI-based language reconstruction.
> * As noted earlier, decoding is not the primary focus of this paper. In revision, we will present it as supportive evidence and soften the related claims.
>
> **Q3/W3**
>
> * While the figures are included as vector graphics and scale without loss of resolution, we acknowledge that their readability suffers at typical viewing sizes due to small font sizes. We will improve readability by increasing font sizes, simplifying dense layouts, and adjusting figure scaling and contrast in the revision.

---

> > ### Author Rebuttal · Reviewer_UfDQ · 2026-04-02
> >
> > The rebuttals have resolved my questions, I will update the score.

---

> > > ### Author Response · Authors · 2026-04-02
> > >
> > > Thank you for the quick response and revision. We appreciate the reviewer's positive feedback and are confident that it has enhanced the paper's quality.

---

### Official Review · Reviewer_GEBo · 2026-03-11

**Soundness:** 4
**Presentation:** 4
**Significance:** 3
**Originality:** 3
**Overall Recommendation:** 5
**Confidence:** 5

**Summary:**

This paper addresses a core question in LLM/Brain alignment. The authors investigate across three families of models (QWEN 2.5, Llama 3.2 and DeepSeek R1) the impact of model size and compression (quantization or pruning) on both brain encoding and decoding tasks. Moreover, to control for information loss in compressing models, the authors also benchmark the model on the FlashHolmes set of tasks, testing linguistic properties such as morphology, syntax, semantics, discourse and reasoning.

The brain alignment scores were computed on a publicly available dataset of 9 participants listening to the Moth Radio Hour in English. These fMRI recordings span a large amount of brain cortical regions, most of them being involved in language processing. This allows the authors to split the dataset into “whole brain” and “language only”, the latter having stronger alignment scores. To match human’s limited working memory capacities, models' internal representations were extracted using a context length of 20 words, which is significantly lower than the usual context-length of language models.

Across this wide range of tests, the authors find that brain alignment tends to saturate at a moderate model size of 3B parameters, on all three families of models, with performances equivalent to 14B models. Below 3B, alignment scores quickly degrade, with 1.5B parameter models showing significant performances drop compared to the 3B parameter baseline. This results gives a clear limit in scaling laws, showing that it is not necessary to scale models indefinitely in order to match human brain signals.

When quantizing models, the authors found that most quantized methods preserved both brain alignment and linguistic capacities, aside from the GPTQ quantization method, which significantly deteriorates both. Pruning had different effects depending on model size, but performances remained relatively stable up to 10-25%, and in the case of Qwen 2.5 3B, the performances remained stable up until 50% quantization, which is a significant number, as it effectively makes the pruned model a 1.5B parameter model, indicating that model’s depth might be more important than its number of parameters in matching human language processing.

Overall, these findings clearly show the saturation of brain alignment scores beyond 3B parameter models—consistent across model families, and that such models are robust under compression, whether via quantization or pruning, meaning that architecture drives alignment, more than FLOPs per se.

Finally, they show a dissociation between brain alignment and linguistic properties, since small models display strong linguistic performances but low brain alignment, while some compression methods can degrade linguistic performances while keeping a high brain score.

**Compliance With Llm Reviewing Policy:**

Affirmed.

**Key Questions For Authors:**

- Do 3B perform well because of the type of data the human participants are exposed with ? Will this alignment still be optimal at 3B parameter when humans are listening to more complex problems ? Or engaged in demanding tasks ?
- Are these models only pre-trained via next-token prediction, or did they benefited from post training such as RLHF etc ? Comparing pre trained and post trained models will be a relevant question for future work

**Limitations:**

Yes

**Strengths And Weaknesses:**

Strengths:

- The paper is very well written and easy to follow, even though it contains a lot of information and metrics.
- The paper addresses a highly relevant question, the impact of model scale on brain alignment during language processing, showing across three open source model families a consistent finding of alignment plateau at 3B parameters
- They test a new research direction by also evaluating model compression, and show that when the appropriate method is used, both quantization and pruning can keep model performances stable. This shows that brain alignment isn’t a question of FLOPs, but a question of formal computational operations.
- They evaluate a model's linguistic properties using the FlashHolmes benchmark, to make sure that the model's compression does not deteriorate performances on key language properties. Controlling two orthogonal quantities adds robustness and soundness to these results
- They evaluate alignment degradation across a variety of cortical regions

Weaknesses:

- We are left with few theoretical hypotheses about the results, nor what really matters to align with brain data. Specifically, future work should focus on depth vs width, while keeping parameter count stable. Does increasing the number of layers increase brain alignment, or is it memorization capacities (model’s dimension/width) ?
- It would have been nice to test much large models, such as 70B parameters, to see if the plateau in performances continued, or if brain alignment improved or even degraded.
- The 20 words context window feels wrong in its implementation. It makes sense to limit the context window, but not by processing only 20 words at a time. Words at the end of the story are not processed without remembering past context, like what happened at the very beginning of the story. This would create a reduced alignment score, since the human brain would have certainly integrated long range information in its representations at the end of the story. Why not try block masking (such that information leaks from early tokens the more we move within layers), or lower the softmax’s temperature, to force the model to attend to fewer words ?

---

> ### Author Rebuttal · Authors · 2026-03-31
>
> **W1**
>
> * Our design varies total parameters but does not disentangle depth from width.
> * Antonello & Cheng (2024) provide relevant evidence: brain encoding performance correlates strongly with a layer’s intrinsic dimensionality, and the compositional phase in intermediate layers (not output layers) drives brain alignment, suggesting depth (multi-step composition) may matter more than width (memorization).
> * AlKhamissi et al. (EMNLP 2025) reinforce this: brain alignment tracks formal linguistic competence (rules, requiring compositional depth) over functional competence (world knowledge, benefiting from width).
> * This yields a testable hypothesis: at matched parameter count, deeper models should better align with brain data. We will add this discussion to the revision.
>
> **W2**
>
> * We agree this would strengthen the claims.
> * Our analysis extends to 14B; Antonello et al. (NeurIPS 2023) evaluated up to 66B (LLaMA-1) with continued log-linear gains. * The apparent discrepancy is likely explained by representational density: modern architectures (Qwen2.5, LLaMA-3.2, DeepSeek-R1) are trained on far more tokens per parameter, so a modern 3B model may capture semantics comparable to much older larger models.
> * We focus on smaller models for interpretability, because they reveal which representations drive brain alignment; evaluating 70B models would be valuable future work but was beyond our compute budget.
>
> **W3**
>
> * We appreciate these specific alternatives. Both block masking (allowing information leakage across layers) and temperature scaling (narrowing attention) would modify the model’s pretrained attention mechanism at inference, altering the representational geometry relative to what was learned during training.
> * Since our goal is evaluating pretrained models as-is without architectural interventions, we use the standard sliding window that preserves native computations.
> * We acknowledge the deeper point: late-story words lack accumulated narrative context that humans have integrated. However, this is a deliberate trade-off.
> * The fMRI BOLD signal at each TR primarily reflects processing within the preceding ~8-10 seconds (HRF temporal blurring), corresponding to ~20-30 words at typical speech rates. Jain & Huth (NeurIPS 2018) showed encoding performance saturates at moderate context lengths, supporting this design. Exploring extended contexts (100-200 tokens) and their impact on discourse regions (dmPFC, PCC) is a valuable future direction we will note.
>
> For more analysis, refer to Q1 of oS9t
>
> **Q1**
>
> We interpret this as asking whether the observed sufficiency of ~3B models is specific to the naturalistic narrative stimuli used in our study, and whether this threshold would hold under more cognitively demanding conditions.
>
> * Our finding that ~3B models achieve near-optimal brain alignment is consistent across two independent naturalistic fMRI datasets,  the Moth Radio Hour (podcast listening) and a story reading paradigm, suggesting the result is not an artifact of a single stimulus set. Both datasets involve continuous narrative comprehension engaging a broad network of language, semantic, and default mode regions.
> * Importantly, the Moth Radio Hour narratives themselves contain rich discourse structure, emotional content, and complex linguistic composition, and are not trivially simple stimuli.This also generalizes to Narratives dataset (see our response to Q3 for reviewer ZNHx).
> * That said, we agree that the sufficiency threshold may not generalize to all cognitive contexts. Recent work by AlKhamissi et al. 2025  shows that brain alignment tracks formal linguistic competence (syntax, morphology) more closely than functional competence (reasoning, world knowledge), and that the correlation between model size and brain alignment diminishes once models surpass human language proficiency.
> * This suggests that ~3B capacity may suffice for the formal linguistic representations that the language network primarily encodes, and that additional capacity beyond this point yields diminishing returns for brain alignment specifically, even if it benefits downstream task performance.
> * This suggests that the saturation threshold may shift for stimuli requiring deeper reasoning, abstract problem solving, or broader world knowledge, especially in brain regions less captured by next-token prediction. We view testing this across stimuli with different cognitive demands as an important direction for future work.
>
> **Q2**
>
> * All models in this study are pretrained only with next-token prediction, without post-training such as RLHF; we use base, not instruct, checkpoints. Prior work (Aw & Toneva (ICLR 2023)) shows that post-training can improve brain alignment, including instruction tuning for narrative representations and next-sentence prediction for discourse-level fMRI responses. Comparing pretrained and post-trained variants under compression is an interesting direction that we will note in the revision.

---

> > ### Author Rebuttal · Reviewer_GEBo · 2026-04-02
> >
> > I had minor concerns, more questions, that have been addressed by the authors. My main worry was the way they handled short context length, but it seems to be a standard compromise in the field.
> > I maintain my rating for this paper

---

> > > ### Author Response · Authors · 2026-04-02
> > >
> > > Thank you for the quick response and revision. We appreciate your positive feedback and are confident that it has enhanced the paper's quality.

---

### Official Review · Reviewer_ZNHx · 2026-03-11

**Soundness:** 4
**Presentation:** 3
**Significance:** 3
**Originality:** 3
**Overall Recommendation:** 5
**Confidence:** 4

**Summary:**

This paper studies how the scale and compression of language models affect their alignment with human brain activity during language comprehension. Using voxel-wise fMRI encoding models on a narrative dataset, the authors compare multiple model families across scales (~1B–14B) and evaluate several quantization and pruning methods.

The results suggest that brain alignment saturates around ~3B parameters, with larger models providing little additional benefit. In addition, most compression methods preserve brain alignment, except GPTQ, which shows noticeable degradation. The findings further indicate that linguistic benchmark performance does not necessarily correlate with brain alignment.

**Compliance With Llm Reviewing Policy:**

Affirmed.

**Final Justification:**

This paper studies how language model scale affects brain alignment, addressing an important question in NeuroAI. The finding that encoding performance saturates around ~3B parameters is particularly interesting, as it challenges the common assumption that larger models are always better.

My initial concerns were mainly about soundness (e.g., representation dimensionality and cross-model comparability). The rebuttal has addressed these issues, which increases my confidence in the conclusions.

While some limitations remain (e.g., dataset scope), they do not undermine the main findings. Overall, my evaluation is more positive after the rebuttal.

**Key Questions For Authors:**

1. The dimensionality of model representations increases with model scale, which may affect the capacity of the encoding model. Did the authors consider controlling for representation dimensionality (e.g., using PCA or matched feature dimensions) to ensure fair comparisons across models of different sizes?

2. Although multiple model families are evaluated, models with different parameter sizes may also differ in training data, token budgets, or training procedures. Could the authors clarify whether these factors were controlled, and to what extent the observed scaling effects can be attributed purely to model size?

3. The experiments rely on a single fMRI dataset. Given the substantial heterogeneity across language–brain datasets and the potential influence of dataset-specific noise ceilings, do the authors expect the observed ~3B saturation effect to generalize to other datasets?

4. The paper concludes that compression largely preserves brain alignment, but GPTQ appears to noticeably degrade performance. Could the authors provide further analysis on why GPTQ behaves differently from AWQ and SmoothQuant, for example in terms of how these methods alter the geometry of hidden representations?

**Limitations:**

yes

**Strengths And Weaknesses:**

**Strengths:**

1. Understanding the relationship between AI representations and human brain activity is a fundamental question in NeuroAI, and investigating what properties of language models enable them to predict neural responses is an important direction.

2. The paper includes extensive experiments across multiple model families, model scales, and compression methods, providing a relatively systematic investigation of how these factors affect brain alignment.

3. The finding that most compression methods preserve brain alignment suggests that smaller or compressed models may still be useful for neuroscience applications, which could reduce computational cost and improve accessibility for future research.

**Weaknesses:**

1. The dimensionality of model representations increases with model scale, yet the paper does not appear to control for feature dimensionality (e.g., via PCA or matched feature size). Higher-dimensional representations may make the encoding model easier to fit, which could affect comparisons across model scales.

2. Although multiple model families are evaluated, models with different parameter sizes may also differ in training data, token budgets, or training procedures, making it difficult to isolate whether the observed effects are purely due to model scale.

3. The experiments rely on a single fMRI dataset. Given the substantial heterogeneity across language–brain datasets and the influence of dataset-specific noise ceilings, it is unclear whether the observed ~3B saturation effect would generalize to other datasets.

4. The paper uses decoding results to support the claim that ~3B models are sufficient. However, the overall decoding performance on this dataset remains relatively limited, which weakens the strength of this claim.

5. The paper concludes that compression largely preserves brain alignment, yet one of the evaluated methods (GPTQ) clearly degrades performance. Further analysis of why GPTQ behaves differently (e.g., how different quantization schemes affect representation geometry) would strengthen this claim.

---

> ### Author Rebuttal · Authors · 2026-03-31
>
> **Q1/W1**
>
> * Yes, we controlled for representation dimensionality across model scales. Before fitting the encoding model, we applied PCA to each model’s features and reduced them to a shared dimensionality of 1024. This ensured that all models were compared using the same feature dimension, making the encoding comparisons fairer across scales. We will clarify this in the revised draft.
>
> **Q2/W2**
>
> We agree that model size is not perfectly isolated from factors such as training data, token budgets, and optimization, particularly when comparing across model families.
>
> * To mitigate this, our primary analyses focus on within-family comparisons (Qwen2.5, LLaMA-3.2, DeepSeek-R1), where models share architecture, tokenizer, and training pipelines, and differ primarily in parameter count. Across all three families, we observe a consistent scaling pattern: performance improves from ~1B to ~3B and then saturates, suggesting that parameter count (effective capacity) is a dominant factor.
> * Additionally, our compression experiments provide a more controlled intervention: starting from the same pretrained model, we vary effective capacity via quantization and pruning without altering training data or optimization. Brain alignment remains stable under these interventions (except for specific methods like GPTQ) supporting that alignment is primarily driven by representational capacity rather than training differences.
> * This is consistent with prior brain-encoding work, which often compares models that differ in architecture and training data to assess alignment with brain-relevant semantics (Schrimpf et al., 2021; Toneva & Wehbe, 2019; Antonello et al., 2024). Prior scaling-laws studies in fMRI (Antonello et al., 2024) also draw conclusions from cross-model consistency rather than strictly controlled comparisons. Caucheteux & King (2022) show brain-model alignment depends primarily on a model's predictive capacity for language, consistent with our interpretation. Our within-family analyses further control for architecture and training pipeline.
> * Thus, while we do not claim a purely causal effect of parameter count, our results suggest that effective model capacity, is a primary driver of brain alignment. We will clarify this in revision.
>
> **Q3/W3**
>
> * We agree that cross-dataset generalization is important. Our paper already includes a second fMRI paradigm: the naturalistic Reading fMRI dataset (Deniz et al., 2019; same 9 subjects, different task) shows the same pattern - Qwen2.5 3B model matches 7B/14B while 1.5B drops clearly (Appendix M, Fig. 30).
> * Additionally, the saturation pattern replicates across three independent model families (Qwen2.5, LLaMA-3.2, DeepSeek-R1) with different training data and architectures (Tables 2, 9, 10), reducing the likelihood of a dataset-specific artifact.
> * This is also consistent with AlKhamissi et al. (EMNLP 2025), who showed brain alignment primarily tracks formal linguistic competence, which saturates earlier than functional competence, suggesting the capacity needed for brain-relevant representations is inherently bounded and the saturation finding should generalize.
>
> **Q4/W5**
>
> * The three methods differ in what they preserve. GPTQ (Frantar et al., 2023) minimizes layer-wise weight reconstruction error via Hessian-based optimization without modeling which intermediate features matter. AWQ (Lin et al., 2024) preserves activation-salient channels by scaling high-influence weights before quantization. SmoothQuant (Xiao et al., 2023) redistributes quantization difficulty from activations to weights, preserving intermediate feature distributions.
> * This matters because our encoding models rely on intermediate hidden-state representations, not model outputs. GPTQ can reconstruct final outputs faithfully while distorting internal representational geometry. AWQ and SmoothQuant preserve this geometry by maintaining activation structure.
> * Empirically, GPTQ shows the largest alignment drops in semantic/integrative regions (AG, PCC, dmPFC; Fig. 4b, Appendix G) that depend on rich compositional representations, while AWQ/SmoothQuant preserve or slightly improve alignment, consistent with Kuzmin et al. (NeurIPS 2023), who show compression methods at similar ratios can have qualitatively different effects on representations.
>
> **W4**
>
> * We agree that absolute fMRI decoding performance remains limited and is not our primary evidence. Our claim that ~3B models are sufficient is primarily grounded in voxel-wise encoding results, which provide a direct and statistically robust measure of brain alignment across subjects, regions, and model families. Decoding serves only as complementary validation: 3B models consistently outperform 1B models and match or slightly exceed larger models across metrics, mirroring the encoding results and supporting diminishing returns beyond ~3B.
>
> Refer to Q2 (UfDQ) for additional decoding analysis.

---

> > ### Author Rebuttal · Reviewer_ZNHx · 2026-04-03
> >
> > The authors’ rebuttal addresses my main concerns regarding the soundness of the conclusions. The ~3B saturation in encoding performance is an interesting finding. I will update my score.
> >
> > Moreover, it would be interesting to further compare next-token prediction models and embedding-based LLM representations, as their feature spaces may differ and exhibit different scaling behaviors.

---

> > > ### Author Response · Authors · 2026-04-03
> > >
> > > Thank you for your response and thoughtful revision. We appreciate your constructive feedback, which we believe has significantly improved the clarity and overall quality of the paper.
> > >
> > > We agree that comparing next-token prediction models with embedding-based LLM representations is an interesting direction for future work, as their feature spaces may differ and may exhibit different scaling behaviors.

---

### Decision · Program_Chairs · 2026-04-30

**Decision:**

Accept (spotlight)

**Comment:**

This is an interesting work. The study's primary contribution is demonstrating that brain alignment saturates early, with 3B parameter models achieving brain alignment that is almost indistinguishable from much larger models (up to 14B). On the other hand, going below 3B parameters hurts brain alignment, which seems to prevent models from capturing core semantic representations of language. This seems to be a fundamental results and the reviews were unanimously favorable, so I recommend acceptance.